# Deep SPI: Safe Policy Improvement via World Models

**Florent Delgrange**
AI Lab, Vrije Universiteit Brussel
Flanders Make

**Raphael Avalos**
AI Lab, Vrije Universiteit Brussel
Cohere

**Willem Röpke**
AI Lab, Vrije Universiteit Brussel
Cohere

## Abstract

Safe policy improvement (SPI) offers theoretical control over policy updates, yet existing guarantees largely concern offline, tabular reinforcement learning (RL). We study SPI in general online settings, when combined with world model and representation learning. We develop a theoretical framework showing that restricting policy updates to a well-defined neighborhood of the current policy ensures monotonic improvement and convergence. This analysis links transition and reward prediction losses to representation quality, yielding online, "deep" analogues of classical SPI theorems from the offline RL literature. Building on these results, we introduce `DeepSPI`, a principled on-policy algorithm that couples local transition and reward losses with regularised policy updates. On the ALE-57 benchmark, `DeepSPI` matches or exceeds strong behaviorals, including PPO and `DeepMDPs`, while retaining theoretical guarantees.

## 1 Introduction

*Reinforcement learning* (RL) trains agents to act in complex environments through trial and error (Sutton and Barto, 2018). To scale to high-dimensional domains, modern approaches rely on function approximation, making *representation learning* essential for learning latent spaces where behaviorally similar states are grouped and policies and value functions are easier to estimate (Echchahed and Castro, 2025). A complementary approach is *model learning*, where a predictive model of the environment is trained (Ha and Schmidhuber, 2018). Such models support planning, latent simulation, and improving value estimates (Hafner et al., 2021; Schrittwieser et al., 2020; Xiao et al., 2019).

In the online setting, where the agent updates its policy during interaction, avoiding catastrophic errors is critical. Two key challenges arise: *out-of-trajectory (OOT) world models* and *confounding policy updates*. OOT issues arise when the world model fails to capture rarely visited regions of the state space, leading to unreliable predictions and unsafe updates when the latent policy explores these regions (Suau et al., 2024). Confounding updates occur when both the policy and its underlying representation are updated simultaneously: poor representations can lock the agent into suboptimal behavior, while the policy itself prevents corrective updates to the representation. *Safe Policy Improvement* (SPI) mitigates such risks by ensuring that new policies are not substantially worse than their predecessors (Thomas et al., 2015). Classical SPI methods provide rigorous results in tabular MDPs but depend on exhaustive state–action coverage, making them unsuitable for continuous or high-dimensional spaces.

We address this gap by connecting representation and model learning with safe policy improvement in general state spaces. Our contributions are threefold. First, we introduce a neighborhood operator that constrains policy updates and guarantees convergence. Second, we combine this operator with principled model losses to bound the gap between a policy's performance in the world model and in the true environment, thereby enabling safe policy improvement in complex MDPs. This analysis also shows that our scheme enforces representation quality by ensuring that states with similar values remain close in the learned latent space. Third, we connect our theory to PPO (Schulman et al., 2017)

and propose `DeepSPI`, a practical algorithm that achieves strong empirical performance on the Arcade Learning Environment (ALE; Bellemare et al. 2013) while retaining theoretical guarantees.

## 1.1 RELATED WORK

**Regularizing policy improvements.** Regularized updates, as in TRPO, PPO, and related analyses, are now standard for stabilizing policy optimization (Schulman et al., 2015; 2017; Geist et al., 2019; Kuba et al., 2022). Our work extends this perspective to the joint training of a world model and a representation, where we constrain policy updates in a principled neighborhood while controlling model quality through transition and reward losses.

**SPI** methods provide principled guarantees on policy updates from fixed datasets (offline RL) (Thomas et al., 2015; Ghavamzadeh et al., 2016a; Laroche et al., 2019; Simão et al., 2020; Castellini et al., 2023). These methods assume tabular state spaces and offline data, where error bounds must hold globally across all state–action pairs, often via robust MDP formulations (Iyengar, 2005; Nilim and Ghaoui, 2005). Our setting is fundamentally different: we study *online* RL with high-dimensional inputs, where such global constraints are intractable. We take inspiration from the SPI literature but introduce local, on-policy losses that make safe improvement feasible in practice. In spirit, other model-based methods share the goal of providing SPI-like guarantees in more general settings, but are again purely offline, omit any form of representation learning, and rely on assumptions that differ substantially from ours (Yu et al., 2020; 2021; Kidambi et al., 2020).

**Representation learning and model-based RL.** Auxiliary transition and reward prediction losses are central to many model-based methods, from `DeepMDP` to `Dreamer` and related world-model approaches (Gelada et al., 2019; Hafner et al., 2021). In particular, the losses we consider for learning transitions and rewards generalize a wide range of objectives used across the model-based RL literature (François-Lavet et al., 2019; van der Pol et al., 2020; Kidambi et al., 2020; Delgrange et al., 2022; Dong et al., 2023; Alegre et al., 2023). Conceptually, our representation guarantees are closely connected to classical notions of state abstraction in MDPs (Li et al., 2006) and to *bisimulation* (Larsen and Skou, 1991; Desharnais et al., 1998). Building on bisimulation, prior work develops representations that group states into areas in which the agent is guaranteed to behave similarly under the current policy (Castro, 2020; Zhang et al., 2021; Castro et al., 2021; Agarwal et al., 2021a; Avalos et al., 2024). By contrast, we directly link representation quality and model accuracy to our safe policy improvement analysis, yielding tractable guarantees in the online setting.

## 2 BACKGROUND

In the following, given a measurable space $\mathcal{X}$, we write $\Delta(\mathcal{X})$ for the set of distributions over $\mathcal{X}$. For any distribution $\mu \in \Delta(\mathcal{X})$, we denote by $\text{supp}(\mu)$ its support.

**Markov Decision Processes** (MDPs) offer a formalism for sequential decision-making under uncertainty. Formally, an MDP is a tuple of the form $\mathcal{M} = \langle \mathcal{S}, \mathcal{A}, P, R, s_I, \gamma \rangle$ consisting of a set of states $\mathcal{S}$, actions $\mathcal{A}$, a transition function $P : \mathcal{S} \times \mathcal{A} \to \Delta(\mathcal{S})$, a bounded reward function $R : \mathcal{S} \times \mathcal{A} \to \mathbb{R}$ with $\|R\|_\infty = R_{\text{MAX}}$, an initial state $s_I \in \mathcal{S}$, and a discount factor $\gamma \in [0, 1)$. Unless otherwise stated, we generally assume that $\mathcal{S}$ and $\mathcal{A}$ are compact. An agent interacting in $\mathcal{M}$ produces *trajectories*, i.e., infinite sequences of states and actions $(s_t, a_t)_{t \geq 0}$ visited along the interaction so that $s_0 = s_I$ and $s_{t+1} \sim P(\cdot \mid s_t, a_t)$ for all $t \geq 0$.

At each time step $t$, the agent selects an action according to a (stationary) *policy* $\pi : \mathcal{S} \to \Delta(\mathcal{A})$ mapping states to distributions over actions. Running an MDP under $\pi$ induces a unique probability measure $\mathbb{P}_\pi$ over trajectories (Revuz, 1984), with associated expectation operator $\mathbb{E}_\pi$; we write $\mathbb{E}_\pi[\cdot \mid s_0 = s]$ when the initial state is fixed to $s \in \mathcal{S}$. A policy has *full support* if $\text{supp}(\pi(\cdot \mid s)) = \mathcal{A}$ for all $s \in \mathcal{S}$, and we denote the set of all policies by $\Pi$. A *stationary measure* of $\pi$ is a distribution over states visited under $\pi$, and is defined as a solution of $\xi_\pi(\cdot) = \mathbb{E}_{s \sim \xi_\pi} \mathbb{E}_{a \sim \pi(\cdot \mid s)}[P(\cdot \mid s, a)]$. Such a measure is often assumed to exist in continual RL (Sutton and Barto, 2018), is *unique* in episodic RL (Huang, 2020), and defines the *occupancy measure* in discounted RL (Metelli et al., 2023).[1]

---

[1]Details on the formalization of episodic processes and value functions can be found in Appendix A.

**Value functions.** The performance of the agent executing a policy $\pi \in \Pi$ in each single state $s \in \mathcal{S}$ can be evaluated through the *value function* $V^\pi(s) = \mathbb{E}_\pi \left[ \sum_{t=0}^\infty \gamma^t R(s_t, a_t) \mid s_0 = s \right]$. The goal of an agent is to maximize the *return* from the initial state, given by $\rho(\pi, \mathcal{M}) = V^\pi(s_I)$. To evaluate the quality of any action $a \in \mathcal{A}$, we consider the *action value function* $Q^\pi(s, a) = R(s, a) + \gamma \, \mathbb{E}_{s' \sim P(\cdot|s,a)} V^\pi(s')$, being the unique solution of Bellman's equation with $V^\pi(s) = \mathbb{E}_{a \sim \pi(\cdot|s)} Q^\pi(s, a)$. Alternatively, any given action can be evaluated through the *advantage function* $A^\pi(s, a) = Q^\pi(s, a) - V^\pi(s)$, giving the advantage of selecting an action over the current policy.

**Representation learning in RL.** In realistic environments, the state–action space is too large for tabular policies or value functions. Instead, deep RL employs an encoder $\phi \colon \mathcal{S} \to \overline{\mathcal{S}}$ that maps states to a tractable *latent space* $\overline{\mathcal{S}}$, from which value functions can be approximated. Learning such encoders is referred to as *representation learning* (Echchahed and Castro, 2025). To improve representations, agents are often trained with additional objectives, commonly *auxiliary tasks* requiring predictive signals. Policy-based methods then optimize a *latent policy* $\overline{\pi} \colon \overline{\mathcal{S}} \to \Delta(\mathcal{A})$ jointly with $\phi$, executed in the environment as $\overline{\pi}(\cdot \mid \phi(s))$. By convention, we write $\overline{\pi}(\cdot \mid s)$ for $\overline{\pi} \circ \phi(s)$ when $\phi$ is clear, and denote the set of all latent policies by $\overline{\Pi}$. For any $\overline{\pi} \in \overline{\Pi}$, the composed policy $\overline{\pi} \circ \phi$ belongs to $\Pi$.

**Model-based RL** augments policy learning with a *world model* $\overline{\mathcal{M}} = \langle \overline{\mathcal{S}}, \mathcal{A}, \overline{P}, \overline{R}, \overline{s}_I, \gamma \rangle$, which can improve (i) sample efficiency by generating trajectories (e.g., Hafner et al. 2021), (ii) value estimation through planning (e.g., Buckman et al. 2018), and (iii) representation learning by grouping states with similar behavior (e.g., Gelada et al. 2019; Zhang et al. 2021). When $\overline{\mathcal{S}} = \mathcal{S}$, the model must replicate environment dynamics, which is often intractable. Instead, we focus on $\overline{\mathcal{S}}$ defined by the learned representation $\phi$, so that $\overline{\mathcal{M}}$ becomes an abstraction of $\mathcal{M}$. Learning transition and reward functions then additionally serves as an auxiliary signal for the representation, encouraging states with similar behavior to map close in $\overline{\mathcal{S}}$. Since $\overline{\mathcal{S}}$ is the latent space, $\overline{\Pi}$ corresponds to the policies of $\overline{\mathcal{M}}$. We further assume $\overline{\mathcal{S}}$ is equipped with a metric $\overline{d} \colon \overline{\mathcal{S}} \times \overline{\mathcal{S}} \to [0, \infty)$ to measure distances.

# 3 NO WAY HOME: WHEN WORLD MODELS AND POLICIES GO OUT OF TRAJECTORIES

World models are usually learned toward minimizing a **reward loss** $L_R$ and/or **transition loss** $L_P$ from experiences $\eta$ collected along the agent's trajectories. Those experiences are either gathered in the form of a *batch* or a *replay buffer* $\mathcal{B}$. In general, the loss functions take the following form: $L_R = \mathbb{E}_{\eta \sim \mathcal{B}} \, f_R\big(\phi, \overline{R}; \eta\big)$ and $L_P = \mathbb{E}_{\eta \sim \mathcal{B}} \, f_P\big(\phi, \overline{P}; \eta\big)$, where $f_R$ (resp. $f_P$) assign a "cost" relative to the error between $R$ and $\overline{R}$ (resp. $P$ and $\overline{P}$) according to the experiences $\eta$ and their representation. Henceforth, we refer to the policy $\pi_b$ used to insert experiences in $\mathcal{B}$ as the **behavioral policy**.

## 3.1 OUT-OF-TRAJECTORY WORLD MODEL

One may consider leveraging the model $\overline{\mathcal{M}}$ to improve the policy $\pi_b$. This can be achieved by directly planning a new policy $\overline{\pi}$ in $\overline{\mathcal{M}}$ or drawing imagined trajectories in the world model to evaluate new actions and improve on sample complexity during RL. However, since the world model is learned from experiences stored in $\mathcal{B}$, we can only be certain of its average accuracy according to this data. This is problematic because some regions of the state space of $\mathcal{M}$ may have been rarely, or not at all, visited under $\pi_b$. In that case, the predictions made in $\overline{\mathcal{M}}$ might cause the agent to "hallucinate" inaccurate trajectories in the latent space and spoil the policy improvement. This problem, known as the **out-of-trajectory** (OOT) issue (Suau et al., 2024), arises when a policy in $\overline{\mathcal{M}}$ deviates substantially from $\pi_b$, which can render the model unreliable.

To illustrate this problem, consider the world model of Figure 1. Assume the model is trained by collecting trajectories produced by $\pi_b$ in $\mathcal{M}$ where $\pi_b(a_2 \mid s) \leq \epsilon$ for all $s \in \mathcal{S}_1$, with $\epsilon > 0$. For a sufficiently small $\epsilon$, the region $\mathcal{S}_3$ in the original environment would remain largely unexplored while having almost no impact on the losses $L_R, L_P$. Therefore, the representation of states in $\mathcal{S}_3$ ($\overline{s}_3$ and $\overline{s}_3'$) may turn completely inaccurate. Here, the model incorrectly assigns a reward of 20 to $\overline{s}_3'$, whereas the true reward is strictly negative. Consequently, the optimal policy in $\overline{\mathcal{M}}$ deterministically selects $a_2$ in $\overline{s}_1$. When executed in the original environment, this policy drives the agent to $\mathcal{S}_3$ thereby degrading the behavioral policy $\pi_b$.

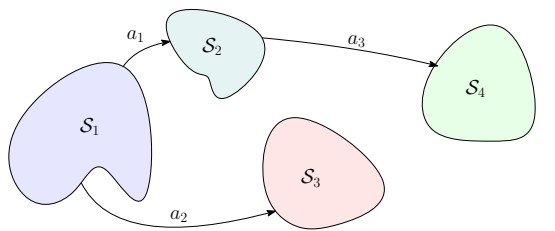

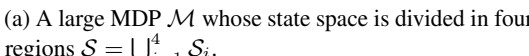

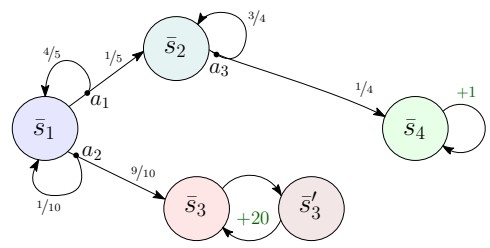

(a) A large MDP $\mathcal{M}$ whose state space is divided in four regions $\mathcal{S} = \bigcup_{i=1}^{4} \mathcal{S}_i$.

(b) A simple world model $\overline{\mathcal{M}}$ whose state space is $\overline{\mathcal{S}} = \{\bar{s}_1, \bar{s}_2, \bar{s}_3, \bar{s}_3', \bar{s}_4\}$.

Figure 1: In $\mathcal{M}$, continuously playing $a_1$ in states from $\mathcal{S}_1$ eventually leads the agent to the region $\mathcal{S}_2$, and playing $a_3$ in $\mathcal{S}_2$ eventually leads the agent to $\mathcal{S}_4$ where a reward of 1 is incurred at each time step, whatever the action played. Playing $a_2$ in $\mathcal{S}_1$ leads the agent to the region $\mathcal{S}_3$, where all actions incur negative rewards. Here, $\phi(s) = \bar{s}_i$ for any $s \in \mathcal{S}_i$ and $i = \{1, 2, 4\}$. For $s \in \mathcal{S}_3$, we have either $\phi(s) = \bar{s}_3$ or $\phi(s) = \bar{s}_3'$.

## 3.2 CONFOUNDING POLICY UPDATE

Updating both the representation and the policy solely from experience collected under a behavioral policy can *degrade* performance rather than improve it. In the same spirit as *policy confounding* (Suau et al., 2024), we call this phenomenon **confounding policy update**. The MDP in Figure 2 illustrates the issue.

The agent maps the states $s_2$ and $s_3$ to the *same* latent state $\bar{s}$, i.e. $\phi(s) = \bar{s}$ iff $s \in \{s_2, s_3\}$. States $s_1$ and $s_4$ each have their own latent state. We consider the behavioral policy $\pi_{\mathrm{b}} := \bar{\pi}_b \circ \phi$, where $\bar{\pi}_b$ is a stochastic policy with a small exploration rate $\zeta$:

$$\bar{\pi}_b(a_1 \mid \bar{s}) = 1 - \zeta, \qquad \bar{\pi}_b(a_2 \mid \bar{s}) = \zeta, \qquad (1)$$

for $0 < \zeta \ll \epsilon$. A good representation would ideally group states from which the agent behaves similarly. Because trajectories that reach $s_3$ *and* pick $a_2$ are unlikely, the two states appear identical under $\pi_{\mathrm{b}}$: $|V^{\pi_{\mathrm{b}}}(s_2) - V^{\pi_{\mathrm{b}}}(s_3)| \approx 0$. Therefore, this justifies using $\phi$ as representation for $\pi_{\mathrm{b}}$, because the values of $s_2$ and $s_3$ are nearly identical: the agent exhibits close behaviors under $\pi_{\mathrm{b}}$ from those states.

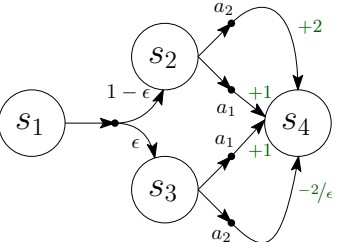

Figure 2: MDP where the probability of transitioning from $s_1$ to $s_2$ is $1 - \epsilon$, for $0 < \epsilon < 1/4$.

Suppose exploration under $\bar{\pi}_b$ eventually discovers that playing $a_2$ in $\bar{s}$ sometimes yields the $+2$ reward. Based on exploration data, an RL agent might therefore be tempted to change the latent policy to $\bar{\pi}(a_2 \mid \bar{s}) = 1$ *without modifying* the representation $\phi$. With the representation still grouping $s_2$ and $s_3$, the new policy would now deterministically pick $a_2$ in *both concrete* states. Whenever the agent actually reaches $s_3$, it would receive the large negative reward $-2/\epsilon$, which turns the overall return (from $s_1$) negative, thus *worse* than under $\pi_{\mathrm{b}}$ even though $a_2$ is indeed optimal in $s_2$.

A solution to this problem would have been to split the representation of $s_2$ and $s_3$ in two distinct latent states. In general, representation and policy learning must be *coupled* since any change in the policy that alters the distribution over states can invalidate a previously adequate representation. However, in this example, the agent has no incentive to do so based on the experiences collected under $\pi_{\mathrm{b}}$. As we will show below, updating both the policy and the representation jointly should be handled carefully to ensure *policy improvement*.

Our goal is to *establish sufficient conditions* to guarantee **safe policy improvement** during the RL process, either based on world models, state representations, or both, thus alleviating OOT world model and confounding policy update issues. Notice that, in the examples, both problems occur when performing *aggressive* updates from $\pi_{\mathrm{b}}$ to a new policy $\bar{\pi}$ (the mode of the distributions drastically shifts). Intuitively, *smooth* updates indeed ensure to alleviate those issues: constraining the policy search to policies "close" to $\pi_{\mathrm{b}}$ (i) prevents hallucinations in parts of the world model that have been underexplored; (ii) reduces the risk of significantly degrading the return when updating the policy. While the benefits of regularizing policy improvements have already been both theoretically and practically justified (e.g., Geist et al. 2019; Kuba et al. 2022), their implications when mixing model-based and representation learning in RL have been underexplored.

**Roadmap.** To rigorously address the OOT and confounding-update issues, the next sections develop the theoretical foundations of our approach, showing how controlled policy updates, local model losses, and representation stability interact. We briefly summarize how the main results connect.

Our analysis combines **neighborhood-restricted policy updates**, **model-quality bounds**, and **representation guarantees**. Sect. 4 introduces the neighborhood operator defining a trust region around the behavioral policy; restricting updates to this region ensures monotonic improvement and convergence (Thm. 1). Sect. 5 then links the reward and transition losses to value discrepancies: Thm. 2 shows that, when these losses are small, and updates remain in the neighborhood, the world model stays accurate under the learned representation. Combining these ingredients yields our first SPI result (Thm. 3), guaranteeing that direct policy updates in the world model translate to improvement under controlled error. Finally, Thm. 4 shows that the same loss-based control stabilizes the encoder, ensuring that value-distinct states remain separated in the latent space.

## 4 YOUR FRIENDLY NEIGHBORHOOD POLICY

Motivated by the intuition that constraining policy updates can mitigate OOT and confounding policy issues, we consider measuring the update as the **importance ratio** (IR) of the policies. This measure provides guarantees for constraining policy and representation updates, and with an appropriate optimisation scheme, ensures both policy improvement and convergence. In Section 5, we will further show that properly constraining the IR allows for safe policy improvements in world models while providing representation guarantees.

Let $\pi, \pi' \in \Pi$, the **extremal importance ratios** are defined as $D_{\text{IR}}^{\text{ext}}(\pi, \pi') = \text{ext}\left\{\pi'(a|s)/\pi(a|s) \colon s \in \mathcal{S}, a \in \text{supp}(\pi(\cdot \mid s))\right\}$, where $\text{ext} \in \{\inf, \sup\}$. We define a **neighborhood operator**[2] based on the IR, $\mathcal{N}^C \colon \Pi \to 2^\Pi$ for some constant $1 < C < 2$, establishing a trust region for policies updates that constraints the IR between $2 - C$ and $C$:

$$\mathcal{N}^C(\pi) = \left\{\pi' \in \Pi \;\middle|\; \begin{array}{l} 2 - C \leq D_{\text{IR}}^{\inf}(\pi, \pi') \leq D_{\text{IR}}^{\sup}(\pi, \pi') \leq C, \\ \text{and } \text{supp}(\pi(\cdot \mid s)) = \text{supp}(\pi'(\cdot \mid s)) \quad \forall s \in \mathcal{S} \end{array}\right\} \qquad \forall \pi \in \Pi. \quad (2)$$

A critical question is whether an agent that restricts its policy updates to a defined neighborhood is truly following a sound **policy improvement** scheme. The following theorem shows that it does and further guarantees convergence.

**Theorem 1.** (Policy improvement and convergence guarantees) *Assume $\mathcal{S}$ and $\mathcal{A}$ are finite spaces. Let $\pi_0 \in \Pi$ be a policy with full support and $(\pi_n)_{n \geq 0}$ be a sequence of policy updates defined as*

$$\pi_{n+1} \coloneqq \arg\sup_{\pi' \in \mathcal{N}^C(\pi_n)} \mathbb{E}_{s \sim \mu_{\pi_n}} \mathbb{E}_{a \sim \pi'(\cdot|s)} A^{\pi_n}(s, a), \quad (3)$$

*where $\mu_{\pi_n}$ is a sampling distribution with $\text{supp}(\mu_{\pi_n}) = \mathcal{S}$ for each $n \geq 0$. Then, the value function $V^{\pi_n}$ is monotonically improving, converges to $V^*$, and so is the return $\rho(\pi_n, \mathcal{M})$.*

The proof consists in showing the resulting policy update scheme is an instance of *mirror learning* (Kuba et al., 2022), which yields the guarantees. Notice that since $\pi_0$ has full support, all the subsequent policies $\pi_n$ have full support as well. To maintain the guarantees, considering a stationary measure $\xi_{\pi_n}$ as the sampling distribution is only possible when $\text{supp}(\xi_{\pi_n}) = \mathcal{S}$. Note that this is always the case in episodic tasks (as the policy itself has full support). This is more generally true in ergodic MDPs (Puterman, 1994).

## 5 WITH GREAT WORLD MODELS COMES GREAT REPRESENTATION

This section explains how the neighborhood operator of Eq. 2 enables safe policy improvement during world-model planning and representation updates in complex environments. Standard SPI methods ignore representation learning and require exhaustive state–action coverage in $\mathcal{B}$ to obtain guarantees, making them unsuitable for general state-action spaces. Even in finite domains, bounding the count

---

[2]There are clear similarities between the IR, our neighborhood operator, and the PPO loss function (Schulman et al., 2017). We discuss this connection in Section 6.

of each state–action pair does not scale. Laroche et al. (2019) proposed *baseline bootstrapping* for under-sampled pairs, but their approach remains impractical in large-scale settings despite conceptual similarities to our operator. Further discussion of SPI limitations is provided in Appendix D.

**Learning an accurate world model.** SPI typically relies on optimizing a policy with respect to a latent model learned from the data stored in $\mathcal{B}$. In contrast to previous methods, our approach scales to high-dimensional feature spaces by (i) learning a representation $\phi$ and (ii) considering **local error measures** as opposed to global measures across the whole state-action space. We formalize them as tractable *loss functions*. Their local nature makes them compliant with stochastic gradient descent methods. Formally, given a distribution $\mathcal{B} \in \Delta(\mathcal{S} \times \mathcal{A})$, we define the *reward loss* $L_R^{\mathcal{B}}$ and the *transition loss* $L_P^{\mathcal{B}}$ as

$$L_R^{\mathcal{B}} \coloneqq \mathbb{E}_{s,a \sim \mathcal{B}} \left| R(s,a) - \overline{R}(\bar{s}, a) \right|, \qquad L_P^{\mathcal{B}} \coloneqq \mathbb{E}_{s,a \sim \mathcal{B}} \, \mathcal{W}\left( \phi_\sharp P(\cdot \mid s, a), \overline{P}(\cdot \mid \phi(s), a) \right) \quad (4)$$

where $\phi_\sharp P$ is the *pushforward measure* of $P$ by $\phi$, and $\mathcal{W}$ the *Wasserstein distance* (Vaserstein, 1969). $\mathcal{W}$ between $\mu, \nu \in \Delta(\overline{\mathcal{S}})$ is defined as $\mathcal{W}(\mu, \nu) = \inf_{\lambda \in \Lambda(\mu, \nu)} \mathbb{E}_{(\bar{s}, \bar{s}') \sim \lambda} \, \bar{d}(\bar{s}, \bar{s}')$, where $\Lambda(\mu, \nu)$ is the set of all couplings of $\mu$ and $\nu$. While the Wasserstein operator may seem scary at first glance, it generalizes over transition losses that can be found in the literature (cf. Sect. 1.1). In particular, when the latent space is discrete, this distance boils down to the *total variation distance*. Another notable case is when the transition dynamics are deterministic, in which case the transition loss reduces to $L_P^{\mathcal{B}} = \mathbb{E}_{s,a,s' \sim \mathcal{B}} \, \bar{d}\left( \phi(s'), \overline{P}(\phi(s), a) \right)$. Finally, in general, a tractable upper bound can be obtained as $L_P^{\mathcal{B}} \leq \mathbb{E}_{s,a,s' \sim \mathcal{B}} \, \mathbb{E}_{\bar{s}' \sim \overline{P}(\cdot | \phi(s), a)} \, \bar{d}(\phi(s'), \bar{s}')$ (proof in Appendix C).

**Lipschitz constants.** To provide the guarantees, for any particular policy $\bar{\pi} \in \overline{\Pi}$, we assume the world model is equipped with *Lipschitz constants* $K_{\overline{R}}^{\bar{\pi}}$, $K_{\overline{P}}^{\bar{\pi}}$ defined as follows: for all $\bar{s}_1, \bar{s}_2 \in \overline{\mathcal{S}}$,

$$\left| \mathbb{E}_{a_1 \sim \bar{\pi}(\cdot | \bar{s}_1)} \overline{R}(\bar{s}_1, a_1) - \mathbb{E}_{a_2 \sim \bar{\pi}(\cdot | \bar{s}_2)} \overline{R}(\bar{s}_2, a_2) \right| \leq K_{\overline{R}}^{\bar{\pi}} \cdot \bar{d}(\bar{s}_1, \bar{s}_2),$$

$$\mathcal{W}\left( \mathbb{E}_{a_1 \sim \bar{\pi}(\cdot | \bar{s}_1)} \overline{P}(\cdot \mid \bar{s}_1, a_1), \mathbb{E}_{a_2 \sim \bar{\pi}(\cdot | \bar{s}_2)} \overline{P}(\cdot \mid \bar{s}_2, a_2) \right) \leq K_{\overline{P}}^{\bar{\pi}} \cdot \bar{d}(\bar{s}_1, \bar{s}_2).$$

Intuitively, the Lipschitzness of the latent reward and transition functions guarantees that the latent space is well-structured, so that nearby latent states exhibit similar latent dynamics. Gelada et al. (2019) control those bounds by adding a *gradient penalty term* to the loss and enforce Lipschitzness (Gulrajani et al., 2017). One can also obtain constrained Lipchitz constants as a side effect by enforcing the metric $\bar{d}$ to match the *bisimulation distance* in the latent space (Zhang et al., 2021). Interestingly, when the latent space is discrete, Lipschitz constants can be trivially inferred since $K_{\overline{R}}^{\bar{\pi}} = 2R_{\text{MAX}}$ and $K_{\overline{P}}^{\bar{\pi}} = 1$ (Delgrange et al., 2022). Note also that as the spaces are assumed compact, restricting to continuous functions ensures Lipschitz continuity.

For the sake of presentation, we restrict our attention to the following assumption for Thms. 2 and 3:

**Assumption 1.** *We assume that the agent operates in the episodic RL setting, i.e., we consider the standard RL framework where the environment is eventually reset with probability one.*

Our results extend to general settings where a stationary distribution is accessible (c.f. Remark 3).

**World model quality.** Before introducing our safe policy improvement theorem, we first show that the local losses effectively measure the world model's quality with respect to the original environment. Namely, their difference in return obtained **under any latent policy in a well-defined neighborhood** is bounded by the local losses **derived from the reference, behavioral policy's state-action distribution**. This is formalized in the following theorem.

**Theorem 2.** *Suppose $\gamma > \frac{1}{2}$ and $K_{\overline{P}}^{\bar{\pi}} < \frac{1}{\gamma}$. Let $C \in (1, \frac{1}{\gamma})$, $\pi_b \in \Pi$ be the base policy, $(\bar{\pi} \circ \phi) \in \mathcal{N}^C(\pi_b)$ where $\bar{\pi} \in \overline{\Pi}$ is a latent policy and $\phi \colon \mathcal{S} \to \overline{\mathcal{S}}$ a state representation. Then,*

$$\left| \rho(\bar{\pi} \circ \phi, \mathcal{M}) - \rho(\bar{\pi}, \overline{\mathcal{M}}) \right| \leq \text{AEL}(\pi_b) \cdot \frac{L_R^{\xi_{\pi_b}}/\gamma + K_V \cdot L_P^{\xi_{\pi_b}}}{1/D_{IR}^{\sup}(\pi_b, \pi) - \gamma},$$

*where $\text{AEL}(\pi_b)$ denotes the* average episode length *when $\mathcal{M}$ runs under $\pi_b$, $K_V = \frac{K_{\overline{R}}^{\bar{\pi}}}{(1 - \gamma K_{\overline{P}}^{\bar{\pi}})}$, and $L_R^{\xi_{\pi_b}}, L_P^{\xi_{\pi_b}}$ are the local losses of Eq. 4 over the stationary distribution $\xi_{\pi_b}$ induced by $\pi_b$.*

In simpler terms, if the deviation (*supremum* IR, or SIR for short) between the behavioral policy and *any **new** policy* $\bar{\pi}$ stays strictly lower than $1/\gamma$, the gap in return between the environment and the world model for this new policy can be bounded using data collected via $\pi_b$. Minimizing local losses from $\pi_b$'s data ensures that refining the representation $\phi$ for $\bar{\pi}$ improves model quality: when these losses vanish, $\mathcal{M}$ and $\overline{\mathcal{M}}$ are almost surely equivalent under $\bar{\pi}$. The bound depends on the Average Episode Length (AEL), but even a loose upper bound is sufficient to preserve guarantees. It is also strongly influenced by the discount factor $\gamma$, which defines an implicit horizon. Smaller values permit larger deviations from $\pi_b$ and relax the accuracy required of the world model.

**Safe policy improvement.** We consider the setting where the world model is used to improve the behavioral policy $\pi_b = \bar{\pi}_b \circ \phi$, with $\bar{\pi}_b \in \overline{\Pi}$ and the representation $\phi$ is fixed during each update. Restricting updates to a well-defined neighborhood guarantees that $\rho(\bar{\pi} \circ \phi, \mathcal{M}) - \rho(\pi_b, \mathcal{M}) \geq \rho(\bar{\pi}, \overline{\mathcal{M}}) - \rho(\bar{\pi}_b, \overline{\mathcal{M}}) - \zeta$, where $\zeta$ is defined as the cumulative *modeling error* from the local losses.

**Theorem 3.** (Deep, Safe Policy Improvement) *Under the same preamble as in Thm. 2, assume that $\phi$ if fixed during the policy update and the behavioral is a latent policy with $\pi_b := \bar{\pi}_b \circ \phi$ and $\bar{\pi}_b \in \overline{\Pi}$. Then, the improvement of the return of $\mathcal{M}$ under $\bar{\pi}$ can be guaranteed on $\pi_b$ as*

$$\rho(\bar{\pi} \circ \phi, \mathcal{M}) - \rho(\pi_b, \mathcal{M}) \geq \rho(\bar{\pi}, \overline{\mathcal{M}}) - \rho(\bar{\pi}_b, \overline{\mathcal{M}}) - \zeta,$$

$$where \; \zeta := \mathrm{AEL}(\pi_b) \cdot \left( L_R^{\xi_{\pi_b}}/\gamma + K_V L_P^{\xi_{\pi_b}} \right) \left( \frac{1}{1/D_{IR}^{\sup}(\pi_b, \bar{\pi}) - \gamma} + \frac{1}{1 - \gamma} \right).$$

Theorem 3 addresses the OOT issue (Section 3.1): if the SIR of the behavioral remains strictly below $1/\gamma$, then minimizing the local losses reduces the error $\zeta$, ensuring safe policy improvement when the world model is used to enhance the policy. While our focus is not on offline SPI, Appendix E (Thm. 5) additionally provides a PAC variant of the result, following the standard use of confidence bounds in the SPI literature.

**Representation learning.** Finally, we analyze how learning a world model using our loss functions as an auxiliary task facilitates the learning of a useful representation. A good representation should ensure that environment states that are close in the representation also have close values, directly supporting policy learning. Specifically, we seek "*almost*" Lipschitz continuity (Vanderbei, 1991) of the form $\exists K : \forall s_1, s_2 \in \mathcal{S}, |V^{\pi_b}(s_1) - V^{\pi_b}(s_2)| \leq K \cdot \bar{d}(\phi_{old}(s_1), \phi_{old}(s_2)) + \mathcal{L}_{\pi_b}(\phi_{old})$ where $\mathcal{L}_{\pi_b}$ is an auxiliary loss **depending on the data collected by $\pi_b$**. Notably, a critical question is whether updating the policy and its representation, respectively to $\bar{\pi}$ and $\phi$, maintains Lipschitz continuity. Crucially, as the behavioral $\pi_b$ is updated to $\bar{\pi} \circ \phi$ with respect to the experience collected under $\pi_b$, the bound must hold for $\mathcal{L}_{\pi_b}$. The following theorem is a probabilistic version of this statement, formalized as a concentration inequality:

**Theorem 4.** (Deep SPI for representation learning) *Under the same preamble as in Thm. 2, let $\varepsilon > 0$ and $\delta := 4 \cdot \frac{L_R^{\xi_{\pi_b}} + \gamma K_V \cdot L_P^{\xi_{\pi_b}}}{\varepsilon \cdot \left( 1/D_{IR}^{\sup}(\pi_b, \bar{\pi}) - \gamma \right)}$. Then, with probability at least $1 - \delta$ under $\xi_{\pi_b}$, we have for all $s_1, s_2 \in \mathcal{S}$ that*

$$\left| V^{\bar{\pi}}(s_1) - V^{\bar{\pi}}(s_2) \right| \leq K_V \cdot \bar{d}(\phi(s_1), \phi(s_2)) + \varepsilon.$$

Theorem 4 addresses confounding policy updates (Section 3.2): minimizing the losses increases the probability that learned representations remain almost Lipschitz under controlled policy changes (with an SIR below $1/\gamma$). This prevents distinct states from collapsing into identical latent representations that degrade performance. We note that Gelada et al. (2019) proved a similar bound when $\pi_b = \bar{\pi}$ (the policy update was disregarded), which in contrast to ours, *surely* holds with

$$\varepsilon := \frac{L_R^{\xi_{\bar{\pi}}} + \gamma K_V \cdot L_P^{\xi_{\bar{\pi}}}}{1 - \gamma} \cdot \left( \frac{1}{\xi_{\bar{\pi}}(s_1)} + \frac{1}{\xi_{\bar{\pi}}(s_2)} \right).$$

However, in general spaces, for any specific $s \in \mathcal{S}$, $\xi_{\bar{\pi}}(s)$ might simply equal zero, making the bound undefined. In particular, in the continuous setting, $\mathcal{S}$ is widely assumed to be endowed with a Borel sigma-algebra, where the probability of every single point is indeed zero.

## 6 ACROSS THE SPI-VERSE: PPO COMES INTO PLAY

These theorems inspire a practical RL algorithm that combines policy improvement and guarantees with solid empirical performance. The critical part of our approach is to make sure updates are

restricted to the policy neighborhood while minimizing the auxiliary losses $L_R, L_P$. In fact, our neighborhood operator has close connections to PPO (Schulman et al., 2017), where the policy update is given by[3]

$$\pi_{n+1} \coloneqq \arg\sup_{\pi' \in \Pi} \mathbb{E}_{s \sim \xi_{\pi_n}} \left[ \mathbb{E}_{a \sim \pi'(\cdot|s)} A^{\pi_n}(s,a) - \mathfrak{D}_{\pi_n}(\pi' \mid s) \right], \qquad (5)$$

with $\mathfrak{D}_{\pi_n}(\pi' \mid s) = \mathbb{E}_{a \sim \pi_n(\cdot|s)} \mathrm{ReLu}\Big( \big[ \pi'(a|s)/\pi_n(a|s) - \mathrm{clip}(\pi'(a|s)/\pi_n(a|s),\, 1 \pm \epsilon) \big] \cdot A^{\pi_n}(s,a) \Big)$, for some $\epsilon > 0$. By fixing $\epsilon = C - 1$, instead of strictly constraining the updates to the neighborhood, the regularization $\mathfrak{D}_{\pi_n}(\pi' \mid s)$ corrects the utility $\mathbb{E}_{a \sim \pi'(\cdot|s)} A^{\pi_n}(s,a)$ (compare Eq. 3 and Eq. 5), so that there is no incentive for $\pi'$ to deviate from $\pi_n$ with an IR outside the range $[2 - C, C]$. Under the same assumption as in Theorem 1, PPO is also an instance of mirror learning (Kuba et al., 2022), meaning it also benefits from the same convergence guarantees.

Strictly restricting the IR in a neighborhood is much harder in practice, considering a PPO objective is thus an appealing alternative. However, it is not sufficient to add the auxiliary losses $L_P, L_R$ to the objective of Eq. 5 to maintain the guarantees. Indeed, updating the representation $\phi$ by minimizing the additional losses may push the the policy $\bar{\pi} \circ \phi$ outside the neighborhood. As a solution we propose to incorporate the local losses by replacing all occurrences of $A^{\pi_n}$ in Eq. 5 by the utility

$$U^{\pi_n}(s,a,s') \coloneqq A^{\pi_n}(s,a) - \alpha_R \cdot \ell_R(s,a) - \alpha_P \cdot \ell_P(s,a,s'), \qquad (6)$$

where $\ell_R(s,a) \coloneqq \big| R(s,a) - \bar{R}(\phi(s),a) \big|$, $\ell_P(s,a,s') \coloneqq \mathbb{E}_{\bar{s}' \sim \bar{P}(\cdot|\phi(s),a)} \bar{d}(\phi(s'), \bar{s}')$, $s' \sim P(\cdot \mid s,a)$, and $\alpha_R, \alpha_P \in (0,1]$. Intuitively, $\ell_R, \ell_P$ are transition-wise auxiliary losses that allow retrieving $L_R^{\xi_{\pi_n}}$ and $L_P^{\xi_{\pi_n}}$ in expectation w.r.t. the current policy $\pi_n$. When optimized, since they are clipped in a PPO-fashion, $U^{\pi_n}$ allows restricting the policy updates to the neighborhood.

---

**Algorithm 1: DeepSPI**

**Inputs:** Horizon $T$, batch size $B$, vectorized environment env, parameters $\theta$

Initialize vectors
$s \in \mathcal{S}^{(T+1) \times B}, a \in \mathcal{A}^{T \times B}, r \in \mathbb{R}^{T \times B}$

**repeat**
  **for** $t \leftarrow 1$ *to* $T$ **do**
    Draw actions from the current policy:
    $a_{t,i} \sim \bar{\pi}(\cdot \mid \phi(s_{t,i})) \quad \forall 1 \le i \le B$
    Perform a single parallelized ($B$) step:
    $r_t, s_{t+1} \leftarrow$ env.step$(s_t, a_t)$
  Update $\theta$ by descending
  $\nabla_\theta$ DeepSPI_loss$(s, a, r, U^{\bar{\pi} \circ \phi}, \theta)$
    ▷ *change $A$ in Eq. 5 by $U$ from Eq. 6*
  $s_1 \leftarrow s_{T+1}$
**until** *convergence*
**return** $\theta$

---

From this loss, we propose DeepSPI, **a principled algorithm leveraging the policy improvement and representation learning capabilities developed in our theory**. As our losses rely on distributions defined over the current policy, we focus on the on-policy setting. While model-based approaches are not standard in this setting, we stress that **highly parallelized collection of data** (e.g., via vectorized environments) **enables a wide coverage of the state space** (cf. Mayor et al., 2025; Gallici et al., 2025), which is suitable to optimize the latent model. DeepSPI updates the world model, the encoder, and the policy simultaneously while guaranteeing the representation is suited to perform safe policy updates.

## 6.1 ILLUSTRATIVE EXAMPLE

To illustrate the representation learning capabilities of DeepSPI, we consider the toy grid-world shown in Fig. 3. This environment mirrors the confounding policy update discussed in Sect. 3.2, instantiated earlier in Fig. 2.

The agent starts in the cell labeled I. Upon leaving the orange cell immediately to its right, it is sent to the top branch with probability $1 - \epsilon$ and to the bottom branch with probability $\epsilon$. It must then traverse a corridor of $n$ blue cells (here $n$=5). Moving one cell to the right yields a reward of +1, and the agent cannot move backwards.

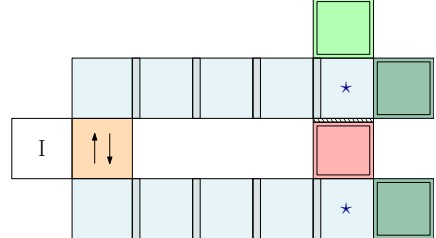

Figure 3: Toy maze environment illustrating the confounding policy update problem.

At the final corridor cell, marked with a $\star$, moving right yields a reward of +1 regardless of whether the agent is in the top or bottom branch, and the episode terminates. *The difference is when the agent*

---

[3]we give the formulation of Kuba et al. (2022), which is equal to the one of Schulman et al. (2017).

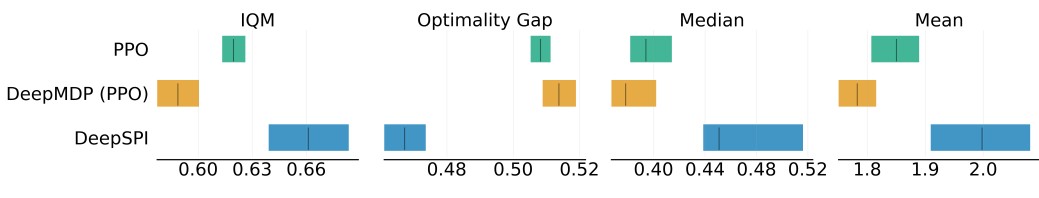

Figure 5: Aggregate results on **stochastic** versions of the standard 57 environments from ALE, with 95% confidence intervals (CIs). Higher values for the mean, median, and *interquartile mean* (IQM) indicate better performance, while a lower optimality gap is preferable (cf. Agarwal et al. 2021b). CIs are obtained through percentile bootstrapping with stratified resampling. Plots per environment available in Appendix H.3.

*moves up from the $\star$ cell*: in the top branch, it receives a reward of $+n/\gamma^n$, whereas in the bottom branch it receives $-(2-\epsilon)n/(\epsilon\gamma^n)$ before termination. As in Sect. 3.2, this construction ensures that if both $\star$ states are merged in the latent space, choosing "right" remains acceptable (their values coincide), but choosing "up" produces a negative expected return from the initial state I (details in Appendix G). To improve upon the policy that always chooses "right," the agent must learn to assign distinct representations to the two $\star$ cells, select "up" in the top one, and "right" in the bottom one.

We compare the behaviour of PPO and `DeepSPI` in this environment. Since our goal is to highlight the agent's representation-learning capabilities, each observation is provided as raw pixels. The agent must therefore learn both a policy and an encoder mapping pixels to a structured latent space. Further details on the environment and observation scheme are given in Appendix G. As shown in Fig. 4, the representation learned by PPO collapses the top and bottom $\star$ cells into a single latent state. With such a representation, the best policy PPO can learn is to always choose "right," which leads to a

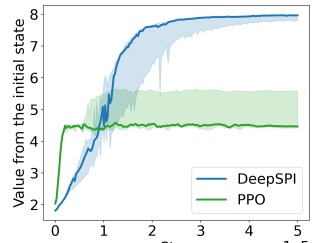
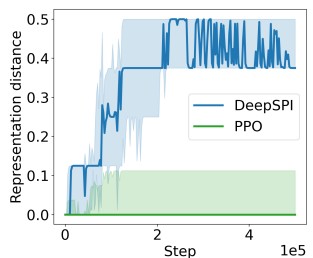

Figure 4: Value from cell I in the maze (left) and distance between the representation of the $\star$ cell from the top and bottom branches (right).

return of $\sim 4.8$. In contrast, `DeepSPI` benefits from the representation quality guarantees of Thm. 4, which ensure that states with different values remain separated in the latent space for all policies in a suitable neighborhood. This is exactly what we observe: the learned representation distinguishes the two $\star$ cells. As a result, the agent learns to choose "up" in the top $\star$ cell and "right" in the bottom one, achieving a return of $\sim 8$.

## 7 EXPERIMENTS

In this section, we evaluate `DeepSPI` on the Atari Arcade Learning Environment (ALE; Bellemare et al. 2013), where each frame is four stacked frames. While ALE domains feature a wide range of dynamics, to further highlight the robustness of our approach to probabilistic settings, we inject stochasticity by following Machado et al. (2018) with *sticky actions* (repeat the previous action with probability $p_a$) and *random initialisation* (games (re)start after $n_{\text{NOOP}}$ frames), using $p_a = 0.3$ and $n_{\text{NOOP}} = 60$.

As baselines, we use PPO (vectorized `cleanRL`; Huang et al., 2022) and `DeepMDPs` (Gelada et al., 2019). `DeepMDPs` are auxiliary transition and reward losses

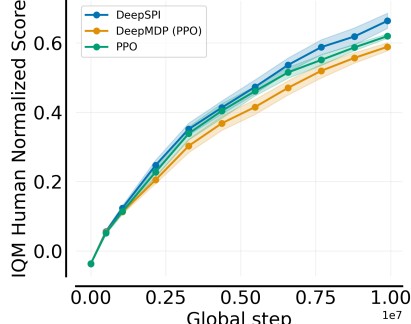

Figure 6: Sample efficiency w.r.t. IQM normalized scores on the stochastic ALE-57. Shaded regions give pointwise 95% CIs obtained via percentile stratified bootstrap.

(Sect. 5) that can be plugged into RL algorithms to improve representations, with guarantees. Unlike `DeepSPI`, these representation updates are unconstrained and can push policies out of the neighborhood, so *none of the SPI guarantees presented in this paper apply* to `DeepMDPs`. For a fair comparison, we plugged the `DeepMDP` losses to (vectorized) PPO, and we use the architecture as for

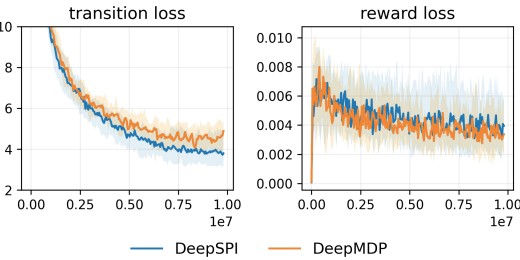

Figure 8: Sample environments from ALE where `DreamSPI` learns meaningful behaviors.

`DeepSPI`. We use default `cleanRL` hyperparameters for all three algorithms, except data collection (128 environments, horizon 8 steps).

As latent space, we use the raw 3D representation obtained after the convolution layers (as recommended and used by Gelada et al., 2019). For the transition function, we use a mixture of multivariate normal distributions (the transition network outputs 5 means/diagonal matrices). To enforce Lipschitz constraints on reward and transition functions, we model $\overline{R}, \overline{P}$ with Lipschitz networks, using norm-constrained GroupSort architectures to enforce 1-Lipschitzness Anil et al., 2019.

As shown in Fig. 5 and 6, `DeepSPI` delivers strong performance, improving on both PPO and `DeepMDP`. Notably, these results are obtained while preserving SPI-style properties; a valuable combination, as such theoretical control typically comes at the expense of performance and substantial data requirements. Beyond performance, we assess whether the `DeepSPI` world model exhibits accurate dynamics. Fig. 7 reports $L_P$, $L_R$ during training. Note that `DeepSPI` consistently achieves lower transition losses, indicating more accurate predictions. We discuss the statistical significance of that statement in Appendix H.3. In contrast to the off-policy setting of Gelada et al. (2019), we did not observe competing transition and reward losses in our parallel on-policy setting, as our losses are always computed under the current policy rather than replay-buffer data.

Figure 7: Median transition and reward losses during training, aggregated across all the ALE. For the sake of visualization, we cut $L_P$ lower values from the plot.

To probe the predictive quality of the latent model and illustrate Thm. 3, we introduced `DreamSPI`, a naïve variant where `DeepSPI` learns the world model and representation, and PPO updates the policy *from imagined trajectories* (Appendix F). Unlike off-policy approaches that exploit replay buffers, our on-policy setting updates the world model only from fresh interaction data, making model learning and planning harder to combine. Even so, `DreamSPI` learns in several environments and exhibits coherent behaviours (cf. Fig. 8 & Appendix H.3). Although its aggregate median score remains below the baselines, this is expected given stricter data requirements than standard model-based approaches. Importantly, the ability to maintain a model offers benefits that extend far beyond raw scores, enabling future applications in safety, verification, and reactive synthesis.

## 8 CONCLUSION AND FUTURE WORK

We developed a theoretical framework for safe policy improvement (SPI) that combines world-model and representation learning in nontrivial settings. Our results show that constraining policy updates within a well-defined neighborhood yields monotonic improvement and convergence, while auxiliary transition and reward losses ensure that the latent space remains suitable for policy optimisation. We further provided model-quality guarantees in the form of a "deep" SPI theorem, which jointly accounts for the learned representation and the reward/transition losses. These results directly address two critical issues in model-based RL: out-of-trajectory errors and confounding policy updates. Building on this analysis, we proposed `DeepSPI`, a principled algorithm that integrates the theoretical ingredients with PPO. On ALE, `DeepSPI` is competitive with and often improves upon PPO and `DeepMDPs`, while providing SPI guarantees.

This work opens several directions. A first avenue is to make pure deep SPI model-based planning practical. Our experiments with `DreamSPI` suggest that this is feasible but requires improved sample efficiency. Another direction goes beyond return optimization: a principled world model, grounded in our theory, can support safe reinforcement learning via formal methods, through synthesis (Delgrange et al., 2025; Lechner et al., 2022), or shielding (Jansen et al., 2020).

## REPRODUCIBILITY STATEMENT

All theoretical results are stated with explicit assumptions, and complete proofs are included in the appendix. The experimental setup is described in detail in the main text and supplementary material, including environments, hyperparameters, and training procedures. We provide the full source code as supplementary material to enable reproduction of our results. Datasets used in the experiments are publicly available (we use `envpool` Atari).

## ACKNOWLEDGMENTS

We thank Marnix Suilen and Guillermo A. Pérez for their valuable feedback during the preparation of this manuscript. This research was realized under F. Delgrange's VUB OZR mandate (VUB-OZR4417) and was supported by the "DESCARTES" iBOF project. W. Röpke and R. Avalos are supported by the Research Foundation – Flanders (FWO), with respective grant numbers 1197622N and 11F5721N.

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

# Appendix

## A  REMARK ON VALUE FUNCTIONS AND EPISODIC PROCESSES

An *episodic process* is formally defined as an MDP $\mathcal{M} = \langle \mathcal{S}, \mathcal{A}, P, R, s_I, \gamma \rangle$ where:

(i) there is a special state $s_{reset} \in \mathcal{S}$, intuitively indicating the termination of any *episode*;

(ii) the reset state does not incur any reward: $R(s_{reset}, a) = 0$ for all actions $a \in \mathcal{A}$;

(iii) $s_{reset}$ is almost surely visited under any policy: for all policies $\pi \in \Pi$, $\mathbb{P}_\pi \left( \left\{ (s_t, a_t)_{t \geq 0} \mid \exists i \colon s_i = s_{reset} \right\} \right) = 1$; and

(iv) $\mathcal{M}$ restarts from the initial state once reset: $P(\{s_I\} \mid s_{reset}, a) = 1$ for all $a \in \mathcal{A}$.

Note that by items (iii) and (iv), $s_{reset}$ is almost surely **infinitely often** visited: we have for all $\pi \in \Pi$ that

$$\mathbb{P}_\pi \left( \left\{ (s_t, a_t)_{t \geq 0} \mid \forall i \geq 0, \exists j > i \colon s_j = s_{reset} \right\} \right) = 1.$$

Alternatively and equivalently, an episodic process may also be defined without a unique reset state by the means of several *terminal states*, which go back to the initial state with probability one.

An *episode* of $\mathcal{M}$ is thus the prefix $s_0, a_0, \ldots, a_{t-1}, s_t$ of a trajectory where $s_t = s_{reset}$ and for all $i < t$, $s_i \neq s_{reset}$. Notice that our formulation embeds (but is not limited to) finite-horizon tasks, where an upper bound on the length of the episodes is fixed. The *average episode length* (AEL) of $\pi$ is then formally defined as $\text{AEL}(\pi) = \mathbb{E}_\pi [\mathbf{T}]$ with

$$\mathbf{T}(\tau) = \sum_{i=0}^{\infty} (i + 1) \cdot \mathbb{1} \left\{ s_i = s_{reset} \text{ and } \forall j < i, s_j \neq s_{reset} \right\}$$

for any trajectory $\tau = (s_t, a_t)_{t \geq 0}$.

Often, when considering episodic tasks, RL algorithms stops accumulating rewards upon the termination of every episode. In practical implementations, this corresponds to discarding rewards when a flag `done`, indicating episode termination, is set to `true`. In such case, we may slightly adapt our value functions as:

$$V^\pi(s) = \begin{cases} \mathbb{E}_\pi \left[ \sum_{t=0}^{\infty} \left( \prod_{i=1}^{t} \mathbb{1} \left\{ s_i \neq s_{reset} \right\} \cdot \gamma \right) R(s_t, a_t) \,\middle|\, s_0 = s \right] & \text{if } s \neq s_{reset} \\ 0 & \text{otherwise;} \end{cases}$$

or, when formalized as Bellman's equation:

$$Q^\pi(s, a) = \begin{cases} R(s, a) + \gamma \cdot \mathbb{E}_{s' \sim P(\cdot \mid s, a)} V^\pi(s') & \text{if } s \neq s_{reset} \\ 0 & \text{otherwise; and} \end{cases}$$

$$V^\pi(s) = \mathbb{E}_{a \sim \pi(\cdot \mid s)} Q^\pi(s, a).$$

All our results extend to this formulation (cf. Remark 2).

*Remark* 1 (Occupancy measure). In RL theory, the discounted occupancy measure

$$\mu_\pi^\gamma(s) := (1 - \gamma) \cdot \sum_{t=0}^{\infty} \gamma^t \mathbb{P}_\pi \left( \left\{ (s_i, a_i)_{i \geq 0} \mid s_t = s \right\} \right)$$

is often considered as the default marginal distribution over states the agent visit along the interaction, mostly because of its suitable theoretical properties. In fact, for any arbitrary MDP, $\mu_\pi^\gamma$ **is the stationary distribution** of the episodic process obtained by considering a reset probability of $1 - \gamma$ from every state of the original MDP (Puterman, 1994; Metelli et al., 2023). Again, we contend that all our results can be extended to the occupancy measure with little effort.

# B    POLICY IMPROVEMENTS THROUGH MIRROR LEARNING AND CONVERGENCE GUARANTEES

In this section, we prove that $\mathcal{N}^C$ (Eq. 2) is a proper *mirror learning neighborhood operator*. As a consequence, appropriately updating the policy according to $\mathcal{N}^C$ is guaranteed to be an instance of mirror learning, yielding the convergence guarantees of Theorem 1.

For completeness, we recall the definition of neighborhood operator from Kuba et al. (2022).

**Definition 1** (Neighborhood operator). *The mapping $\mathcal{N}\colon \Pi \to 2^\Pi$ is a (mirror learning) neighborhood operator, if*

1. (continuity) *It is a continuous map;*
2. (compactness) *Every $\mathcal{N}(\pi)$ is a compact set; and*
3. (closed ball) *There exists a metric $d\colon \Pi \times \Pi \to [0,\infty)$, such that for all policies $\pi \in \Pi$, there exists $\epsilon > 0$, such that $d(\pi, \pi') \leq \epsilon$ implies $\pi' \in \mathcal{N}(\pi)$.*

*The trivial neighborhood operator is $\mathcal{N}(\pi) = \Pi$.*

**Lemma 1.** $\mathcal{N}^C$ *is a neighborhood operator.*

*Proof.* Henceforth, fix a policy $\pi \in \Pi$. *When taking the supremum, infimum, maximum, or minimum value over states and actions, we always consider actions to be taken from the support of the behavioral policy (in the denominator of the quotient).*

Item 2 (compactness) is trivial due to $D_{\text{IR}}^{\inf}(\pi, \pi') \geq 2 - C$ and $D_{\text{IR}}^{\sup}(\pi, \pi') \leq C$ for any $\pi' \in \mathcal{N}^C(\pi)$. This means $\mathcal{N}^C(\pi)$ contains its extrema, i.e., all the policies $\pi'$ satisfying $D_{\text{IR}}^{\inf}(\pi, \pi') = 2 - C$ and $D_{\text{IR}}^{\sup}(\pi, \pi') \leq C$, or $D_{\text{IR}}^{\inf}(\pi, \pi') \geq 2 - C$ and $D_{\text{IR}}^{\sup}(\pi, \pi') = C$.

In the following, for any $\pi \in \Pi$ and sequence $(\pi_n)_{n \geq 0}$, we write $\pi_n \to \pi$ for the convergence of the sequence to $\pi$ with respect to the metric

$$d(\pi_1, \pi_2) = \begin{cases} \|\pi_1 - \pi_2\|_\infty & \text{if } \operatorname{supp}(\pi_1(\cdot \mid s)) = \operatorname{supp}(\pi_2(\cdot \mid s)) \quad \forall s \in \mathcal{S}, \text{ and} \\ 1 & \text{otherwise.} \end{cases} \qquad (7)$$

In other words, $\pi_n \to \pi$ means that $\pi_n$ converges to $\pi$ in supremum norm as $n \to \infty$ when the support of the converging policy stabilizes and becomes the same as the limit policy.

Let us prove item 1 (continuity). We show that $\mathcal{N}^C$ is a continuous *correspondence* by showing it is upper and lower *hemicontinuous* (Ok, 2007).

$\mathcal{N}^C$ is *upper hemicontinuous* (uhc) if it is *compact-valued* (item 1) and, for all policies $\pi \in \Pi$ and every sequences $(\pi_n)_{n \geq 0}$ and $(\pi'_n)_{n \geq 0}$ with $\pi'_n \in \mathcal{N}^C(\pi_n)$ for all $n \geq 0$, $\pi_n \to \pi$ and $\pi'_n \to \pi'$ implies $\pi' \in \mathcal{N}^C(\pi)$. Let $(\pi_n)_{n \geq 0}$ and $(\pi'_n)_{n \geq 0}$ be sequences of policies with $\pi'_n \in \mathcal{N}^C(\pi_n)$ for all $n \geq 0$.

Fix $s \in \mathcal{S}$ and $a \in \mathcal{A}$. Consider the mapping

$$f_{s,a}\colon \{(\pi, \pi') \in \Pi \times \Pi \mid a \in \operatorname{supp}(\pi(\cdot \mid s))\} \to [0,\infty), \quad (\pi, \pi') \mapsto \frac{\pi'(a \mid s)}{\pi(a \mid s)}.$$

It is clear $f_{s,a}$ is continuous since the application of $\pi$ to $\pi(a \mid s)$ is continuous and the division of two continuous functions is also continuous (when considering actions from the support of $\pi(\cdot \mid s)$). Importantly, for $\text{ext} \in \{\sup, \inf\}$, $D_{\text{IR}}^{\text{ext}}(\pi, \pi') = \text{ext}\{f_{s,a}(\pi, \pi')\colon s \in \mathcal{S}, a \in \operatorname{supp}(\pi(\cdot \mid s))\}$ is also continuous: since $\mathcal{S}$ and $\mathcal{A}$ are finite, the supremum (resp. infimum) boils down to taking the maximum (resp. minimum) of finitely many many continuous functions, which is a continuous operation.

Now, assume that $\pi_n \to \pi$ and $\pi'_n \to \pi'$. The continuity of $D_{\text{IR}}^{\text{ext}}$ means that $D_{\text{IR}}^{\text{ext}}(\pi_n, \pi'_n) \to D_{\text{IR}}^{\text{ext}}(\pi, \pi')$. Since $\pi'_n \in \mathcal{N}^C(\pi_n)$, we have $D_{\text{IR}}^{\inf}(\pi_n, \pi'_n) \geq 2 - C$ and $D_{\text{IR}}^{\sup}(\pi_n, \pi'_n) \leq C$ for all $n \geq 0$. By the fact that $D_{\text{IR}}^{\text{ext}}(\pi_n, \pi'_n)$ converges to $D_{\text{IR}}^{\text{ext}}(\pi, \pi')$ for $\text{ext} \in \{\inf, \sup\}$, we also have that $D_{\text{IR}}^{\inf}(\pi, \pi') \geq 2 - C$ and $D_{\text{IR}}^{\sup}(\pi, \pi') \leq C$.

Then, $\mathcal{N}^C$ is uhc.

$\mathcal{N}^C$ is *lower hemicontinuous* (lhc) if, for every policy $\pi$, sequence $(\pi_n)_{n \geq 0}$ with $\pi_n \to \pi$, and policy $\pi' \in \mathcal{N}^C(\pi)$, there exists a sequence $(\pi'_n)_{n \geq 0}$ with $\pi'_n \to \pi'$ and such that there is a $n_0 \geq 0$ from which, for all $n \geq n_0$, $\pi'_n \in \mathcal{N}^C(\pi_n)$.

Therefore, let $(\pi_n)_{n\geq0}$ be a sequence of policies so that $\pi_n \to \pi$ and $\pi' \in \mathcal{N}(\pi)$. Since $\pi_n \to \pi$, we have

$$\forall \delta > 0, \exists n_0 \in \mathbb{N}\colon \forall n \geq n_0, \|\pi_n - \pi\|_\infty \leq \delta \ \text{ and } \ \mathrm{supp}(\pi_n(\cdot \mid s)) = \mathrm{supp}(\pi(\cdot \mid s)) \quad \forall s \in \mathcal{S}.$$

In particular, this holds for $\delta < {}^{\pi_{\min}}/_2$, where $\pi_{\min} = \min\{\pi(a \mid s)\colon s \in \mathcal{S}, a \in \mathrm{supp}(\pi(\cdot \mid s))\}$. Let $n_0 \geq 0$ be the step associated with $\delta < {}^{\pi_{\min}}/_2$ and $n \geq n_0$. Write $\delta_n = \|\pi_n - \pi\|_\infty$ and let

$$\epsilon_n = \frac{2C\delta_n}{\pi_{\min}(C-1) + 2C\delta_n} \in (0,1)$$

Construct a sequence $(\pi'_n)_{n\geq0}$ so that, for all $s \in \mathcal{S}$, $a \in \mathcal{A}$, and $n \geq n_0$,

$$\pi'_n(a \mid s) = (1 - \epsilon_n) \cdot \pi'(a \mid s) + \epsilon_n \cdot \pi_n(a \mid s).$$

Intuitively, $\pi'_n$ is a mixture of distributions $\pi'(\cdot \mid s)$ and $\pi_n(\cdot \mid s)$. Consequently, $\pi'_n(\cdot \mid s)$ is a well-defined distribution. Finally, note that $\pi'_n \to \pi'$ because $\delta_n \to 0$, and so does $\epsilon_n$.

Now, we restrict our attention to $a \in \mathrm{supp}(\pi_n(\cdot \mid s))$. Note that since $\pi_n$ stably converges to $\pi$ with its support, $\pi$ has the same support as $\pi_n$. Furthermore, since $\pi' \in \mathcal{N}^C(C)$, $\pi'$ has also the same support as $\pi_n$. In consequence, $\pi'_n$ has the same support as $\pi_n$.

Having that said, we start by showing the upper bound:

$$\begin{aligned}
\frac{\pi'_n(a \mid s)}{\pi_n(a \mid s)} &= (1 - \epsilon_n)\frac{\pi'(a \mid s)}{\pi_n(a \mid s)} + \epsilon_n \\
&\leq (1 - \epsilon_n)\frac{C \cdot \pi(a \mid s)}{\pi_n(a \mid s)} + \epsilon_n && (\text{because } \pi'(a \mid s) \leq C \cdot \pi(a \mid s)) \\
&\leq (1 - \epsilon_n) \cdot \frac{C \cdot \pi(a \mid s)}{\pi(a \mid s) - \delta_n} + \epsilon_n && (\text{because } \pi_n(a \mid s) \geq \pi(a \mid s) - \delta_n) \\
&= (1 - \epsilon_n)\frac{C}{1 - {}^{\delta_n}/_{\pi(a|s)}} + \epsilon_n \\
&\leq (1 - \epsilon_n)\frac{C}{1 - {}^{\delta_n}/_{\pi_{\min}}} + \epsilon_n.
\end{aligned}$$

Note that for all $x \in [0, {}^1/_2]$,

$$\frac{1}{1-x} \leq 1 + 2x \ \text{ because } \ 1 + 2x - \frac{1}{1-x} \geq 0 \iff \frac{(1+2x)(1-x)-1}{1-x} \geq 0 \iff \frac{x(1-2x)}{1-x} \geq 0.$$

Then, since $0 < {}^{\delta_n}/_{\pi_{\min}} < {}^1/_2$, we have

$$\frac{\pi'_n(a \mid s)}{\pi_n(\pi \mid s)} \leq (1 - \epsilon_n) \cdot C \cdot (1 + {}^{2\delta_n}/_{\pi_{\min}}) + \epsilon_n.$$

Let $x_n = 1 + \frac{2\delta_n}{\pi_{\min}}$, and note that

$$\epsilon_n = \frac{2C\delta_n}{\pi_{\min}(C-1) + 2C\delta_n} = \frac{2C \cdot {}^{\delta_n}/_{\pi_{\min}}}{C + 2C \cdot {}^{\delta_n}/_{\pi_{\min}} - 1} = \frac{-2C \cdot {}^{\delta_n}/_{\pi_{\min}}}{1 - C - 2C \cdot {}^{\delta_n}/_{\pi_{\min}}} = \frac{C(1-x_n)}{1 - x_n \cdot C}.$$

Then,

$$\begin{aligned}
\frac{\pi'_n(a \mid s)}{\pi_n(a \mid s)} &\leq (1 - \epsilon_n)x_n \cdot C + \epsilon_n \\
&= x_n \cdot C - \epsilon_n \cdot x_n \cdot C + \epsilon_n \\
&= x_n \cdot C - \frac{C(1-x_n)}{1 - x_n \cdot C} \cdot x_n \cdot C + \frac{C(1-x_n)}{1 - x_n \cdot C} \\
&= \frac{x_n \cdot C(1 - x_n \cdot C) - x_n \cdot C^2(1-x_n) + C(1-x_n)}{1 - x_n \cdot C} \\
&= \frac{x_n \cdot C - x_n^2 C^2 - x_n \cdot C^2 + x_n^2 C^2 + C - x_n \cdot C}{1 - x_n \cdot C}
\end{aligned}$$

$$= \frac{-x_n \cdot C^2 + C}{1 - x_n \cdot C}$$

$$= C \cdot \frac{1 - x_n \cdot C}{1 - x_n \cdot C}$$

$$= C,$$

which means that $D_{\text{IR}}^{\text{sup}}(\pi_n, \pi'_n) \leq C$.

We now show the lower bound:

$$\frac{\pi'_n(a \mid s)}{\pi_n(a \mid s)} = (1 - \epsilon_n)\frac{\pi'(a \mid s)}{\pi_n(a \mid s)} + \epsilon_n$$

$$\geq (1 - \epsilon_n)\frac{(2 - C) \cdot \pi(a \mid s)}{\pi_n(a \mid s)} + \epsilon_n \qquad \text{(because } \pi'(a \mid s) \geq (2 - C) \cdot \pi(a \mid s)\text{)}$$

$$\geq (1 - \epsilon_n)\frac{(2 - C) \cdot \pi(a \mid s)}{\pi(a \mid s) + \delta_n} + \epsilon_n \qquad \text{(because } \pi_n(a \mid s) \leq \pi(a \mid s) + \delta_n\text{)}$$

$$= (1 - \epsilon_n)\frac{(2 - C)}{1 + \delta_n/\pi(a|s)} + \epsilon_n$$

$$\geq (1 - \epsilon_n)\frac{(2 - C)}{1 + \delta_n/\pi_{\min}} + \epsilon_n$$

$$\geq (1 - \epsilon_n) \cdot (2 - C) \cdot (1 - \delta_n/\pi_{\min}) + \epsilon_n \qquad \text{(because for all } x \in \mathbb{R}, \frac{1}{1+x} \geq 1 - x\text{)}$$

$$= (1 - \epsilon_n) \cdot (2 - C - 2 \cdot \delta_n/\pi_{\min} + C \cdot \delta_n/\pi_{\min}) + \epsilon_n$$

$$= (1 - \epsilon_n) \cdot (2 - C + (C - 2) \cdot \delta_n/\pi_{\min}) + \epsilon_n$$

$$= 2 - C + (C - 2) \cdot \delta_n/\pi_{\min} - \epsilon_n(2 - C + (C - 2) \cdot \delta_n/\pi_{\min}) + \epsilon_n$$

$$= 2 - C + (C - 2) \cdot \delta_n/\pi_{\min} + \epsilon_n(C - 1 + (2 - C) \cdot \delta_n/\pi_{\min})$$

$$= 2 - C + (C - 2) \cdot \delta_n/\pi_{\min} + \epsilon_n (C - 1) + \epsilon_n \cdot (2 - C) \cdot \delta_n/\pi_{\min}$$

$$= 2 - C + (C - 2) \cdot \delta_n/\pi_{\min} + \frac{2C\delta_n \cdot (C - 1)}{\pi_{\min}(C - 1) + 2C\delta_n} + \frac{2C\delta_n \cdot (2 - C)}{\pi_{\min}(C - 1) + 2C\delta_n} \cdot \delta_n/\pi_{\min}$$

$$= 2 - C + \delta_n \cdot \left( \frac{C - 2}{\pi_{\min}} + \frac{2C \cdot (C - 1)}{\pi_{\min}(C - 1) + 2C\delta_n} + \frac{2C\delta_n \cdot \pi_{\min}^{-1} \cdot (2 - C)}{\pi_{\min}(C - 1) + 2C\delta_n} \right)$$

$$\geq 2 - C.$$

To see how we obtain the last line, note that it suffices to show the content of the parenthesis multiplied by $\delta_n$ is greater than zero, i.e.,

$$\frac{C - 2}{\pi_{\min}} + \frac{2C \cdot (C - 1)}{\pi_{\min}(C - 1) + 2C\delta_n} + \frac{2C\delta_n \cdot \pi_{\min}^{-1} \cdot (2 - C)}{\pi_{\min}(C - 1) + 2C\delta_n} \geq 0$$

$$\Longleftrightarrow \qquad \frac{2C \cdot (C - 1) + 2C\delta_n \cdot \pi_{\min}^{-1} \cdot (2 - C)}{\pi_{\min}(C - 1) + 2C\delta_n} \geq \frac{2 - C}{\pi_{\min}}$$

$$\Longleftrightarrow \qquad 2C\pi_{\min} \cdot (C - 1) + 2C\delta_n \cdot (2 - C) \geq (2 - C) \cdot (\pi_{\min}(C - 1) + 2C\delta_n)$$

$$\Longleftrightarrow \qquad 2C\pi_{\min} \cdot (C - 1) \geq (2 - C) \cdot (\pi_{\min}(C - 1) + 2C\delta_n - 2C\delta_n)$$

$$\Longleftrightarrow \qquad 2C\pi_{\min} \cdot (C - 1) \geq (2 - C) \cdot (\pi_{\min}(C - 1))$$

$$\Longleftrightarrow \qquad 2C \geq 2 - C,$$

which is always satisfied because $C \geq 1$. Therefore, since this holds for any $s \in \mathcal{S}$ and both $\pi'_n$ and $\pi_n$ have the same support, we have that $D_{\text{IR}}^{\inf}(\pi, \pi') \geq 2 - C$.

Thus, we have $D_{\text{IR}}^{\inf}(\pi, \pi') \geq 2 - C$ and $D_{\text{IR}}^{\text{sup}}(\pi, \pi') \leq C$, $\pi'_n \in \mathcal{N}^C(\pi_n)$. Therefore, $\mathcal{N}^C$ is lhc.

Since $\mathcal{N}^C$ is uhc and lhc, it is continuous. This concludes the proof of item 1.

It remains to show item 3. Let $\epsilon = (C - 1) \cdot \min_{s,a} \pi(a \mid s)$, with $a$ taken from $\text{supp}(\pi(\cdot \mid s))$. Assume $d(\pi, \pi') \leq \epsilon$ (cf. Eq. 7). For all $s \in \mathcal{S}, a \in \text{supp}(\pi(\cdot \mid s))$, we have

$$\pi'(a \mid s)$$

$$\leq \pi(a \mid s) + \epsilon$$
$$\leq \pi(a \mid s) + (C - 1) \cdot \min_{s,a} \pi(a \mid s)$$
$$\leq \pi(a \mid s) + (C - 1) \cdot \pi(a \mid s)$$
$$= \pi(a \mid s) \cdot (1 + C - 1)$$
$$= \pi(a \mid s) \cdot C,$$

or equivalently:

$$\frac{\pi'(a \mid s)}{\pi(a \mid s)} \leq C.$$

It remains to show the lower bound:

$$\pi'(a \mid s) \geq \pi(a \mid s) - \epsilon$$
$$= \pi(a \mid s) - (C - 1) \cdot \min_{s,a} \pi(a \mid s)$$
$$\geq \pi(a \mid s) - (C - 1) \cdot \pi(a \mid s)$$
$$= \pi(a \mid s) \cdot (1 - C + 1)$$
$$= \pi(a \mid s) \cdot C$$
$$\geq \pi(a \mid s)(2 - C),$$

or equivalently:

$$\frac{\pi'(a \mid s)}{\pi(a \mid s)} \geq 2 - C.$$

This concludes the proof of item 3. □

Then, Theorem 1 is obtained as a corollary of Lemma 1, and the fact that the update process

$$\pi_{n+1} := \arg\sup_{\pi' \in \mathcal{N}^C(\pi_n)} \mathbb{E}_{s \sim \xi_{\pi_n}} \mathbb{E}_{a \sim \pi'(\cdot|s)} [A^{\pi_n}(s, a)],$$

is an instance of mirror learning (Kuba et al., 2022).

## C  CRUDE WASSERSTEIN UPPER BOUND

**Lemma 2.** *Let $s \in \mathcal{S}$ and $a \in \mathcal{A}$, the following upper bound holds:*

$$\mathcal{W}\big(\phi_\sharp P(\cdot \mid s, a),\ \overline{P}(\cdot \mid \phi(s), a)\big) \leq \mathbb{E}_{s' \sim P(\cdot|s,a)} \mathbb{E}_{\bar{s}' \sim \overline{P}(\cdot|\phi(s),a)} \bar{d}\big(\phi(s'), \bar{s}'\big).$$

*Proof.*

$$\mathcal{W}\big(\phi_\sharp P(\cdot \mid s, a),\ \overline{P}(\cdot \mid \phi(s), a)\big)$$

$$= \sup_{\|f\|_{\mathrm{Lip}} \leq 1} \left[ \mathbb{E}_{s' \sim P(\cdot|s,a)} f\big(\phi(s')\big) - \mathbb{E}_{\bar{s}' \sim \overline{P}(\cdot|\phi(s),a)} f(\bar{s}') \right] \tag{1}$$

$$\leq \mathbb{E}_{s' \sim P(\cdot|s,a)} \left[ \sup_{\|f\|_{\mathrm{Lip}} \leq 1} f\big(\phi(s')\big) - \mathbb{E}_{\bar{s}' \sim \overline{P}(\cdot|\phi(s),a)} f(\bar{s}') \right]$$

$$= \mathbb{E}_{s' \sim P(\cdot|s,a)} \mathcal{W}\big(\delta_{\phi(s')},\ \overline{P}(\cdot \mid \phi(s), a)\big)$$

$$= \mathbb{E}_{s' \sim P(\cdot|s,a)} \left[ \min_{\lambda \in \Lambda(\delta_{\phi(s')}, \overline{P}(\cdot|\phi(s),a))} \mathbb{E}_{(\bar{s}_1, \bar{s}_2) \sim \lambda} d(\bar{s}_1, \bar{s}_2) \right] \tag{2}$$

$$= \mathbb{E}_{s' \sim P(\cdot|s,a)} \mathbb{E}_{\bar{s}' \sim \overline{P}(\cdot|\phi(s),a)} \bar{d}\big(\phi(s'), \bar{s}'\big).$$

Here, (1) corresponds to the dual Kantorovich–Rubinstein formulation (Kantorovich and Rubinstein, 1958) where $\|\cdot\|_{\mathrm{Lip}}$ corresponds to the Lipschitz norm, while (2) follows from the primal Monge formulation (Monge, 1781), with a trivial coupling induced by $\delta_{\phi(s')}$, the Dirac measure with impulse $\phi(s')$. □

## D    REMARK ON SAFE POLICY IMPROVEMENT METHODS

Standard principled *safe policy improvement* methods (SPI; Thomas et al., 2015; Ghavamzadeh et al., 2016a; Laroche et al., 2019; Simão et al., 2020; Castellini et al., 2023; Wienhöft et al., 2023) do not consider representation learning. Instead, SPI methods assume $\overline{\mathcal{S}} := \mathcal{S}$ and learn $\overline{R}, \overline{P}$ by maximum likelihood estimation with respect to the experience stored in $\mathcal{B}$ collected by the behavioral $\pi_b$. Then, the policy improvement relies on finding the best policy in $\overline{\mathcal{M}}$ that is (probably approximately correctly) guaranteed to improves on the behavioral policy (up to an error term $\zeta > 0$) against a set of all admissible MDPs, called *robust MDPs* (Iyengar, 2005; Nilim and Ghaoui, 2005; Wiesemann et al., 2013; Ghavamzadeh et al., 2016b; Suilen et al., 2024):

$$\arg\sup_{\overline{\pi} \in \overline{\Pi}} \rho\big(\overline{\pi}, \overline{\mathcal{M}}\big) \qquad \text{such that} \qquad \arg\inf_{\mathcal{M}' \in \Xi(\overline{\mathcal{M}}, e)} \rho(\pi, \mathcal{M}') \geq \rho(\pi_b, \mathcal{M}') - \zeta, \text{ where}$$

$$\Xi(\overline{\mathcal{M}}, e) := \left\{ \mathcal{M} = \langle \mathcal{S}, \mathcal{A}, P, R, s_I, \gamma \rangle \ \middle| \ \begin{array}{cc} \big|R(s,a) - \overline{R}(s,a)\big| \leq R_{\text{MAX}} \cdot e(s,a) & \text{and} \\ d_{TV}\big(P(\cdot \mid s,a), \overline{P}(\cdot \mid s,a)\big) \leq e(s,a) & \forall s \in \mathcal{S}, a \in \mathcal{A} \end{array} \right\},$$

$e(s,a)$ being an *error* term depending on the number of times each state $s$ and action $a$ are present in the dataset $\mathcal{B}$, and $d_{TV}$ being the *total variation distance* (Müller, 1997) which boils down to the $L_1$ distance when the state-action space is finite. To provide *probably approximately correct* (PAC) guarantees, the state-action pairs need to be visited a *sufficient amount of time*, depending on the size of the state-action space, to ensure $e$ is sufficiently small.

Note that the reward and total variation constraints are very related to our local losses $L_R$ and $L_P$: the representation corresponds here to the identity and $d_{TV}$ coincides with Wasserstein as the state space is discrete (Villani, 2009). The major difference here is that the bounds need to hold *globally*, i.e., for all state-action pairs, which make their computation typically intractable in complex settings (e.g., high-dimensional feature spaces).

We argue **this objective is ill-suited to complex settings**. First, classic SPI does not apply to general spaces. Second, assuming we deal with *finite*, high-dimensional feature spaces (e.g., visual inputs or the RAM of a video game), it is simply unlikely that $\mathcal{B}$ contains all state-action pairs. *SPI with baseline bootstrapping* (Laroche et al., 2019) allows bypassing this requirement by updating $\pi_b$ only in state-action pairs where a *sufficient* number of samples are present in $\mathcal{B}$. Nevertheless, this number is gigantic and is linear in the state-action space while being exponential in the size of the encoding of $\gamma$ and the desired error $\zeta$. This deems the policy update intractable. Finally, as mentioned, standard SPI does not consider representation learning. This is a further obstacle to its application in complex settings.

## E    SAFE POLICY IMPROVEMENTS: PROOFS

**Notations** Henceforth, we denote by $\overline{V}^{\overline{\pi}}$ the value function of the world model $\overline{\mathcal{M}}$ obtained under any latent policy $\overline{\pi} \in \overline{\Pi}$. When it is clear from the context that $\phi$ is the representation used jointly with a latent policy $\overline{\pi}$, we may simply write $V^{\overline{\pi}}$ instead of $V^{(\overline{\pi} \circ \phi)}$ for the value function of executing $\overline{\pi}$ in $\mathcal{M}$. In the following, we may also write $(s, a) \sim \xi_\pi$ as a shorthand for first drawing $s \sim \xi_\pi$ and then $a \sim \pi(\cdot \mid s)$ for any policy $\pi \in \Pi$.

We start by recalling a result from Gelada et al. (2019) that will be useful in the subsequent proofs.
**Lemma 3** (Lipschitzness of the *latent* value function)**.** *Let $\overline{\mathcal{M}}$ be a latent MDP and $\overline{\pi}$ be a policy for $\overline{\mathcal{M}}$. Assume that $\overline{\mathcal{M}}$ has reward and transition constants $K_{\overline{R}}^{\overline{\pi}}$ and $K_{\overline{P}}^{\overline{\pi}}$ with $K_{\overline{P}}^{\overline{\pi}} < 1/\gamma$. Then, the latent value function is $K_{\overline{R}}^{\overline{\pi}}/(1 - \gamma K_{\overline{P}}^{\overline{\pi}})$-Lipschitz, i.e., for all $\bar{s}_1, \bar{s}_2 \in \overline{\mathcal{S}}$,*

$$\big|\overline{V}^{\overline{\pi}}(\bar{s}_1) - \overline{V}^{\overline{\pi}}(\bar{s}_2)\big| \leq \frac{K_{\overline{R}}^{\overline{\pi}}}{1 - \gamma K_{\overline{P}}^{\overline{\pi}}} \cdot \bar{d}(\bar{s}_1, \bar{s}_2)$$

Note that the bound is straightforward when the latent space is discrete and the discrete metric $\mathbb{1}\{\neq\}$ is chosen for $\bar{d}$: the largest possible difference in values is $2R_{\text{MAX}}/1-\gamma$.

We also consider bounding expected value difference between the original MDP and the latent MDP by the local losses evaluated with respect to a behavioral policy $\pi_b$. Importantly, the expectation is measured over states and actions generated according to $\pi_b$, whereas the values correspond to those evaluated under *another latent policy* $\overline{\pi}$. The following Lemma states that the value difference yielded by a latent policy can be measured according to another behavioral policy, provided that the latent policy lies within a well-defined neighborhood of the behavioral policy.

**Lemma 4** (Average value difference bound). *Let $\pi_b \in \Pi$ be the behavioral policy, $(\bar{\pi} \circ \phi) \in \mathcal{N}^{1/\gamma}(\pi_b)$ so that $\bar{\pi} \in \overline{\Pi}$ and $\phi \colon \mathcal{S} \to \overline{\mathcal{S}}$ is a state representation. Assume $\overline{\mathcal{M}}$ is equipped by the Lipschitz constants $K_{\overline{R}}^{\bar{\pi}}$ and $K_{\overline{P}}^{\bar{\pi}}$ and let $K_V = K_{\overline{R}}^{\bar{\pi}}/(1-\gamma K_{\overline{P}}^{\bar{\pi}})$. Assume that $K_{\overline{P}}^{\bar{\pi}}$ is strictly lower than $1/\gamma$. Then, the average difference of value of $\mathcal{M}$ and $\overline{\mathcal{M}}$ under $\bar{\pi}$ is bounded by*

$$\mathbb{E}_{s\sim\xi_{\pi_b}} \left| V^{\bar{\pi}}(s) - \overline{V}^{\bar{\pi}}(\phi(s)) \right| \le \frac{L_R^{\xi_{\pi_b}} + \gamma K_V \cdot L_P^{\xi_{\pi_b}}}{1/D_{IR}^{\sup}(\pi_b,\bar{\pi}) - \gamma}.$$

*Proof.* The proof follows by adapting the proof of (Gelada et al., 2019, Lemma 3) by taking extra care of the behavioral policy. Namely, we want to evaluate the value difference bound for the latent policy $\bar{\pi}$, assuming states and actions are/have been produced by executing the behavioral policy $\pi_b$. The idea is to incorporate the divergence from $\pi_b$ to $\bar{\pi}$ in the bound, formalized as the supremum IR between the underlying distribution of the two policies.

$$\mathbb{E}_{s\sim\xi_{\pi_b}} \left| V^{\bar{\pi}}(s) - \overline{V}^{\bar{\pi}}(\phi(s)) \right|$$

$$= \mathbb{E}_{s\sim\xi_{\pi_b}} \left| \mathbb{E}_{a\sim\bar{\pi}(\cdot|\phi(s))} \left[ R(s,a) + \gamma \mathbb{E}_{s'\sim P(\cdot|s,a)} \left[ V^{\bar{\pi}}(s') \right] \right] - \mathbb{E}_{a\sim\bar{\pi}(\cdot|\phi(s))} \left[ \overline{R}(\phi(s),a) + \gamma \mathbb{E}_{\bar{s}'\sim\overline{P}(\cdot|\phi(s),a)} \left[ \overline{V}^{\bar{\pi}}(\bar{s}') \right] \right] \right|$$

$$= \mathbb{E}_{s\sim\xi_{\pi_b}} \left| \mathbb{E}_{a\sim\bar{\pi}(\cdot|\phi(s))} \left[ R(s,a) - \overline{R}(\phi(s),a) \right] + \gamma \mathbb{E}_{a\sim\bar{\pi}(\cdot|\phi(s))} \left[ \mathbb{E}_{\substack{s'\sim P(\cdot|s,a)\\ \bar{s}'\sim\overline{P}(\cdot|\phi(s),a)}} \left[ V^{\bar{\pi}}(s') - \overline{V}^{\bar{\pi}}(\bar{s}') \right] \right] \right|$$

$$= \mathbb{E}_{s\sim\xi_{\pi_b}} \left| \mathbb{E}_{a\sim\bar{\pi}(\cdot|\phi(s))} \left[ R(s,a) - \overline{R}(\phi(s),a) \right] + \gamma \mathbb{E}_{a\sim\bar{\pi}(\cdot|\phi(s))} \left[ \mathbb{E}_{\substack{s'\sim P(\cdot|s,a)\\ \bar{s}'\sim\overline{P}(\cdot|\phi(s),a)}} \left[ V^{\bar{\pi}}(s') - \overline{V}^{\bar{\pi}}(\phi(s')) + \overline{V}^{\bar{\pi}}(\phi(s')) - \overline{V}^{\bar{\pi}}(\bar{s}') \right] \right] \right|$$

$$= \mathbb{E}_{s\sim\xi_{\pi_b}} \left| \mathbb{E}_{a\sim\bar{\pi}(\cdot|\phi(s))} \left[ R(s,a) - \overline{R}(\phi(s),a) \right] \right.$$
$$\left. + \gamma \mathbb{E}_{a\sim\bar{\pi}(\cdot|\phi(s))} \left[ \mathbb{E}_{s'\sim P(\cdot|s,a)} \left[ V^{\bar{\pi}}(s') - \overline{V}^{\bar{\pi}}(\phi(s')) \right] + \mathbb{E}_{\substack{s'\sim P(\cdot|s,a)\\ \bar{s}'\sim\overline{P}(\cdot|\phi(s),a)}} \left[ \overline{V}^{\bar{\pi}}(\phi(s')) - \overline{V}^{\bar{\pi}}(\bar{s}') \right] \right] \right|$$

$$\le \mathbb{E}_{s\sim\xi_{\pi_b}} \mathbb{E}_{a\sim\bar{\pi}(\cdot|\phi(s))} \left| \left[ R(s,a) - \overline{R}(\phi(s),a) \right] + \gamma \mathbb{E}_{s'\sim P(\cdot|s,a)} \left[ V^{\bar{\pi}}(s') - \overline{V}^{\bar{\pi}}(\phi(s')) \right] + \gamma \mathbb{E}_{\substack{s'\sim P(\cdot|s,a)\\ \bar{s}'\sim\overline{P}(\cdot|\phi(s),a)}} \left[ \overline{V}^{\bar{\pi}}(\phi(s')) - \overline{V}^{\bar{\pi}}(\bar{s}') \right] \right|$$
$$\text{(Jensen's inequality)}$$

$$\le \mathbb{E}_{s\sim\xi_{\pi_b}} \mathbb{E}_{a\sim\bar{\pi}(\cdot|\phi(s))} \left| R(s,a) - \overline{R}(\phi(s),a) \right| + \gamma \mathbb{E}_{s\sim\xi_{\pi_b}} \mathbb{E}_{a\sim\bar{\pi}(\cdot|\phi(s))} \left| \mathbb{E}_{\substack{s'\sim P(\cdot|s,a)\\ \bar{s}'\sim\overline{P}(\cdot|\phi(s),a)}} \left[ \overline{V}^{\bar{\pi}}(\phi(s')) - \overline{V}^{\bar{\pi}}(\bar{s}') \right] \right|$$
$$+ \gamma \mathbb{E}_{s\sim\xi_{\pi_b}} \mathbb{E}_{a\sim\bar{\pi}(\cdot|\phi(s))} \left| \mathbb{E}_{s'\sim P(\cdot|s,a)} \left[ V^{\bar{\pi}}(s') - \overline{V}^{\bar{\pi}}(\phi(s')) \right] \right|$$
$$\text{(Triangle inequality)}$$

$$=$$
$$+ \gamma \mathbb{E}_{s\sim\xi_{\pi_b}} \mathbb{E}_{a\sim\pi_b(\cdot|s)} \left| \frac{\bar{\pi}(a\mid\phi(s))}{\pi_b(a\mid s)} \mathbb{E}_{\substack{s'\sim P(\cdot|s,a)\\ \bar{s}'\sim\overline{P}(\cdot|\phi(s),a)}} \left[ \overline{V}^{\bar{\pi}}(\phi(s')) - \overline{V}^{\bar{\pi}}(\bar{s}') \right] \right|$$
$$+ \gamma \mathbb{E}_{s\sim\xi_{\pi_b}} \mathbb{E}_{a\sim\pi_b(\cdot|s)} \left| \frac{\bar{\pi}(a\mid\phi(s))}{\pi_b(a\mid s)} \mathbb{E}_{s'\sim P(\cdot|s,a)} \left[ V^{\bar{\pi}}(s') - \overline{V}^{\bar{\pi}}(\phi(s')) \right] \right|$$
$$\text{(because } \operatorname{supp}(\bar{\pi}(\cdot\mid\phi(s))) = \operatorname{supp}(\pi_b(\cdot\mid s)) \text{ for all } s \in \mathcal{S})$$

$$\leq D_{\text{IR}}^{\sup}(\pi_{\text{b}}, \bar{\pi}) \underset{s,a \sim \xi_{\pi_{\text{b}}}}{\mathbb{E}} \left| R(s,a) - \bar{R}(\phi(s),a) \right| + \gamma \cdot D_{\text{IR}}^{\sup}(\pi_{\text{b}}, \bar{\pi}) \underset{s,a \sim \xi_{\pi_{\text{b}}}}{\mathbb{E}} \left| \underset{\bar{s}' \sim \phi_\sharp P(\cdot|s,a)}{\mathbb{E}} \bar{V}^{\bar{\pi}}(\bar{s}') - \underset{\bar{s}' \sim \bar{P}(\cdot|\phi(s),a)}{\mathbb{E}} \bar{V}^{\bar{\pi}}(\bar{s}') \right|$$

$$+ \gamma \cdot D_{\text{IR}}^{\sup}(\pi_{\text{b}}, \bar{\pi}) \underset{s,a \sim \xi_{\pi_{\text{b}}}}{\mathbb{E}} \left| \underset{s' \sim P(\cdot|s,a)}{\mathbb{E}} \left[ V^{\bar{\pi}}(s') - \bar{V}^{\bar{\pi}}(\phi(s')) \right] \right|$$

$$\left( \text{because } D_{\text{IR}}^{\sup}(\pi_{\text{b}}, \bar{\pi}) = \sup_{s,a} \left[ \frac{\bar{\pi}(a|\phi(s))}{\pi_{\text{b}}(\cdot|s)} \right] \right)$$

$$= D_{\text{IR}}^{\sup}(\pi_{\text{b}}, \bar{\pi}) \cdot L_R^{\xi_{\pi_{\text{b}}}} + \gamma \cdot D_{\text{IR}}^{\sup}(\pi_{\text{b}}, \bar{\pi}) \underset{s,a \sim \xi_{\pi_{\text{b}}}}{\mathbb{E}} \left| \underset{\bar{s}' \sim \phi_\sharp P(\cdot|s,a)}{\mathbb{E}} \bar{V}^{\bar{\pi}}(\bar{s}') - \underset{\bar{s}' \sim \bar{P}(\cdot|\phi(s),a)}{\mathbb{E}} \bar{V}^{\bar{\pi}}(\bar{s}') \right|$$

$$+ \gamma \cdot D_{\text{IR}}^{\sup}(\pi_{\text{b}}, \bar{\pi}) \underset{s,a \sim \xi_{\pi_{\text{b}}}}{\mathbb{E}} \left| \underset{s' \sim P(\cdot|s,a)}{\mathbb{E}} \left[ V^{\bar{\pi}}(s') - \bar{V}^{\bar{\pi}}(\phi(s')) \right] \right|$$

$$(\text{by definition of } L_R^{\xi_{\pi_{\text{b}}}})$$

$$\leq D_{\text{IR}}^{\sup}(\pi_{\text{b}}, \bar{\pi}) \cdot L_R^{\xi_{\pi_{\text{b}}}} + \gamma K_V \cdot D_{\text{IR}}^{\sup}(\pi_{\text{b}}, \bar{\pi}) \underset{s,a \sim \xi_{\pi_{\text{b}}}}{\mathbb{E}} \mathcal{W}_{\bar{d}} \left( \phi_\sharp P(\cdot \mid s,a), \bar{P}(\cdot \mid \phi(s),a) \right)$$

$$+ \gamma \cdot D_{\text{IR}}^{\sup}(\pi_{\text{b}}, \bar{\pi}) \underset{s,a \sim \xi_{\pi_{\text{b}}}}{\mathbb{E}} \left| \underset{s' \sim P(\cdot|s,a)}{\mathbb{E}} \left[ V^{\bar{\pi}}(s') - \bar{V}^{\bar{\pi}}(\phi(s')) \right] \right|$$

$$(\text{by Theorem 3 and the dual formulation of Wasserstein})$$

$$= D_{\text{IR}}^{\sup}(\pi_{\text{b}}, \bar{\pi}) \cdot \left( L_R^{\xi_{\pi_{\text{b}}}} + \gamma K_V \cdot L_P^{\xi_{\pi_{\text{b}}}} \right) + \gamma D_{\text{IR}}^{\sup}(\pi_{\text{b}}, \bar{\pi}) \cdot \underset{s,a \sim \xi_{\pi_{\text{b}}}}{\mathbb{E}} \left| \underset{s' \sim P(\cdot|s,a)}{\mathbb{E}} \left[ V^{\bar{\pi}}(s') - \bar{V}^{\bar{\pi}}(\phi(s')) \right] \right|$$

$$(\text{by definition of } L_P^{\xi_{\pi_{\text{b}}}})$$

$$\leq D_{\text{IR}}^{\sup}(\pi_{\text{b}}, \bar{\pi}) \cdot \left( L_R^{\xi_{\pi_{\text{b}}}} + \gamma K_V \cdot L_P^{\xi_{\pi_{\text{b}}}} \right) + \gamma D_{\text{IR}}^{\sup}(\pi_{\text{b}}, \bar{\pi}) \cdot \underset{s,a \sim \xi_{\pi_{\text{b}}}}{\mathbb{E}} \underset{s' \sim P(\cdot|s,a)}{\mathbb{E}} \left| V^{\bar{\pi}}(s') - \bar{V}^{\bar{\pi}}(\phi(s')) \right|$$

$$(\text{Jensen's inequality})$$

$$= D_{\text{IR}}^{\sup}(\pi_{\text{b}}, \bar{\pi}) \cdot \left( L_R^{\xi_{\pi_{\text{b}}}} + \gamma K_V \cdot L_P^{\xi_{\pi_{\text{b}}}} \right) + \gamma D_{\text{IR}}^{\sup}(\pi_{\text{b}}, \bar{\pi}) \cdot \underset{s \sim \xi_{\pi_{\text{b}}}}{\mathbb{E}} \left| V^{\bar{\pi}}(s) - \bar{V}^{\bar{\pi}}(\phi(s)) \right|$$

$$(\text{as } \xi_{\pi_{\text{b}}} \text{ is a stationary measure})$$

To summarize, we have:

$$\underset{s \sim \xi_{\pi_{\text{b}}}}{\mathbb{E}} \left| V^{\bar{\pi}}(s) - \bar{V}^{\bar{\pi}}(\phi(s)) \right| \leq D_{\text{IR}}^{\sup}(\pi_{\text{b}}, \bar{\pi}) \cdot \left( L_R^{\xi_{\pi_{\text{b}}}} + \gamma K_V \cdot L_P^{\xi_{\pi_{\text{b}}}} \right) + \gamma D_{\text{IR}}^{\sup}(\pi_{\text{b}}, \bar{\pi}) \cdot \underset{s \sim \xi_{\pi_{\text{b}}}}{\mathbb{E}} \left| V^{\bar{\pi}}(s) - \bar{V}^{\bar{\pi}}(\phi(s)) \right|.$$

Or equivalently,

$$(1 - \gamma D_{\text{IR}}^{\sup}(\pi_{\text{b}}, \bar{\pi})) \underset{s \sim \xi_{\pi_{\text{b}}}}{\mathbb{E}} \left| V^{\bar{\pi}}(s) - \bar{V}^{\bar{\pi}}(\phi(s)) \right| \leq D_{\text{IR}}^{\sup}(\pi_{\text{b}}, \bar{\pi}) \cdot \left( L_R^{\xi_{\pi_{\text{b}}}} + \gamma K_V \cdot L_P^{\xi_{\pi_{\text{b}}}} \right)$$

$$\underset{s \sim \xi_{\pi_{\text{b}}}}{\mathbb{E}} \left| V^{\bar{\pi}}(s) - \bar{V}^{\bar{\pi}}(\phi(s)) \right| \leq D_{\text{IR}}^{\sup}(\pi_{\text{b}}, \bar{\pi}) \cdot \frac{L_R^{\xi_{\pi_{\text{b}}}} + \gamma K_V \cdot L_P^{\xi_{\pi_{\text{b}}}}}{1 - \gamma D_{\text{IR}}^{\sup}(\pi_{\text{b}}, \bar{\pi})}$$

$$= \frac{L_R^{\xi_{\pi_{\text{b}}}} + \gamma K_V \cdot L_P^{\xi_{\pi_{\text{b}}}}}{1/D_{\text{IR}}^{\sup}(\pi_{\text{b}}, \bar{\pi}) - \gamma},$$

which is well-defined because $D_{\text{IR}}^{\sup}(\pi_{\text{b}}, \bar{\pi})$ is assumed strictly lower than $1/\gamma$. $\qquad\square$

In the main text, we made the assumption the environment is episodic. Let us formally restate this assumption:

**Assumption 2.** *The environment $\mathcal{M}$ and the world model $\bar{\mathcal{M}}$ are episodic.*

**Assumption 3.** $\forall s \in \mathcal{S}$, $\phi(s) = \bar{s}_{reset}$ *if and only if* $s = s_{reset}$.

Note that, as mentioned in Section 2, Assumption 2 ensures the existence of a stationary distribution $\xi_\pi$ and the ergodicity of both the original environment and the latent model. Assumption 3 guarantees that the reset states are aligned in the original and latent MDPs.

We are now ready to prove Theorem 2.

**Theorem 2.** *Suppose $\gamma > 1/2$ and $K_{\overline{P}}^{\overline{\pi}} < 1/\gamma$. Let $C \in (1, 1/\gamma)$, $\pi_b \in \Pi$ be the base policy, $(\overline{\pi} \circ \phi) \in \mathcal{N}^C(\pi_b)$ where $\overline{\pi} \in \overline{\overline{\Pi}}$ is a latent policy and $\phi \colon \mathcal{S} \to \overline{\mathcal{S}}$ a state representation. Then,*

$$\left| \rho(\overline{\pi} \circ \phi, \mathcal{M}) - \rho(\overline{\pi}, \overline{\mathcal{M}}) \right| \leq \mathrm{AEL}(\pi_b) \cdot \frac{L_R^{\xi_{\pi_b}}/\gamma + K_V \cdot L_P^{\xi_{\pi_b}}}{1/D_{\mathrm{IR}}^{\mathrm{sup}}(\pi_b, \overline{\pi}) - \gamma},$$

*where $\mathrm{AEL}(\pi_b)$ denotes the* average episode length *when $\mathcal{M}$ runs under $\pi_b$, $K_V = K_{\overline{R}}^{\overline{\pi}}/(1 - \gamma K_{\overline{P}}^{\overline{\pi}})$, and $L_R^{\xi_{\pi_b}}, L_P^{\xi_{\pi_b}}$ are the local losses of Eq. 4 over the stationary distribution $\xi_{\pi_b}$ induced by $\pi_b$.*

*Proof.* The first part of the proof follows by the expected value difference bound of Lemma 4. The second part of the proof follows by adapting of the one of Delgrange et al., 2025, Theorem 1, where the authors considered discrete latent MDPs and reach-avoid objectives (rewards were disregarded).

Our goal is to get rid of the expectation. First, note that for any *measurable state* so that $\xi_{\pi_b}(\{s\}) > 0$, we have $\left| V^{\overline{\pi}}(s) - \overline{V}^{\overline{\pi}}(\phi(s)) \right| \leq 1/\xi_{\pi_b}(\{s\}) \cdot \mathbb{E}_{s' \sim \xi_{\pi_b}} \left| V^{\overline{\pi}}(s') - \overline{V}^{\overline{\pi}}(\phi(s')) \right|$. For simplicity, we write $\xi_{\pi_b}(s)$ as shorthand for $\xi_{\pi_b}(\{s\})$ when considering such states. Second, note that as $s_{reset}$ is almost surely visited episodically (Assumption 2), *restarting* the MDP (i.e., visiting $s_{reset}$) is a measurable event, meaning that $s_{reset}$ has a non-zero probability $\xi_{\pi_b}(s_{reset}) \in (0, 1)$. Then,

$$\left| \rho(\overline{\pi} \circ \phi, \mathcal{M}) - \rho(\overline{\pi}, \overline{\mathcal{M}}) \right| \tag{8}$$

$$= \left| V^{\overline{\pi}}(s_I) - \overline{V}^{\overline{\pi}}(\overline{s}_I) \right| \tag{9}$$

$$= \frac{1}{\gamma} \left| \gamma \cdot V^{\overline{\pi}}(s_I) - \gamma \cdot \overline{V}^{\overline{\pi}}(\overline{s}_I) \right| \tag{10}$$

$$= \frac{1}{\gamma} \left| V^{\overline{\pi}}(s_{reset}) - \overline{V}^{\overline{\pi}}(\phi(s_{reset})) \right| \qquad \text{(by Assumptions 2 and 3)}$$

$$\leq \frac{1}{\gamma \cdot \xi_{\pi_b}(s_{reset})} \mathop{\mathbb{E}}_{s \sim \xi_{\pi_b}} \left| V^{\overline{\pi}}(s) - \overline{V}^{\overline{\pi}}(\phi(s)) \right| \tag{11}$$

$$\leq \frac{L_R^{\xi_{\pi_b}}/\gamma + K_V \cdot L_P^{\xi_{\pi_b}}}{\xi_{\pi_b}(s_{reset})(1/D_{\mathrm{IR}}^{\mathrm{sup}}(\pi_b, \overline{\pi}) - \gamma)}. \tag{12}$$

Finally, the result follows from the fact that $1/\xi_{\pi_b}(s_{reset})$ corresponds to the AEL. Indeed, when $\mathcal{M}$ is episodic, it is irreducible and recurrent (Huang, 2020); thus, given the random variable

$$\mathbf{T}_s(\tau = s_0, a_0, s_1, a_1, \ldots) = \sum_{T=1}^{\infty} T \cdot \mathbb{1}\left\{ s_T = s \text{ and } s_t \neq s \text{ for all } 0 < t < T \right\},$$

we have $\xi_\pi(s) = 1/\mathbb{E}_\pi[\mathbf{T}_s | s_0 = s]$ for any $s \in \mathcal{S}$ and stationary policy $\pi$, where $\mathbb{E}_\pi[\mathbf{T}_s \mid s_0 = s]$ is the *mean recurrence time* of $s$ under $\pi$ (Serfozo, 2009, Chapter 1, Theorem 54). In particular, this means that $1/\xi_{\pi_b}(s_{reset}) = \mathbb{E}_{\pi_b}[\mathbf{T}_{s_{reset}} \mid s_0 = s_{reset}] = \mathbb{E}_{\pi_b}[\mathbf{T}]$ is the AEL of $\mathcal{M}$ under $\pi_b$, which yields

$$\left| \rho(\overline{\pi} \circ \phi, \mathcal{M}) - \rho(\overline{\pi}, \overline{\mathcal{M}}) \right| \leq \mathbb{E}_{\pi_b}[\mathbf{T}] \cdot \frac{L_R^{\xi_{\pi_b}}/\gamma + K_V \cdot L_P^{\xi_{\pi_b}}}{1/D_{\mathrm{IR}}^{\mathrm{sup}}(\pi_b, \overline{\pi}) - \gamma}.$$

$\square$

*Remark* 2 (Extension to episodic value functions). In Lemma 4 and Theorem 2, we considered the standard definition of value function. One may wonder whether the results hold when considering episodic value functions, as defined in Appendix A. It turns out that it is the case, as one can easily adapt the proofs for those particular value functions.

We start by adapting the proof of Lemma 4:

$$\mathop{\mathbb{E}}_{s \sim \xi_{\pi_b}} \left| V^{\overline{\pi}}(s) - \overline{V}^{\overline{\pi}}(\phi(s)) \right|$$

$$= \mathop{\mathbb{E}}_{s \sim \xi_{\pi_b}} \left| \mathbb{1}\{s \neq s_{reset}\} \cdot \left( \mathop{\mathbb{E}}_{a \sim \bar{\pi}(\cdot|\phi(s))} \left[ R(s,a) + \gamma \mathop{\mathbb{E}}_{s' \sim P(\cdot|s,a)} \left[ V^{\bar{\pi}}(s') \right] \right] \right. \right.$$
$$\left. \left. - \mathop{\mathbb{E}}_{a \sim \bar{\pi}(\cdot|\phi(s))} \left[ \overline{R}(\phi(s),a) + \gamma \mathop{\mathbb{E}}_{\bar{s}' \sim \overline{P}(\cdot|\phi(s),a)} \left[ \overline{V}^{\bar{\pi}}(\bar{s}') \right] \right] \right) \right|$$
$$\leq \mathop{\mathbb{E}}_{s \sim \xi_{\pi_b}} \left| \mathop{\mathbb{E}}_{a \sim \bar{\pi}(\cdot|\phi(s))} \left[ R(s,a) + \gamma \mathop{\mathbb{E}}_{s' \sim P(\cdot|s,a)} \left[ V^{\bar{\pi}}(s') \right] \right] - \mathop{\mathbb{E}}_{a \sim \bar{\pi}(\cdot|\phi(s))} \left[ \overline{R}(\phi(s),a) + \gamma \mathop{\mathbb{E}}_{\bar{s}' \sim \overline{P}(\cdot|\phi(s),a)} \left[ \overline{V}^{\bar{\pi}}(\bar{s}') \right] \right] \right|.$$

The remaining of the proof is identical.

Concerning Theorem 2, we take a detour by defining a new value function $U$ as

$$U^{\bar{\pi}}(s) = \mathop{\mathbb{E}}_{a \sim \bar{\pi}(\cdot|\phi(s))} \left[ R(s,a) + \gamma \cdot \mathop{\mathbb{E}}_{s' \sim P(\cdot|s,a)} \left[ U^{\bar{\pi}}(s') \cdot \mathbb{1}\{s' \neq s_{reset}\} \right] \right] \qquad \forall s \in \mathcal{S}$$

The latent counterpart $\overline{U}^{\bar{\pi}}$ is defined similarly. By definition of the episodic value function (Appendix A) and since $V^{\bar{\pi}}(s_{reset}) = 0$, it is clear that

$$V^{\bar{\pi}}(s) = \begin{cases} U^{\bar{\pi}}(s) & \text{if } s \neq s_{reset} \\ U^{\bar{\pi}}(s) \cdot \mathbb{1}\{s \neq s_{reset}\} & \text{otherwise; and} \end{cases} \qquad \overline{V}^{\bar{\pi}}(\bar{s}) = \begin{cases} \overline{U}^{\bar{\pi}}(\bar{s}) & \text{if } \bar{s} \neq \phi(s_{reset}) \\ \overline{U}^{\bar{\pi}}(\bar{s}) \cdot \mathbb{1}\{\bar{s} \neq \phi(s_{reset})\} & \text{otherwise.} \end{cases}$$
$$(13)$$

Therefore,

$$\mathop{\mathbb{E}}_{s \sim \xi_{\pi_b}} \left| U^{\bar{\pi}}(s) - \overline{U}^{\bar{\pi}}(\phi(s)) \right|$$
$$\leq \mathop{\mathbb{E}}_{s,a \sim \xi_{\pi_b}} \left[ \frac{\bar{\pi}(a \mid \phi(s))}{\pi_b(a \mid s)} \cdot \left| R(s,a) - \overline{R}(\phi(s),a) \right| \right]$$
$$+ \gamma \mathop{\mathbb{E}}_{s,a \sim \xi_{\pi_b}} \left[ \frac{\bar{\pi}(a \mid \phi(s))}{\pi_b(a \mid s)} \cdot \left| \mathop{\mathbb{E}}_{s' \sim P(\cdot|s,a)} \left[ U^{\bar{\pi}}(s') \cdot \mathbb{1}\{s' \neq s_{reset}\} \right] - \mathop{\mathbb{E}}_{\bar{s}' \sim \overline{P}(\phi(s),a)} \left[ \overline{U}^{\bar{\pi}}(\bar{s}') \cdot \mathbb{1}\{\bar{s}' \neq \phi(s_{reset})\} \right] \right| \right]$$
$$\text{(Triangle inequality and importance sampling)}$$
$$= D_{IR}^{\sup}(\pi_b, \bar{\pi}) \cdot \mathop{\mathbb{E}}_{s,a \sim \xi_{\pi_b}} \left| R(s,a) - \overline{R}(\phi(s),a) \right| + \gamma D_{IR}^{\sup}(\pi_b, \bar{\pi}) \cdot \mathop{\mathbb{E}}_{s,a \sim \xi_{\pi_b}} \left| \mathop{\mathbb{E}}_{s' \sim P(\cdot|s,a)} V^{\bar{\pi}}(s') - \mathop{\mathbb{E}}_{\bar{s}' \sim \overline{P}(\cdot|\phi(s),a)} \overline{V}^{\bar{\pi}}(\bar{s}') \right|$$
$$\text{(by Eq. 13 and definition of the SIR)}$$
$$\leq D_{IR}^{\sup}(\pi_b, \bar{\pi}) \mathop{\mathbb{E}}_{s,a \sim \xi_{\pi_b}} \left| R(s,a) - \overline{R}(\phi(s),a) \right| + \gamma \cdot D_{IR}^{\sup}(\pi_b, \bar{\pi}) \mathop{\mathbb{E}}_{s,a \sim \xi_{\pi_b}} \left| \mathop{\mathbb{E}}_{s' \sim P(\cdot|s,a)} \left[ V^{\bar{\pi}}(s') - \overline{V}^{\bar{\pi}}(\phi(s')) \right] \right|$$
$$+ \gamma \cdot D_{IR}^{\sup}(\pi_b, \bar{\pi}) \mathop{\mathbb{E}}_{s,a \sim \xi_{\pi_b}} \left| \mathop{\mathbb{E}}_{\bar{s}' \sim \phi_\sharp P(\cdot|s,a)} \overline{V}^{\bar{\pi}}(\bar{s}') - \mathop{\mathbb{E}}_{\bar{s}' \sim \overline{P}(\cdot|\phi(s),a)} \overline{V}^{\bar{\pi}}(\bar{s}') \right|$$
$$\text{(Triangle inequality)}$$
$$\leq D_{IR}^{\sup}(\pi_b, \bar{\pi}) \cdot L_R^{\xi_{\pi_b}} + \gamma D_{IR}^{\sup}(\pi_b, \bar{\pi}) \cdot \mathop{\mathbb{E}}_{s \sim \xi_{\pi_b}} \left| V^{\bar{\pi}}(s) - \overline{V}^{\bar{\pi}}(\phi(s)) \right| + \gamma D_{IR}^{\sup}(\pi_b, \bar{\pi}) \cdot K_V \cdot L_P^{\xi_{\bar{\pi}}}$$
$$\text{(by the same developments as in the proof of Lemma 4)}$$
$$\leq D_{IR}^{\sup}(\pi_b, \bar{\pi}) \cdot L_R^{\xi_{\bar{\pi}}} + \gamma D_{IR}^{\sup}(\pi_b, \bar{\pi}) \cdot \frac{L_R^{\xi_{\pi_b}} + \gamma K_V \cdot L_P^{\xi_{\pi_b}}}{1/D_{IR}^{\sup}(\pi_b, \bar{\pi}) - \gamma} + \gamma D_{IR}^{\sup}(\pi_b, \bar{\pi}) \cdot K_V \cdot L_P^{\xi_{\bar{\pi}}} \qquad \text{(Lemma 4)}$$
$$= D_{IR}^{\sup}(\pi_b, \bar{\pi}) \left( L_R^{\xi_{\pi_b}} \left( 1 + \frac{\gamma}{D_{IR}^{\sup}(\pi_b, \bar{\pi})^{-1} - \gamma} \right) + \gamma K_V \cdot L_P^{\xi_{\pi_b}} \left( 1 + \frac{\gamma}{D_{IR}^{\sup}(\pi_b, \bar{\pi})^{-1} - \gamma} \right) \right)$$
$$= D_{IR}^{\sup}(\pi_b, \bar{\pi}) \left( L_R^{\xi_{\pi_b}} \cdot \gamma K_V \cdot L_P^{\xi_{\pi_b}} \right) \left( 1 + \frac{\gamma}{D_{IR}^{\sup}(\pi_b, \bar{\pi})^{-1} - \gamma} \right)$$
$$= \frac{L_R^{\xi_{\pi_b}} + \gamma K_V \cdot L_P^{\xi_{\pi_b}}}{1/D_{IR}^{\sup}(\pi_b, \bar{\pi}) - \gamma}.$$

Now, in the proof of Theorem 2, it suffices to replace Equation 9 by observing that, in the episodic case, we have

$$\left|\rho(\bar{\pi} \circ \phi, \mathcal{M}) - \rho(\bar{\pi}, \overline{\mathcal{M}})\right| = \left|V^{\bar{\pi}}(s_I) - \overline{V}^{\bar{\pi}}(\bar{s}_I)\right| = \left|U^{\bar{\pi}}(s_I) - \overline{U}^{\bar{\pi}}(\bar{s}_I)\right| \qquad \text{(again, by Equation 13)}$$

$$= \frac{1}{\gamma}\left|\gamma \cdot U^{\bar{\pi}}(s_I) - \gamma \cdot \overline{U}^{\bar{\pi}}(\bar{s}_I)\right| = \frac{1}{\gamma}\left|U^{\bar{\pi}}(s_{reset}) - \overline{U}^{\bar{\pi}}(\phi(s_{reset}))\right|$$

Modulo this change, the remaining of the proof remains identical; one just needs to replace the occurrences of $\mathbb{E}_{s \sim \xi_{\pi_b}}\left|V^{\bar{\pi}}(s) - \overline{V}^{\bar{\pi}}(\phi(s))\right|$ by $\mathbb{E}_{s \sim \xi_{\pi_b}}\left|U^{\bar{\pi}}(s) - \overline{U}^{\bar{\pi}}(\phi(s))\right|$.

Since the subsequent results all rely on Lemma 4 and Theorem 2, they all extend to episodic value functions.

**Theorem 3.** (Deep, Safe Policy Improvement) *Under the same preamble as in Thm. 2, assume that $\phi$ if fixed during the policy update and the behavioral is a latent policy with $\pi_b := \bar{\pi}_b \circ \phi$ and $\bar{\pi}_b \in \overline{\Pi}$. Then, the improvement of the return of $\mathcal{M}$ under $\bar{\pi}$ can be guaranteed on $\pi_b$ as*

$$\rho(\bar{\pi} \circ \phi, \mathcal{M}) - \rho(\pi_b, \mathcal{M}) \geq \rho(\bar{\pi}, \overline{\mathcal{M}}) - \rho(\bar{\pi}_b, \overline{\mathcal{M}}) - \zeta,$$

$$\text{where } \zeta := \text{AEL}(\pi_b) \cdot \left(L_R^{\xi_{\pi_b}}/\gamma + K_V L_P^{\xi_{\pi_b}}\right)\left(\frac{1}{1/D_{IR}^{\sup}(\pi_b, \bar{\pi}) - \gamma} + \frac{1}{1 - \gamma}\right).$$

*Proof.* First, note that

$$\rho(\bar{\pi} \circ \phi, \mathcal{M}) - \rho(\pi_b, \mathcal{M})$$
$$= \rho(\bar{\pi} \circ \phi, \mathcal{M}) - \rho(\bar{\pi}, \overline{\mathcal{M}}) + \rho(\bar{\pi}, \overline{\mathcal{M}}) - \rho(\pi_b, \mathcal{M}). \tag{14}$$

By Theorem 2, we have with $D_{IR}^{\sup}(\pi_b, \pi_b) = 1$ that

$$\left|\rho(\pi_b, \mathcal{M}) - \rho(\bar{\pi}_b, \overline{\mathcal{M}})\right| \leq \mathbb{E}_{\pi_b}^{\mathcal{M}}[\mathbf{T}] \cdot \frac{L_R^{\xi_{\pi_b}}/\gamma + K_V \cdot L_P^{\xi_{\pi_b}}}{1 - \gamma},$$

which implies that

$$\rho(\pi_b, \mathcal{M}) - \rho(\bar{\pi}_b, \overline{\mathcal{M}}) \leq \mathbb{E}_{\pi_b}^{\mathcal{M}}[\mathbf{T}] \cdot \frac{L_R^{\xi_{\pi_b}}/\gamma + K_V \cdot L_P^{\xi_{\pi_b}}}{1 - \gamma}$$

$$\iff \quad \rho(\pi_b, \mathcal{M}) \leq \rho(\bar{\pi}_b, \overline{\mathcal{M}}) + \mathbb{E}_{\pi_b}^{\mathcal{M}}[\mathbf{T}] \cdot \frac{L_R^{\xi_{\pi_b}}/\gamma + K_V \cdot L_P^{\xi_{\pi_b}}}{1 - \gamma}. \tag{15}$$

On the other hand, we have

$$\left|\rho(\bar{\pi} \circ \phi, \mathcal{M}) - \rho(\bar{\pi}, \overline{\mathcal{M}})\right| \leq \mathbb{E}_{\pi_b}^{\mathcal{M}}[\mathbf{T}] \cdot \frac{L_R^{\xi_{\pi_b}}/\gamma + K_V \cdot L_P^{\xi_{\pi_b}}}{1/D_{IR}^{\sup}(\pi_b, \bar{\pi}) - \gamma},$$

which implies that

$$\rho(\bar{\pi} \circ \phi, \mathcal{M}) - \rho(\bar{\pi}, \overline{\mathcal{M}}) \geq -\mathbb{E}_{\pi_b}^{\mathcal{M}}[\mathbf{T}] \cdot \frac{L_R^{\xi_{\pi_b}}/\gamma + K_V \cdot L_P^{\xi_{\pi_b}}}{1/D_{IR}^{\sup}(\pi_b, \bar{\pi}) - \gamma}. \tag{16}$$

By plugging Equations 15 and 16 into Equation 14, we get the desired result:

$$\rho(\bar{\pi} \circ \phi, \mathcal{M}) - \rho(\pi_b, \mathcal{M})$$

$$= \quad \underbrace{\rho(\bar{\pi} \circ \phi, \mathcal{M}) - \rho(\bar{\pi}, \overline{\mathcal{M}})}_{\substack{\geq \\ -\mathbb{E}_{\pi_b}^{\mathcal{M}}[\mathbf{T}] \cdot \frac{L_R^{\xi_{\pi_b}}/\gamma + K_V \cdot L_P^{\xi_{\pi_b}}}{1/D_{IR}^{\sup}(\pi_b, \bar{\pi}) - \gamma}}} + \rho(\bar{\pi}, \overline{\mathcal{M}}) - \underbrace{\rho(\pi_b, \mathcal{M})}_{\substack{\leq \\ \rho(\bar{\pi}_b, \overline{\mathcal{M}}) + \mathbb{E}_{\pi_b}^{\mathcal{M}}[\mathbf{T}] \cdot \frac{L_R^{\xi_{\pi_b}}/\gamma + K_V \cdot L_P^{\xi_{\pi_b}}}{1 - \gamma}}}$$

$$\geq -\mathbb{E}_{\pi_b}^{\mathcal{M}}[\mathbf{T}] \cdot \frac{L_R^{\xi_{\pi_b}}/\gamma + K_V \cdot L_P^{\xi_{\pi_b}}}{1/D_{IR}^{\sup}(\pi_b, \bar{\pi}) - \gamma} + \rho(\bar{\pi}, \overline{\mathcal{M}}) - \rho(\bar{\pi}_b, \overline{\mathcal{M}}) - \mathbb{E}_{\pi_b}^{\mathcal{M}}[\mathbf{T}] \cdot \frac{L_R^{\xi_{\pi_b}}/\gamma + K_V \cdot L_P^{\xi_{\pi_b}}}{1 - \gamma}$$

$$= \rho(\bar{\pi}, \overline{\mathcal{M}}) - \rho(\bar{\pi}_b, \overline{\mathcal{M}}) - \mathbb{E}_{\pi_b}^{\mathcal{M}}[\mathbf{T}]\left(L_R^{\xi_{\pi_b}}/\gamma + K_V L_P^{\xi_{\pi_b}}\right)\left(\frac{1}{1/D_{IR}^{\sup}(\pi_b, \bar{\pi}) - \gamma} + \frac{1}{1 - \gamma}\right).$$

$\square$

In the following, we provide a probabilistic version of Theorem 3, as it is standard in the SPI literature. Essentially, we derive probably approximately correct estimations from interaction data of $L_R, L_P$. Then, we use those estimations to get an approximation of $\zeta$, the error term of the safe policy improvement inequality of Theorem 3.

Those PAC guarantees rely on a discrete latent space. While it may seem restrictive, learning discrete latent spaces turns out to be beneficial not only theoretically (e.g., it yields trivial Lipschitz bounds on the latent reward and transition functions), but also in practice (see, e.g., Hafner et al., 2021).

Finally, note that we provide two versions of the theorem: (1) one where we have access to an upper bound of the AEL (which is mild in practice), and (2) another one where this bound cannot be derived. The latter case yields an additional challenge as we need to estimate the AEL from sample states drawn according to the stationary distribution. In this case, the bound yields a probabilistic algorithm that is guaranteed to almost surely terminate without any predefined endpoint, as it depends on the current approximation of the losses.

**Theorem 5** (Probabilistic Deep SPI with confidence bound). *Under the same preamble as in Theorem 3, assume now $\bar{S}$ is discrete. Let $\{\langle s_t, a_t, r_t, s'_t \rangle : 1 \le t \le T\}$ be a set of $T$ transitions drawn from $\xi_{\pi_b}$ by simulating $\mathcal{M}_{\pi_b}$, i.e., $s_t \sim \xi_{\pi_b}$, $a_t \sim \pi_b(\cdot \mid s_t)$, $r_t = R(s_t, a_t)$, and $s'_t \sim P(\cdot \mid s_t, a_t)$ for all $1 \le t \le T$. Let $\varepsilon, \delta > 0$ and define*

$$\hat{L}_P := 1 - \frac{1}{T} \sum_{t=1}^{T} \overline{P}(\phi(s'_t) \mid \phi(s_t), a_t), \quad \hat{L}_R := \frac{1}{T} \sum_{t=1}^{T} \left| r_t - \overline{R}(\phi(s), a) \right|, \quad \hat{\xi}_{reset} := \frac{1}{T} \sum_{t=0}^{T} \mathbb{1}\{s_t = s_{reset}\},$$

$\kappa := \frac{1}{1/D_{IR}^{\sup}(\pi_b, \bar{\pi}) - \gamma} + \frac{1}{1-\gamma}$, *and* $R^* := \max\{1, 4R_{\mathrm{MAX}}^2\}$. *Then, the policy can be safely improved as*

$$\rho(\bar{\pi} \circ \phi, \mathcal{M}) - \rho(\pi_b, \mathcal{M}) \ge \rho(\bar{\pi}, \overline{\mathcal{M}}) - \rho(\bar{\pi}_b, \overline{\mathcal{M}}) - \hat{\zeta}, \tag{17}$$

*with probability at least $1 - \delta$ under the following conditions:*

*(1)  one has access to an upper bound $L \ge \mathrm{AEL}(\pi_b)$, the number of collected transitions is lower-bounded by*
$T \ge L^2 \cdot \left\lceil \frac{-R^* \log\left(\frac{\delta}{2} \cdot \kappa^2 (1/\gamma + K_V)^2\right)}{\varepsilon^2} \right\rceil$, *and* $\hat{\zeta} := L \cdot \left( \hat{L}_R/\gamma + K_V \hat{L}_P \right)\kappa + \varepsilon$; *or*

*(2)  without access to such a bound, we take*

$$T \ge \left\lceil \frac{-R^* \log(\delta/3)}{2} \cdot \max\left\{ 1/\hat{\xi}_{reset}^2, \left( \frac{\kappa/\hat{\xi}_{reset}\left( \hat{L}_R/\gamma + K_V \hat{L}_P \right) + \varepsilon + \kappa \cdot (1/\gamma + K_V)}{\varepsilon \hat{\xi}_{reset}} \right)^2 \right\} \right\rceil,$$

*and* $\hat{\zeta} := \frac{1}{\hat{\xi}_{reset}}\left( \hat{L}_R/\gamma + K_V \hat{L}_P \right)\kappa + \varepsilon$.

*Proof.* Let $\varepsilon, \delta > 0$. First, note that we need $T \ge \left\lceil \frac{-R^* \log(\delta/2)}{\varepsilon^2} \right\rceil$, to satisfy both (a) $\hat{L}_R + \varepsilon > L_R^{\xi_{\pi_b}}$ and (b) $\hat{L}_P + \varepsilon > L_P^{\xi_{\pi_b}}$ with probability $1 - \delta$ and $T \ge \left\lceil \frac{-R^* \log(\delta/3)}{\varepsilon^2} \right\rceil$ to satisfy simultaneously (a), (b), and (c) $\hat{\xi}_{reset} - \varepsilon < \xi_{\pi_b}(s_{reset})$ with probability $1 - \delta$. This statement is proven by Delgrange et al. (2022) and Delgrange et al. (2025). The result is essentially due to a raw application of Hoeffding's inequality and the fact that Wasserstein boils down to total variation when the state space is discrete (Villani, 2009).

Let $\varepsilon' > 0$.

**Case 1.** Assume we have an upper bound on $\mathrm{AEL}(\pi_b)$, say $L$. Then it follows that

$$\zeta \le L \cdot \left( \frac{L_R^{\xi_{\pi_b}}}{\gamma} + K_V L_P^{\xi_{\pi_b}} \right) \cdot \kappa \qquad (\zeta \text{ is the safe policy improvement error term of Theorem 3})$$

$$\le L \cdot \left( \frac{\hat{L}_R + \varepsilon'}{\gamma} + K_V (\hat{L}_P + \varepsilon') \right) \cdot \kappa,$$

with probability at least $1 - \delta$ whenever

$$T \ge \frac{-R^* \log(\delta/2)}{\varepsilon'^2}.$$

To ensure an error of at most $\varepsilon$, choose $\varepsilon'$ such that

$$L \cdot \left( \frac{\hat{L}_R + \varepsilon'}{\gamma} + K_V (\hat{L}_P + \varepsilon') \right)\kappa \le L \cdot \left( \frac{\hat{L}_R}{\gamma} + K_V \hat{L}_P \right)\kappa + \varepsilon.$$

Equivalently,

$$L\kappa\left(\tfrac{\varepsilon'}{\gamma} + K_V\varepsilon'\right) \le \varepsilon$$
$$\iff \varepsilon' \le \frac{\varepsilon}{L\kappa\left(1/\gamma + K_V\right)}.$$

Thus, it suffices that

$$T \ge \frac{-R^* \log(\delta/2)}{\varepsilon'^2} \ge \frac{-R^* \log(\delta/2)}{\varepsilon^2}\left(L\kappa\left(1/\gamma + K_V\right)\right)^2$$

to satisfy $\zeta \le \hat{\zeta}$ with probability at least $1 - \delta$.

**Case 2.** Suppose we do not have an upper bound on $\mathrm{AEL}(\pi_b)$. From the proof of Theorem 2, we know that $\mathrm{AEL}(\pi_b) = 1/\xi_{\pi_b}(s_{reset})$. In this case we include an estimate $\hat{\xi}_{reset}$ in the bound and use the high-probability deviations

$$\hat{L}_R + \varepsilon' > L_R^{\xi_{\pi_b}}, \quad \hat{L}_P + \varepsilon' > \hat{L}_P, \quad \hat{\xi}_{reset} - \varepsilon' < \xi_{\pi_b}(s_{reset}).$$

We have

$$\zeta = \frac{1}{\xi_{\pi_b}(s_{reset})}\left(\tfrac{L_R}{\gamma} + K_V L_P\right)\kappa \tag{18}$$

$$\le \frac{1}{\hat{\xi}_{reset} - \varepsilon'}\left(\tfrac{\hat{L}_R + \varepsilon'}{\gamma} + K_V(\hat{L}_P + \varepsilon')\right)\kappa, \tag{19}$$

with probability at least $1 - \delta$ whenever

$$T \ge \frac{R^* \log(\delta/3)}{2\,\varepsilon'^2}.$$

To guarantee an error at most $\varepsilon$, we require

$$\frac{1}{\hat{\xi}_{reset}}\left(\tfrac{\hat{L}_R}{\gamma} + K_V\hat{L}_P\right)\kappa + \varepsilon \ge \frac{1}{\hat{\xi}_{reset} - \varepsilon'}\left(\tfrac{\hat{L}_R + \varepsilon'}{\gamma} + K_V(\hat{L}_P + \varepsilon')\right)\kappa. \tag{20}$$

Assuming $\varepsilon' < \hat{\xi}_{reset}$, we multiply both sides of (20) by $(\hat{\xi}_{reset} - \varepsilon')$ and expand:

$$\left(\tfrac{\hat{L}_R}{\gamma} + K_V\hat{L}_P\right)\kappa\left(1 - \tfrac{\varepsilon'}{\hat{\xi}_{reset}}\right) + \varepsilon\,\hat{\xi}_{reset} - \varepsilon\,\varepsilon'$$
$$\ge \left(\tfrac{\hat{L}_R}{\gamma} + K_V\hat{L}_P\right)\kappa + \left(\tfrac{1}{\gamma} + K_V\right)\kappa\,\varepsilon'.$$

Cancel the common term $\left(\tfrac{\hat{L}_R}{\gamma} + K_V\hat{L}_P\right)\kappa$ and group the $\varepsilon'$ terms:

$$\varepsilon\,\hat{\xi}_{reset} \ge \varepsilon'\left[\tfrac{\kappa}{\hat{\xi}_{reset}}\left(\tfrac{\hat{L}_R}{\gamma} + K_V\hat{L}_P\right) + \varepsilon + \left(\tfrac{1}{\gamma} + K_V\right)\kappa\right].$$

Therefore a sufficient condition is the explicit upper bound

$$\varepsilon' < \min\left\{\hat{\xi}_{reset},\ \frac{\varepsilon\,\hat{\xi}_{reset}}{\tfrac{\kappa}{\hat{\xi}_{reset}}\left(\tfrac{\hat{L}_R}{\gamma} + K_V\hat{L}_P\right) + \varepsilon + \left(\tfrac{1}{\gamma} + K_V\right)\kappa}\right\}. \tag{21}$$

Together with the concentration requirement on $T$, the choice (21) ensures an error on $\zeta$ of at most $\varepsilon$ with probability at least $1 - \delta$.

Finally, the safe policy improvement bound follows from the fact that $\hat{\zeta}$ is greater than $\zeta$ with probability $1 - \delta$. Then, due to the SPI bound of Theorem 3, the improvement is guaranteed to be even larger when using $\zeta$ instead of $\hat{\zeta}$ as error term. This guarantees the improvement when $\hat{\zeta}$ is small enough. $\qquad\square$

*Remark* 3 (Episodic assumption). For the sake of presentation, we have considered and proved the bounds for episodic processes (cf. Appendix A). One could extend them to more general cases under the assumption that one has access to a stationary distribution $\xi_{\pi_b}$ of $\mathcal{M}$. As mentioned in Section 2, the existence of a stationary distribution is often assumed in continual RL (Sutton and Barto, 2018) and guaranteed unique in the episodic case (Huang, 2020). Then, replacing the difference of returns in Theorem 3 by an expectation (similar to Theorem 2 with Lemma 4) would allow to remove the AEL term and obtain similar results.

**Theorem 4.** (Deep SPI for representation learning) *Under the same preamble as in Thm. 2, let $\varepsilon > 0$ and* $\delta := 4 \cdot \frac{L_R^{\xi_{\pi_b}} + \gamma K_V \cdot L_P^{\xi_{\pi_b}}}{\varepsilon \cdot \left(1/D_{\mathrm{IR}}^{\sup}(\pi_b, \bar{\pi}) - \gamma\right)}$. *Then, with probability at least $1 - \delta$ under $\xi_{\pi_b}$, we have for all $s_1, s_2 \in \mathcal{S}$ that*

$$\left| V^{\bar{\pi}}(s_1) - V^{\bar{\pi}}(s_2) \right| \leq K_V \cdot \bar{d}(\phi(s_1), \phi(s_2)) + \varepsilon.$$

*Proof.* First, let us consider bounding the following absolute value difference for every possible state $s \in \mathcal{S}$, i.e., $\left| V^{\bar{\pi}}(s) - \overline{V}^{\bar{\pi}}(\phi(s)) \right|$. To that aim, we consider Markov's inequality:[4]

$$\xi_{\pi_b}\left(\left\{ s \in \mathcal{S} \colon \left| V^{\bar{\pi}}(s) - \overline{V}^{\bar{\pi}}(\phi(s)) \right| > \varepsilon/2 \right\}\right)$$
$$\leq \xi_{\pi_b}\left(\left\{ s \in \mathcal{S} \colon \left| V^{\bar{\pi}}(s) - \overline{V}^{\bar{\pi}}(\phi(s)) \right| \geq \varepsilon/2 \right\}\right)$$
$$\leq 2 \cdot \frac{\mathbb{E}_{s \sim \xi_{\pi_b}} \left| V^{\bar{\pi}}(s) - \overline{V}^{\bar{\pi}}(\phi(s)) \right|}{\varepsilon} \qquad \text{(Markov's inequality)}$$
$$\leq 2 \cdot \frac{L_R^{\xi_{\pi_b}} + \gamma K_V \cdot L_P^{\xi_{\pi_b}}}{\varepsilon \cdot \left(1/D_{\mathrm{IR}}^{\sup}(\pi_b, \bar{\pi}) - \gamma\right)}. \qquad \text{(by Lemma 4)}$$

Consider *any* joint distribution $\lambda \in \Lambda(\xi_{\pi_b}, \xi_{\pi_b})$, i.e., any joint distribution over $\mathcal{S} \times \mathcal{S}$ whose marginals both match $\xi_{\pi_b}$. Then, by the union bound, we have

$$\lambda\left(\left\{ \langle s_1, s_2 \rangle \in \mathcal{S} \times \mathcal{S} \colon \left| V^{\bar{\pi}}(s_1) - \overline{V}^{\bar{\pi}}(\phi(s_1)) \right| > \varepsilon/2 \text{ or } \left| V^{\bar{\pi}}(s_2) - \overline{V}^{\bar{\pi}}(\phi(s_2)) \right| > \varepsilon/2 \right\}\right)$$
$$\leq \lambda\left(\left\{ \langle s_1, s_2 \rangle \in \mathcal{S} \times \mathcal{S} \colon \left| V^{\bar{\pi}}(s_1) - \overline{V}^{\bar{\pi}}(\phi(s_1)) \right| \geq \varepsilon/2 \text{ or } \left| V^{\bar{\pi}}(s_2) - \overline{V}^{\bar{\pi}}(\phi(s_2)) \right| \geq \varepsilon/2 \right\}\right)$$
$$\leq \lambda\left(\left\{ \langle s_1, s_2 \rangle \in \mathcal{S} \times \mathcal{S} \colon \left| V^{\bar{\pi}}(s_1) - \overline{V}^{\bar{\pi}}(\phi(s_1)) \right| \geq \varepsilon/2 \right\}\right) + \lambda\left(\left\{ \langle s_1, s_2 \rangle \in \mathcal{S} \times \mathcal{S} \colon \left| V^{\bar{\pi}}(s_2) - \overline{V}^{\bar{\pi}}(\phi(s_2)) \right| \geq \varepsilon/2 \right\}\right)$$
$$\text{(union bound)}$$
$$= \xi_{\pi_b}\left(\left\{ s_1 \in \mathcal{S} \colon \left| V^{\bar{\pi}}(s_1) - \overline{V}^{\bar{\pi}}(\phi(s_1)) \right| \geq \varepsilon/2 \right\}\right) + \xi_{\pi_b}\left(\left\{ s_2 \in \mathcal{S} \colon \left| V^{\bar{\pi}}(s_2) - \overline{V}^{\bar{\pi}}(\phi(s_2)) \right| \geq \varepsilon/2 \right\}\right)$$
$$(\lambda \text{ has } \xi_{\pi_b} \text{ as marginal distributions})$$
$$\leq 4 \cdot \frac{L_R^{\xi_{\pi_b}} + \gamma K_V \cdot L_P^{\xi_{\pi_b}}}{\varepsilon \cdot \left(1/D_{\mathrm{IR}}^{\sup}(\pi_b, \bar{\pi}) - \gamma\right)}.$$

Therefore, since this holds for any such $\lambda$, we have with at least probability $1 - \delta$ that for all $s_1, s_2 \in \mathcal{S}$, $\left| V^{\bar{\pi}}(s_1) - \overline{V}^{\bar{\pi}}(\phi(s_1)) \right| \leq \varepsilon/2$ and $\left| V^{\bar{\pi}}(s_2) - \overline{V}^{\bar{\pi}}(\phi(s_2)) \right| \leq \varepsilon/2$. In consequence, with same probability, we have

$$\left| V^{\bar{\pi}}(s_1) - V^{\bar{\pi}}(s_2) \right|$$
$$= \left| V^{\bar{\pi}}(s_1) - \overline{V}^{\bar{\pi}}(\phi(s_1)) + \overline{V}^{\bar{\pi}}(\phi(s_1)) - \overline{V}^{\bar{\pi}}(\phi(s_2)) + \overline{V}^{\bar{\pi}}(\phi(s_2)) - V^{\bar{\pi}}(s_2) \right|$$
$$\leq \left| V^{\bar{\pi}}(s_1) - \overline{V}^{\bar{\pi}}(\phi(s_1)) \right| + \left| \overline{V}^{\bar{\pi}}(\phi(s_1)) - \overline{V}^{\bar{\pi}}(\phi(s_2)) \right| + \left| V^{\bar{\pi}}(s_2) - \overline{V}^{\bar{\pi}}(\phi(s_2)) \right| \qquad \text{(triangle inequality)}$$
$$\leq \left| \overline{V}^{\bar{\pi}}(\phi(s_1)) - \overline{V}^{\bar{\pi}}(\phi(s_2)) \right| + \varepsilon$$
$$\leq K_V \cdot \bar{d}(\phi(s_1), \phi(s_2)) + \varepsilon. \qquad \text{(by Lemma 3)}$$

$\square$

----

[4] also referred to as Chebyshev's inequality (Stein and Shakarchi, 2005).

## F  DREAM SPI

---
**Algorithm 2:** `DreamSPI`

---
**Input:** (others) world model and encoder parameters $\vartheta$, actor/critic parameters $\iota$, imagination
      horizon $H$

Init. $\boldsymbol{s} \in \mathcal{S}^{(T+1)\times B}, \boldsymbol{a} \in \mathcal{A}^{T\times B}, \boldsymbol{r} \in \mathbb{R}^{T\times B}, \bar{\boldsymbol{s}} \in \bar{\mathcal{S}}^{(H+1)\times BT}, \bar{\boldsymbol{a}} \in \mathcal{A}^{H\times BT}, \bar{\boldsymbol{r}} \in \mathbb{R}^{H\times BT}$

**repeat**

    **for** $t \leftarrow 1$ *to* $T$ **do**

        $\boldsymbol{a}_t \sim \bar{\pi}(\cdot \mid \phi(\boldsymbol{s}_t))$

        $\boldsymbol{r}_t, \boldsymbol{s}_{t+1} \leftarrow \texttt{env.step}(\boldsymbol{s}_t, \boldsymbol{a}_t)$

    Update $\vartheta$ by descending $\nabla_\vartheta \texttt{DeepSPI\_loss}(\boldsymbol{s}, \boldsymbol{a}, \boldsymbol{r}, U^{\bar{\pi}\circ\phi}, \vartheta)$

                                        ▷ *Only $\phi$, $\overline{P}$, and $\overline{R}$ are updated here*

    $\texttt{world\_model} \leftarrow \langle \bar{\mathcal{S}}, \mathcal{A}, \overline{P}, \overline{R} \rangle$

    Set latent start states: $\bar{\boldsymbol{s}}_1 \leftarrow \{\phi(\boldsymbol{s}_{t,i}) : 1 \le t \le T, \, 1 \le i \le B\}$

    Perform latent imagination:

    **for** $t \leftarrow 1$ *to* $H$ **do**

        $\bar{\boldsymbol{a}}_t \sim \bar{\pi}(\cdot \mid \bar{\boldsymbol{s}}_t)$

        $\bar{\boldsymbol{r}}_t, \bar{\boldsymbol{s}}_{t+1} \leftarrow \texttt{world\_model.step}(\bar{\boldsymbol{s}}_t, \bar{\boldsymbol{a}}_t)$

    Update $\iota$ by descending $\nabla_\iota \texttt{ppo\_loss}(\bar{\boldsymbol{s}}, \bar{\boldsymbol{a}}, \bar{\boldsymbol{r}}, A^{\bar{\pi}}, \iota)$

         ▷ *Perform a standard PPO update of the actor/critic w.r.t. the imagined trajectories*

    $\boldsymbol{s}_1 \leftarrow \boldsymbol{s}_{T+1}$

**until** *convergence*

**return** $\theta$

---

We report in Algorithm 2 the algorithm we used in our experiments to evaluate the quality of the world model's predictions. Note that the algorithm is on-policy; we leverage parallelized environments to make sure data coming from the interaction covers sufficiently the state space (Mayor et al., 2025). Empirically, we found most beneficial to use discrete latent spaces, and model the transition function with categorical distributions (32 classes of 32 categories, as in `Dreamer`; Hafner et al., 2021). This observation agrees with the observation made by Hafner et al. (2021) on the benefits of categorical latent spaces in world models.

## G  ADDITIONAL DETAILS ON THE ILLUSTRATIVE EXAMPLE

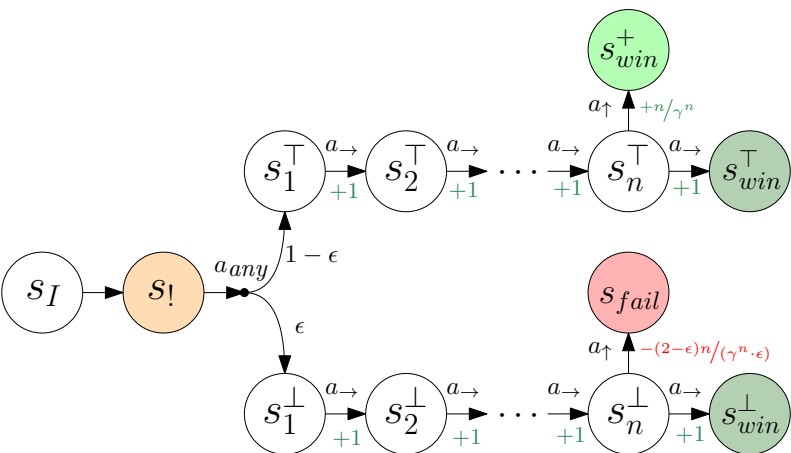

Figure 9: Underlying MDP of the grid world of Fig. 3. Actions leading to self-loops are omitted for clarity.

In this section, we expand on the illustrative example introduced in Sect. 6.1. The underlying MDP for the grid world is shown in Fig.9. Formally, the MDP has four actions $a_{\text{dir}}$ with dir $\in \{\uparrow, \downarrow, \rightarrow, \leftarrow\}$, and $2n + 6$ states:

- the initial state $s_I$, which transitions to $s_!$ whenever $a_\rightarrow$ is played;

- the hazardous state $s_!$, sending the agent to $s_1^\top$ with probability $1-\epsilon$ and to $s_1^\perp$ with probability $\epsilon$, independently of the action played;
- the $2n$ corridor states $s_i^\top$ and $s_i^\perp$ for $i \in \{1, \ldots, n\}$, forming the top and bottom branches; and
- the terminal states $s_{win}^\top$, $s_{win}^\perp$, $s_{win}^+$, and $s_{fail}$.

We focus on the value of the initial state $V^\pi(s_I)$ for policies $\pi \in \Pi$. We highlight three policies of particular interest:

(a) **The good** policy $\pi_{\text{good}}$: the policy moves right everywhere except in $s_n^\top$, where it chooses $a_\uparrow$:

$$V^{\pi_{\text{good}}}(s_I) = \gamma\Big[(1-\epsilon)\Big(\sum_{t=1}^{n-1}\gamma^t + \gamma^n \cdot \frac{n}{\gamma^n}\Big) + \epsilon\Big(\sum_{t=1}^{n}\gamma^t\Big)\Big]$$

$$= \gamma\left(\sum_{t=1}^{n-1}\gamma^t + (1-\epsilon)n + \epsilon\gamma^n\right)$$

$$= \gamma\left(\frac{\gamma-\gamma^n}{1-\gamma} + (1-\epsilon)n + \epsilon\gamma^n\right).$$

Learning $\pi_{\text{good}}$ requires that the representation distinguishes the two branches and assigns distinct latent states to $s_n^\top$ and $s_n^\perp$. With $n = 5$, this corresponds to a return of $\approx 8.01$, which is the value reported in Fig. 4 for `DeepSPI`. This highlight the representation learning capabilities of our algorithm.

(b) **The bad** policy $\pi_{\text{bad}}$: the policy moves right everywhere except in $s_n^\top$ and $s_n^\perp$, where it chooses $a_\uparrow$:

$$V^{\pi_{\text{bad}}}(s_I) = \gamma\Big[(1-\epsilon)\Big(\sum_{t=1}^{n-1}\gamma^t + \gamma^n \cdot \frac{n}{\gamma^n}\Big) + \epsilon\Big(\sum_{t=1}^{n-1}\gamma^t + \gamma^n \cdot \frac{-(2-\epsilon)n}{\gamma^n\epsilon}\Big)\Big]$$

$$= \gamma\left(\gamma\sum_{t=0}^{n-2}\gamma^t - n\right)$$

$$= \gamma\left(\frac{\gamma-\gamma^n}{1-\gamma} - n\right)$$

$$< 0.$$

Such a policy may arise due to the policy confounding update described in Sect. 3.2, where the representation incorrectly merges $s_n^\top$ and $s_n^\perp$. With $n = 5$, this corresponds to a return of $\approx -1.09$.

(c) **... and the ugly** "*always right*" policy $\pi_\to$: this policy deterministically selects $a_\to$ in every state. Its value is

$$V^{\pi_\to}(s_I) = \gamma\sum_{t=1}^{n}\gamma^t = \gamma^2\sum_{t=0}^{n-1}\gamma^t = \frac{\gamma^2-\gamma^{n+2}}{1-\gamma}.$$

With $n = 5$, this corresponds to a return of $\approx 4.8$. This coincides with the values reported in Fig. 4 for PPO, indicating that PPO alone fails to address the confounding policy update in this example.

As mentioned in the main text, we want to highlight the representation learning capabilities of `DreamSPI`. For this reason, we provide a view of the grid in raw pixels to the agent (cf. Fig. 10). In our experiments, we choose $\epsilon = 0.2$. To evaluate PPO and `DeepSPI` in this environment, we use the default parameters from `cleanRL` (Huang et al., 2022), both for PPO and `DeepSPI`. However, we enforce for both algorithms a compact, small discrete representation with a limited capacity of 256 latent states (precisely, we use 4 categories of 4 classes, with the same latent representation as the one used by Hafner et al., 2021). For `DeepSPI`, we restrict the ratio to $1/\gamma - 1$, which leads to a neighborhood constant $C < 1/\gamma$, as the theory suggests. Fig. 4 reports the median of 10 independent runs/seeds per algorithm, as well as the interquartile range (25-75%). Note that the values $V^\pi(s_I)$ reported in Fig. 4 are computed analytically.

# H EXPERIMENTS: EVALUATION ON THE ATARI LEARNING ENVIRONMENTS

## H.1 IMPLEMENTATION

Our implementation can be found at `https://github.com/florentdelgrange/deepspi`.

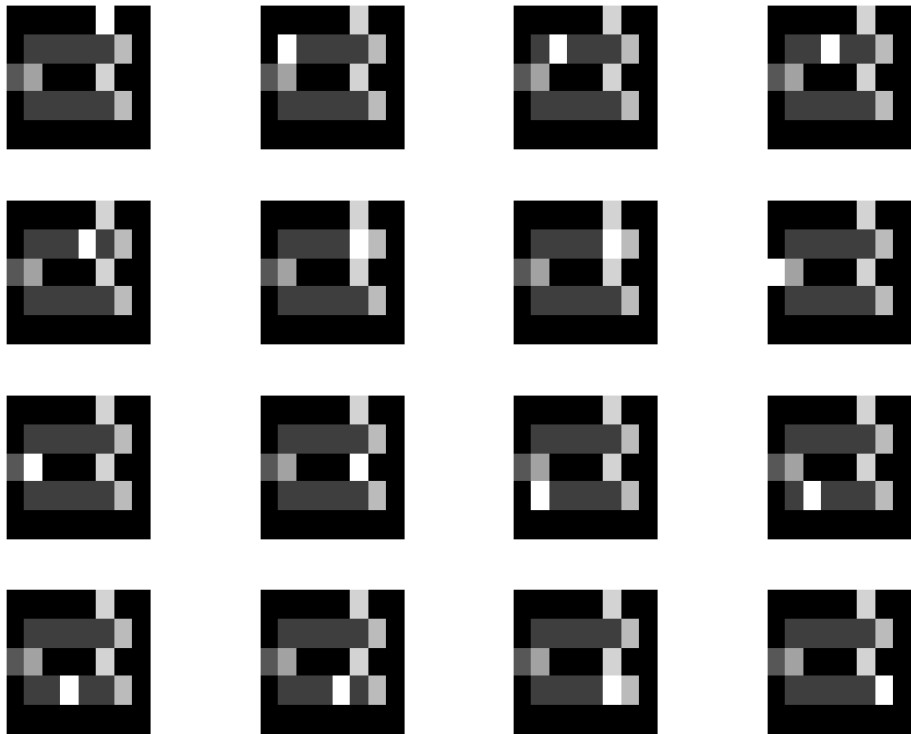

Figure 10: Observations of the grid perceived by the agent. The white-most cell corresponds to the agent's location in the grid. Each observation has size $84 \times 84$.

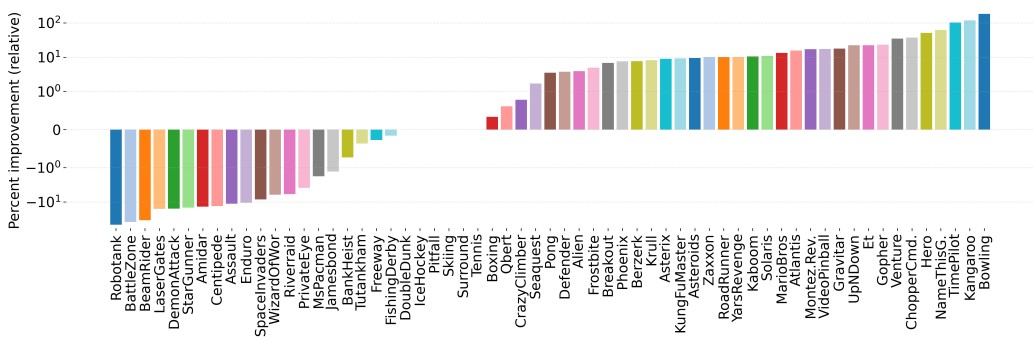

Figure 11: Relative improvement of `DeepSPI` compared to PPO over the full stochastic ALE suite (41/61).

## H.2 SETTING

Each presented experiment on the environments from ALE has been conducted across 8 seeds for each algorithm. Each run requires (mean $\pm$ std) $16.75 \pm 1.7$ min for PPO, $60.24 \pm 20.16$ min for `DeepMDP`, $62.49 \pm 20.71$ min for `DeepSPI`, and $80.8 \pm 1.35$ min for `DreamSPI` on an NVIDIA A40. This corresponds to a $\approx 3.6 \times$ overhead when using `DeepSPI` instead of PPO, which we consider a modest cost given the guarantees we obtain. Because our method is on-policy and fully parallelizable, the wall-clock time remains well below that of off-policy approaches that do not exploit vectorized environments. For comparison, SAC requires roughly 40 hours for the same number of collected frames on an NVIDIA A100 (Huang et al., 2022), and `Dreamer-v2` needs about two days on NVIDIA V100 (Hafner et al., 2021).

## H.3 Additional Plots

In this section, we present additional figures to highlight statistics and the performance of our algorithm, `DeepSPI`. Fig. 11 presents the relative improvement of `DeepSPI` w.r.t. PPO (Fig. 11). We formally compute the *relative improvement* as

$$\frac{score - score_{baseline}}{|score_{baseline}|}$$

and we use the maximum median human normalized score as the metric to compare in each environment. See next page for a comparison of each of the algorithms (average episodic return) per environment, across training steps. We report the median and interquartile range (25–75%) for each environment. Recall that one training step corresponds to gathering four Atari frames in the environment.

**Transition and reward losses.** In the main text, we stated that the transition loss achieved by `DeepSPI` is, in general, lower than for `DeepMDPs`. We elaborate here on the statistical significance of this claim. First, as for the human normalized score, we provide in Fig. 12 aggregate metrics for the transition and reward losses. This analysis already reveals that there is no statistically significant difference between the capacity to predict rewards between the two algorithms. We take a closer look at the transition loss.

For each environment $i$, we summarize the transition loss of `DeepSPI` and `DeepMDP` by scalars $\ell_i^{\text{SPI}}$ and $\ell_i^{\text{MDP}}$, and form paired differences $d_i = \ell_i^{\text{SPI}} - \ell_i^{\text{MDP}}$. The reported mean difference $\bar{d} = \frac{1}{n}\sum_i d_i = -0.1381$ therefore means that, on average across environments, `DeepSPI`'s transition loss is about 0.14 units lower than `DeepMDP`'s. To quantify uncertainty on this average effect, we use a paired bootstrap: we resample the $n$ environments with replacement, recompute $\bar{d}^{(b)}$ for each bootstrap sample $b = 1, \dots, B$, and form the 95% confidence interval as the 2.5th and 97.5th percentiles of $\{\bar{d}^{(b)}\}_{b=1}^{B}$. The resulting interval $[-0.2226, -0.05907]$ lies entirely below zero, which under the usual frequentist interpretation provides strong evidence that the true mean gap in transition loss is negative (`DeepSPI` better) rather than a consequence of sampling noise.

The paired Wilcoxon signed-rank test (Wilcoxon, 1992) further supports this conclusion without invoking normality of the $d_i$: it ranks the absolute differences $|d_i|$, assigns each rank the sign of $d_i$, and uses the signed rank sum as a test statistic for the null hypothesis $H_0 : \text{median}(d_i) = 0$. We obtain a very small two-sided $p$-value $p = 6.6 \times 10^{-4}$ indicating that observing differences this systematically negative would be extremely unlikely if `DeepSPI` and `DeepMDP` had the same typical transition loss.

Finally, the aggregates of Fig. 12 provide a complementary robust view: the interquartile mean (IQM) of transition loss is lower for `DeepSPI` than for `DeepMDP`, indicating that `DeepSPI` improves not only the mean performance but also the performance on the central bulk of environments. Taken together, the negative mean difference with a 95% confidence interval that excludes zero, the significant Wilcoxon test, and the lower IQM all consistently indicate that `DeepSPI` achieves statistically significantly lower transition loss than `DeepMDP` across Atari.

## H.4 Hyperparameters

As mentioned in the main text, we use the same parameters for PPO as the default `cleanRL`'s parameters. We list the `DeepSPI` parameters in Table 1 and those of `DreamSPI` in Table 2. We used the same parameters as `DeepSPI` for `DeepMDPs`. For `DeepSPI`, we performed a grid search for the transition density in {IndependentNormal, MixtureIndependentNormal($n = 5$), Categorical(n_cat $= 32$, n_cls $= 32$)}. The grid search revealed that the mixture of independent normal distributions (i.e., with diagonal covariance matrices) worked best for `DeepSPI`. We also found that using Lipschitz networks to enforce the Lipschitzness of the latent space (cf. Sect. 5) was faster than enforcing a gradient penalty (as used by Gelada et al. 2019) since, in contrast to gradient penalties, enforcing a Lipschitz condition through the architecture does not require additional sampling from the latent transition function (which might turn out costly, especially with mixture distributions). Furthermore, norm-constrained GroupSort architectures ensure Lipschitzness by construction. For the reward and transition coefficients, we performed a grid search in $\alpha_R, \alpha_P \in \left\{10^{-2}, 5 \times 10^{-3}, 10^{-3}, 5 \times 10^{-4}, 10^{-4}\right\}$. We found the best performance at $\alpha_R = 0.01$ and $\alpha_P = 5 \times 10^{-4}$.

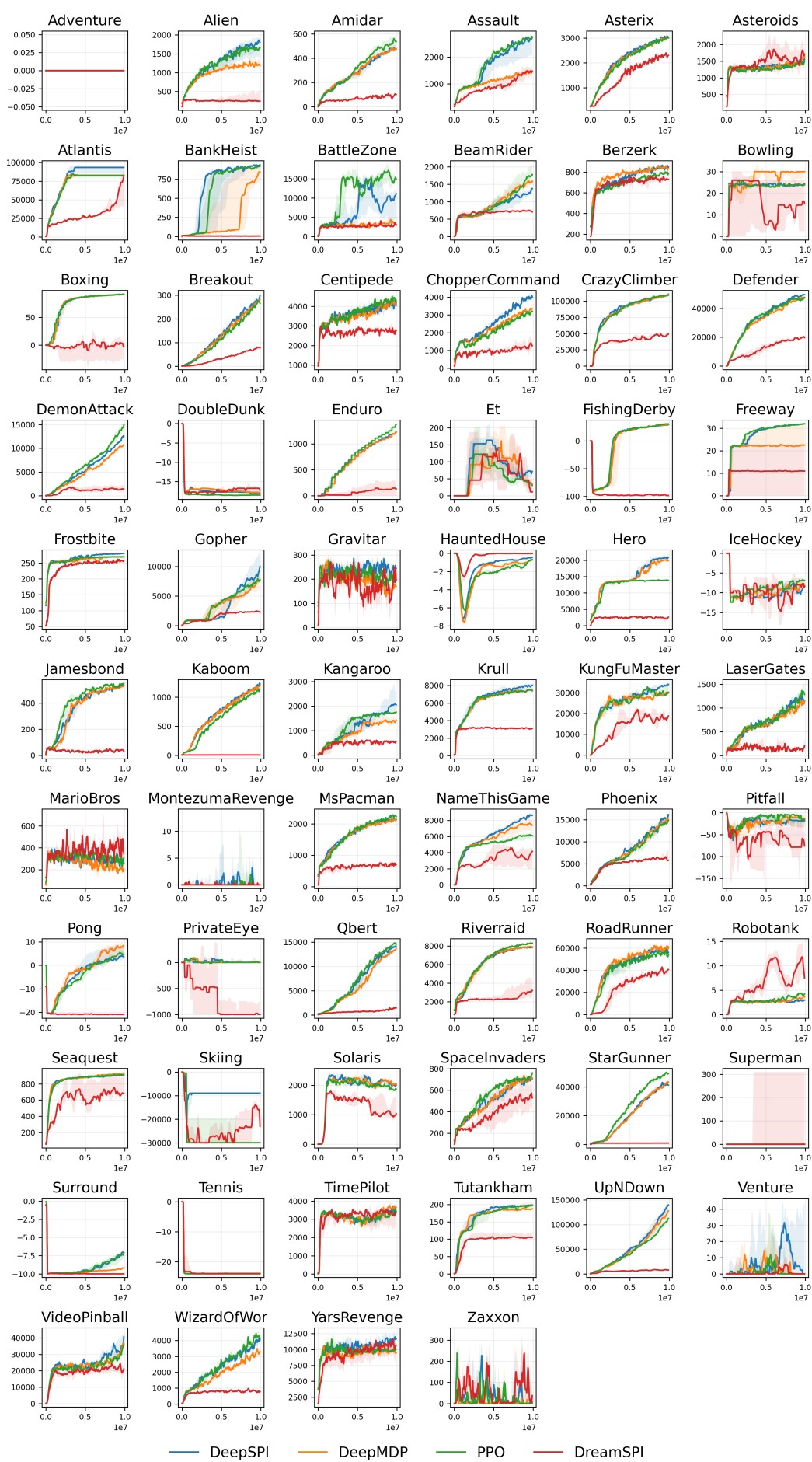

| Hyperparameter | Value |
| --- | --- |
| Learning rate | $2.5 \times 10^{-4}$ |
| Number of envs | 128 |
| Number of rollout steps | 8 |
| LR annealing | True |
| Activation function | ReLU |
| Discount factor $\gamma$ | 0.99 |
| GAE $\lambda$ | 0.95 |
| Number of minibatches | 4 |
| Update epochs | 4 |
| Advantage normalization | True |
| Clipping coefficient $\epsilon$ | 0.1 |
| Entropy coefficient | 0.01 |
| Value loss coefficient | 0.5 |
| Max gradient norm | 0.5 |
| Transition loss coefficient ($\alpha_P$) | $5 \times 10^{-4}$ |
| Reward loss coefficient ($\alpha_R$) | 0.01 |
| Transition density | Mixture of Normal (diagonal covariance matrix) |
| Number of distributions | 5 |
| Lipschitz networks | True |

Table 1: Summary of `DeepSPI` hyperparameters.

| Hyperparameter | Value |
| --- | --- |
| Imagination horizon | 8 |
| actor/critic update epochs | 1 |
| actor/critic number of minibatches | $4 \times 8 = 32$ |
| Discount factor $\gamma$ | 0.995 |
| Encoder learning rate | $2 \times 10^{-4}$ |
| Actor learning rate | $2.75 \times 10^{-5}$ |
| Critic learning rate | $2.75 \times 10^{-5}$ |
| World model learning rate | $2 \times 10^{-4}$ |
| Global LR annealing | False |
| Weight decay (`AdamW`) | True; with decay $10^{-6}$ |
| Transition density | Categorical (32 categories of 32 classes, see Hafner et al., 2021) |
| Transition loss coefficient ($\alpha_P$) | 0.01 |
| Reward loss coefficient ($\alpha_R$) | 0.01 |
| Lipschitz networks | False (unnecessary with discrete random variables) |
| Other parameters | Same as `DeepSPI` |

Table 2: Summary of `DreamSPI` hyperparameters.

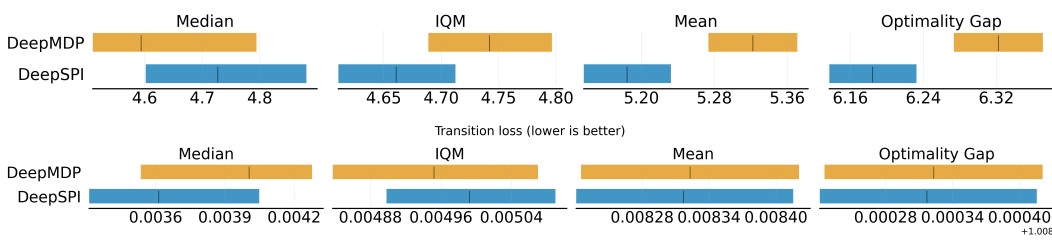

Figure 12: Aggregate median, IQR, Mean, and optimality gap for the reported transition and reward losses over all the Atari environments considered in our experiments, with $95\%$ confidence intervals. The confidence intervals are obtained via percentile bootstrapping with stratified resampling. For more information, refer to Agarwal et al., 2021b.

