# OpenReview forum: "Deep SPI: Safe Policy Improvement via World Models"
_ICLR.cc/2026/Conference — ICLR 2026 Poster_

### Official Review · Reviewer_2xGk · 2025-10-19

**Soundness:** 2
**Presentation:** 4
**Contribution:** 3
**Rating:** 4
**Confidence:** 3

**Summary:**

This paper presents theoretical bounds on expected return degeneration (or values) as a consequence of representation learning composed with policy updates.
The authors theoretically show that making controlled (regularized) updates reduces out-of-trajectory misgeneralization during learning.
Secondly, they show that including the world model losses as part of the returns (aka an auxiliary reward/ objective) reduces the policy confounding problem.
Finally, an extension to PPO with auxiliary losses was implemented to validate the authors theory, which showed marginal performance increase on the ALE benchmark.

**Strengths:**

- Strong mathematical clarity, the background is compact yet comprehensive, and I could understand all nomenclature from this point on. The authors are precise without impeding reading flow.

- Clear communication and examples of model learning issues and confounding in section 3.

- Overall excellent discussion of theory. Almost every theoretical result is followed by a simplifying explanation that discusses its impact and how it will be used later.

- Results are linked to the claims proposed in the introduction.

- Simple and intuitive modification of the PPO algorithm in Section 6 that is decently motivated from the preceding theory.

- Humorous Spiderman references, although reducing the section-title fontsize in section 3 to fit on one line, is probably not allowed.

**Weaknesses:**

- I find the use of the sensitive word "safe" quite imprudent and unfitting. While we can endlessly debate the concept of safety, and the authors are probably not to blame (but prior work is), as far as I understand the paper is not dealing with cost constraints or risk aversion. Instead, "safe" refers to not "destroying" the world model due to policy updates during learning. Considering that there exist actual risk-sensitive RL applications (e.g., in finance or healthcare), I believe the authors should advertise their method in a different way. How about a word play on "conservative-world-model policy improvement" or "unconfounding regularization"? Which is more accurate in my opinion to the fix presented in Eq.4 \& Eq.5.

- Although section 3.2 presents a clear example, as far I understand this idea lies extremely close to well known prior work that should be cited. Specifically, the $Q^\pi$-irrelevance abstractions. See, Li L. et al. (2006). *Towards a Unified Theory of State Abstraction for MDPs.*

- The after-discussion of Theorem 2 should more strongly state the limitation that the result cannot give guarantees for large discount factors $\gamma \rightarrow 1$. This is ok, and aligns with intuition. The last 2 sentences of lines 301-308 already sort-of discusses this, but I think it should be even more explicit.

- The experimental section is weak and I'm not able to gather much from it, other than confirming that "the authors' method does not break PPO". Performance is not marginally improved or decreased, and varies from environment to environment. So, although the setup is decent, the problem is that we do not learn much from these experiments.
	- I am missing a more didactic experiment that compares the authors' method to a baseline method to validate that they "fix" the OOT and confounding policy update issues that they claim to solve. Instead the experiments go straight into performance and loss curves. Although useful to know, they are not that meaningful in the papers' context.
	- Furthermore, I miss a stronger connection from the theory to e.g., parameter choices for say $\epsilon$ (or $C$) that provides some approximate guarantee of modulating the permitted errors derived in the previous sections. Could a more TRPO-like algorithm be derived from the authors' theory, that explicitly tries to satisfy constrained updates? Although the authors decently build up to the PPO extension, I find the auxiliary loss regularization to read somewhat like an afterthought that undermines the interesting preceding results.
	- The authors claim in paragraph 448-458 that their transition loss is improved, but looking at figure 5 this is not statistically significant compared to the DeepMDP baseline.

- Conclusion line 479-481 overstates the result that DeepSPI improves upon PPO, the performance benefit is quite marginal and more noisy.

 *Minor comments:*
- line 148, the "baseline policy" is known in RL as the behavior policy. I don't see a reason to ignore convention here.
- Appendix page 30, figure is too large, the label overlaps with the page number.
- Line 286-288, can you put this paragraph into an "assumption" environment and clearly state that you're working with this "from this point on". This detail is now buried within the surrounding text, and it's not clear whether the assumption belongs to the later results, or to all results.


*My overall verdict:*
This is an extremely clearly written document with theoretical results that relatively nicely align with intuition.
The authors carefully guide the reader through the theory and explain the impact of their result without obfuscating their result with unnecessary detail.
That said, I am not a pure theoretical person in the RL field, so I cannot accurately judge the impact or novelty of the proposed contributions over prior work.
My main problem with the paper at the moment is the weak experiment section, I want to see a setting where the authors can validate their theoretical result, even if it is simple. I think the PPO extension does not do the rest of the paper justice.
Overall, I would be leaning towards acceptance if most of my points can be addressed and questions answered.

**Questions:**

1. I don't understand part of the claim on the support for the policy sequence $(\pi_n)_{n\ge 0}$. Why does $\pi_n, n \rightarrow \infty$ guarantee full support?Theorem 1 claims that the sequence converges to an optimal value, and I can understand this yields a (stochastically) optimal policy. However in lines 234-239 it is stated that since $\pi_0$ has full support, all $\pi_n$ have full support as well, which again guarantees full support over the full state space.
But if $\pi_n \rightarrow \pi^* $ as $n \rightarrow \infty$, then $ \pi^* $ should not have full support. The optimal policy should put zero density/ probability on suboptimal actions. If it doesn't, then $\pi_n$ should not converge such that $V^{\pi_n} = V^*$.

2. Could you shortly reflect on the theory and practical impact of your answer to my Q1? How would you revise your claims/ theorem?

3. Can you add equation numbering to the reward-transition loss of lines 257-259, and then refer to this Eq. again when discussing Theorem 2. It took me a moment to see that $L_R^\xi, L_P^\xi$ referred to these losses over the state-occupancy instead of the buffer $\mathcal{B}$.

4. Could the authors think of (and run) a didactic experiment that validates the newly proposed algorithm. This could perhaps be built on the toy MDPs from figure 1 or figure 2, at the moment, it is not clear experimentally whether the authors proposed method

---

> ### Author Response · Authors · 2025-11-21
>
> We thank the reviewer for the very positive comments on the clarity and structure of the paper, as well as the careful reading of the theory. We address the points raised below.
>
> # Weakness #1 (use of the term “safe”)
> We fully agree that “safe” is used for different purposes in RL. Here, we use the term exactly as in the **safe policy improvement** literature, where “safe” means that policy updates do not catastrophically degrade performance. Appendix D surveys this field. We chose this terminology to remain consistent with prior work, not to suggest risk-sensitive RL. We will add an explicit remark to make this clear.
>
> # Weakness #2 (relation to abstraction/representation papers).
> Thank you for the reference. The cited paper indeed studies ideal abstraction properties in finite state-action spaces with fixed representations. DeepSPI aims at approaching such properties in **general on-policy settings**, where representation learning and policy updates interact. Our work is, in fact, closer to **bisimulation theory** [3, 4], which the cited paper also connects to. Their “malicious example” for Q-learning is very close in spirit to our example on confounding policy updates. We will add this to the related work.
>
> # Weakness #3 (role of $\gamma$).
> To clarify: **Theorem 2 does not require a small $\gamma$.** The result holds for usual high values of $\gamma$. What changes is that large $\gamma$ implies a longer effective horizon, which naturally makes the neighbourhood constraints stricter. We already mentioned this after Theorem 2, and we will make this explanation more explicit in the final version.
>
> # Weakness #4 (experiment strength, TRPO, and breadth).
> We acknowledge the reviewer’s concern. At the same time, our evaluation already spans **61 environments** with widely varying dynamics (see Fig. 3), and we refer again to the discussion of ALE in [1] as a suitable testbed for studying RL behaviour beyond raw performance. We will add a deeper statistical analysis to the paper. We are also currently considering adding a small didactic experiment, as suggested.
>
> On TRPO: the reviewer’s remark is fair. In practice, though, TRPO has been replaced by PPO due to its efficiency. Importantly, as shown in [2], **PPO is not just a heuristic** but **a rigorous instance of mirror learning**, which ensures monotonic improvement and convergence under some assumptions. This motivates our choice of PPO as the basis for DeepSPI.
>
> # Weakness #5 (statistical tests).
> We are currently running experiments to gather additional runs and perform further statistical analyses of our results.
>
> ## Minor comments
> - We will rename “baseline policy” (SPI terminology) to the standard “behaviour policy,” which is equivalent in our on-policy setting.
> - We will fix the figure formatting issue.
> - We will add an explicit assumption environment where appropriate.
>
> # Question #1 (full support and convergence).
> In general MDPs, optimal policies need not be unique, nor necessarily deterministic (we know that at least one deterministic policy exists). Our neighbourhood operator ensures that every policy in the sequence has full support (Equation 2), which is required for the theoretical convergence argument. Theorem 1 concerns the limit of the sequence: as $n \to \infty$, the policies converge to an optimal value function. However, **convergence does not imply that finite-step policies must match a specific optimal policy**. As an analogy, consider a policy with two actions whose respective probabilities are $\left(a_1 \mapsto \frac{1}{n},\; a_2 \mapsto 1 - \frac{1}{n} \right)$; it has full support at every finite step even though **its limit** collapses to a deterministic policy (that will always choose $a_2$). We will clarify this point in the paper.
>
> # Other Questions
> ## Question #2
> Thank you for flagging this potential misunderstanding; we will make this clearer in the final version.
>
> ## Question #3
> Sure, we will make this modification.
>
> ## Question #4
> As mentioned, we are currently considering adding such an example.
>
> ## References
>
> [1] P.S. Castro: The Formalism-Implementation Gap in Reinforcement Learning Research, arXiv 2025 \
> [2] J.G. Kuba et al.: Mirror Learning: A Unifying Framework of Policy Optimisation. ICML 2022 \
> [3] K.G. Larsen, A. Skou: Bisimulation Through Probabilistic Testing. POPL 1989 \
> [4] R. Givan et al.: Equivalence notions and model minimization in Markov decision processes. Artif. Intell. 147(1-2), 2003

---

> ### Comment · Reviewer_2xGk · 2025-11-21
>
> Hi I appreciate the extensive rebuttal. I'm still reading and thinking about your answers to the other reviewers, but had a quick remark on your answer to theorem 2. Specifically the $ ... \cdot \frac{...}{\frac{1}{D^{sup}_{IR}(\pi_b, \bar{\pi})} - \gamma}$
>
> If $\gamma = 1$, then don't you get a division by zero if $\pi_b = \bar{\pi}$. I also believe this can only happen in this exact scenario, and that $\gamma < 1$ or $\pi_b \ne \bar{\pi}$ solves this. But I might have misinterpreted what $D_{IR}^{sup}(\cdot, \cdot)$ does.

---

> ### Author Response · Authors · 2025-11-21
> **Thanks for the prompt response**
>
> Hi, thank you for taking the time to thoroughly check our responses. This is very much appreciated. Indeed, you are fully correct, we strictly focus on the discounted case ($\gamma < 1$), which is the standard expected total discounted reward criterion [1] and the main paradigm considered in RL. The undiscounted case is quite tricky and needs strong additional assumptions that are not straightforward, especially in general spaces (again, refer to [1, Chapter 7] for a discussion and [2]; for an analysis specific to RL, refer to [3]).
>
> We precisely stated at line 88 that we are always considering a discount strictly lower than one. If the space permits it, we will recall this in the main text, from the point where we present the theorems to avoid any confusion.
> Let's now take a closer look at $ratio := {D^{\\sup}\_{IR}(\pi_b, \bar{\pi})}$ and whether something could go wrong here. First of all, notice that we do allow $\pi_b$ to exactly match $\bar{\pi}$ ($\pi_b$ always belongs to its own neighborhood). So we know that $ratio$ is lower-bounded by $D^{\\sup}\_{IR}(\pi_b, \bar{\pi}) = 1$ (since we are dealing with distributions and a supremum). On the other hand, we assumed that $1 < C < \frac{1}{\gamma}$. So we know that $ratio$ is **strictly** upper bounded by $\frac{1}{\gamma}$. Thus, $\gamma < {1}/{ratio} \leq 1$. This also means that in every possible case, $1/ratio - \gamma$ is strictly positive. Then, the denominator of the bound presented in Theorem 2 is always well-defined.
>
> ## References
> [1] M. L. Puterman: Markov Decision Processes: Discrete Stochastic Dynamic Programming (1st. ed.). John Wiley & Sons, Inc., 1994 \
> [2] Dimitri P. Bertsekas, John N. Tsitsiklis: An Analysis of Stochastic Shortest Path Problems. Math. Oper. Res. 16(3): 580-595, 1991 \
> [3] J.N. Tsitsiklis: Asynchronous Stochastic Approximation and Q-Learning. Machine Learning 16, 185–202, 1994

---

> > ### Comment · Reviewer_2xGk · 2025-11-24
> >
> > Thanks to the authors for the clarification on theorem 2, I am satisfied with their answers to my questions. I eagerly await the more didactic experiment that is promised. I increased my score but am inclined to raise it further if all points are adressed in a revision of the PDF.
> >
> > In regards to the experimental section, I understand the authors' point and want to remark that perhaps it is not the quality of the experimental design that is an issue, but the way that it is presented.
> >
> > Instead of presenting the results like in figure 3, could the authors consider grouping the environments according to the characteristics of each problem? Similar as is done in behavior-suite or as suggested by [1]. For example, Montezuma and Pitfall in "sparse reward" and Breakout in "partially observable" + "dense reward". Something like that.
> >
> > This would make it easier to see what one could expect from the authors method and also validate their claim made to Reviewer q5uR, that the proposed method is more robust (safe) under stochasticity. I would imagine that being more conservative/ safe in stochastic MDP settings might lead to a trade-off that gives slower learning on deterministic environments. This is completely ok, but difficult to extract from how the results are currently presented.
> >
> > ---
> >
> > [1] Andrew Patterson, Samuel Neumann, Martha White, Adam White; 25(318):1−63, 2024.

---

> > > ### Author Response · Authors · 2025-11-27
> > >
> > > Thank you again for your careful reading and for already increasing your score. As promised, *we have now added a didactic experiment* (Section 6.1, with further details in Appendix G).
> > >
> > > We also fully acknowledge your concerns about the presentation of the experimental results and the absence of additional statistical evidence. To address this, we followed the recommendations of [1]: instead of relying on point estimates that ignore statistical uncertainty, we now report interval estimates (95% confidence intervals) obtained via stratified bootstrap across the entire Atari benchmark suite. As aggregate metrics, we provide mean, median, interquartile mean (IQM), and optimality gap. IQM yields tighter confidence intervals than median, while the optimality gap is more robust to outliers than the mean. We also include sample-efficiency curves based on IQM human-normalized scores, again with 95% confidence intervals computed via percentile stratified bootstrap. Moreover, we increased the number of seeds to 8 for all algorithms across the full benchmark. We also added a statistical analysis, as you requested, in Appendix H.2, about our statement on the improvement in the transition loss.
> > >
> > > In addition, we strengthened the related work section with your recommendations and incorporated all the modifications you requested. The paper was further updated to address the comments from the other reviewers as well (see the complete changelog in our general response).
> > >
> > > Overall, we believe the revised version presents our contributions more clearly and rigorously. We appreciate your suggestions, which have helped us improve the paper substantially. We hope you will be satisfied with the changes we have made to the paper.
> > >
> > > [1] Rishabh Agarwal, Max Schwarzer, Pablo Samuel Castro, Aaron C. Courville, Marc G. Bellemare: Deep Reinforcement Learning at the Edge of the Statistical Precipice. NeurIPS 2021: 29304-29320

---

> > > > ### Comment · Reviewer_2xGk · 2025-11-27
> > > >
> > > > Dear authors, thanks for adressing my points in your revision and for running additional experiments and repetitions, I have some final notes:
> > > >
> > > > - Line 78/79, "Building on bisumulation, works design represenation that cluster..." This sentence reads broken, I suggest revising it. Something like "Bisumulation methods learn representations that cluster states into groups where the agent is guaranteed" ?
> > > > - Appendix G, I think you forgot to forward declare what values $\epsilon$ and $\gamma$ take in your example calculation of "the good, the bad, and the ugly" values :). I seem to infer it is $\epsilon=0.2$ and $\gamma=0.99$ from crosschecking, but it will help reading to define it earlier.
> > > > - I think the roadmap in 210 is a strong addition, it makes your contributions even more clear.
> > > >
> > > > Also, the illustrative example is excellent, very clear. Could you shortly reflect on why PPO is not able to achieve 8.1 return? I assume it is because the default PPO loss make it less likely to find "the good" representation by chance and fits too quickly on "the ugly" representations from which it doesn't recover... Whereas your method makes uncovering this representation much more likely?
> > > >
> > > > All in all, this is an excellent paper. The presentation is well structured, clear, and coherent. It has a nice mix between problem definition (building on recent related work), theory, and experimental work. So, in my opinion it deserves a nomination for spotlight or oral.

---

> > > > > ### Author Response · Authors · 2025-11-28
> > > > >
> > > > > Dear Reviewer,
> > > > >
> > > > > Thank you very much for updating your score! We are grateful for your positive assessment of the paper.
> > > > >
> > > > > - We will modify this sentence in the related work section accordingly.
> > > > > - For Appendix G: yes, indeed; the values are $\epsilon=0.2$ and $\gamma=0.99$. Thanks for catching this. We will update those details in Appendix G.
> > > > > - About the illustrative example: your intuition is correct. The behaviour of PPO is due to the representation collapse shown in Figure 4 (left). Unlike DeepSPI, which explicitly regularizes the representation to leverage the theoretical guarantees of Theorem 4, PPO lacks a mechanism that prevents such collapse. Because of the environment’s construction, PPO naturally tends to merge the two $\star$ cells in the latent space, which limits the best policy it can learn to "the ugly." In contrast, DeepSPI’s policy improvement mechanism ensures that the two $\star$ cells are treated as distinct in the representation by detecting potential behavioral changes within the behavioral policy neighborhood.

---

### Official Review · Reviewer_9StS · 2025-10-31

**Soundness:** 3
**Presentation:** 3
**Contribution:** 3
**Rating:** 4
**Confidence:** 4

**Summary:**

The paper proposes DeepSPI, a safe policy improvement algorithm that combines mirror learning updates with a learned world model. The work provides theoretical results that jointly consider the impacts of policy updates and world model updates, which motivates an extension of PPO that incorporates a world model for representation learning. Experiments across the Atari benchmark demonstrate improved performance compared to PPO and DeepMDP.

**Strengths:**

**[S1]**: The paper is clear, well-written, and easy to follow.

**[S2]**: The paper introduces interesting theoretical analysis that combines mirror learning policy updates with world model representation learning. This analysis extends results from DeepMDP [1] to consider the interplay between policy updates and representation learning, and results in bounds with less restrictive assumptions.

**References:**

[1] Gelada et al., “DeepMDP: Learning Continuous Latent Space Models for Representation Learning.” In ICML 2019.

**Weaknesses:**

**[W1]**: Experimental results do not demonstrate convincing practical benefits of the proposed approach. The experiments show modest improvements for DeepSPI compared to PPO and DeepMDP, and the performance of DeepSPI is significantly worse than other results that have been reported in the literature on the Atari benchmark. DreamerV3 [2], for example, is also an actor-critic algorithm that leverages a learned world model, and reports better results after only 100k steps (Atari100k results in [2]) compared to the performance of DeepSPI after 10M steps.

**[W2]**: It seems like the learned world model is under-utilized in DeepSPI, only being used for representation learning but not for model-based rollouts as in the Dreamer series. The attempt to consider a model-based implementation using imaginary rollouts (DreamSPI) does not perform well, which the authors suggest may be due to its on-policy nature. It is not clear if the performance guarantees can be extended to account for model-based training with imaginary rollouts.

**References:**

[2] Hafner et al., “Mastering Diverse Domains through World Models.” arXiv, 2023. arXiv:2301.04104.

**Questions:**

**[Q1]**: Are the theoretical results compatible with model-based RL using imaginary rollouts as in the Dreamer series? The experiments suggest that the use of an on-policy approach without imaginary rollouts cannot achieve performance close to state-of-the-art algorithms like DreamerV3, which limits the practical impact of the work.

**[Q2]**: Are the theoretical results compatible with off-policy RL methods?

**[Q3]**: How does the approach relate to conservative model-based methods in offline RL (e.g., [3])?

**[Q4]**: Please provide additional implementation details, such as network architectures and the computation time of DeepSPI relative to PPO.


**References:**

[3] Kidambi et al., “MOReL: Model-Based Offline Reinforcement Learning.” In NeurIPS 2020.

---

> ### Author Response · Authors · 2025-11-21
>
> We thank the reviewer for the positive assessment of the clarity, the theoretical contributions, and the integration of representation learning with mirror-learning updates. We appreciate the careful reading and summarise our responses below.
>
> # Weakness #1
> As mentioned in our general answer **G1**, while some environments show comparable performance to PPO, Figure 3 demonstrates that DeepSPI brings **large improvements** in many games, even exceeding **100%** improvement for some of them. These gains are tied to the stronger representation learning induced by our auxiliary losses.
>
> Importantly, **DeepSPI achieves PPO-level performance** while **providing SPI-type guarantees**, which is not something PPO or model-free baselines offer. As noted in the paper, these guarantees are orthogonal to pure empirical performance and remain valuable for understanding how RL behaves under representation learning.
>
> Regarding Dreamer-V3: as mentioned in our responses to other reviewers, the comparison is not direct. Dreamer-V3 is **off-policy**, trained from a replay buffer, and evaluated under a different regime (lower stochasticity, no parallel environments), and uses a large number of engineering components (cf. general answer **G2**). The experimental regimes are therefore not comparable.
>
> # Weakness #2
> We agree that in DeepSPI the learned model is used primarily for representation learning and to **regularise policy updates**. This design is consistent with prior work such as [1, 2, 3]. These methods also focus on latent-space quality rather than full imagination-based rollouts, and have been considered meaningful contributions.
>
> A full Dreamer-like algorithm combined with DeepSPI-style guarantees would indeed be very interesting. This is, however, a **separate research direction** and would require substantial engineering to stabilise imagination-based training in a strictly on-policy setting. Our **naive DreamSPI prototype** already shows some promising signs in games like _Asteroids, Robotank, and Mario Bros_, indicating that the interplay between SPI regularisation and latent imagination may be beneficial. We will highlight this more clearly in the paper.
>
> Note that Dreamer **is not on-policy**: it is trained entirely from replay and imagination, and does not satisfy any SPI-type properties. This is orthogonal to the setting we study.
>
> # Question #1
> Yes, the theoretical results are compatible. In practice, our **on-policy** algorithm is challenging because it needs to learn (i) the model on-policy, and (ii) a policy inside the model via latent imagination. Our prototype deliberately avoids the many tricks used in Dreamer (general answer **G1**), since the aim was only to test whether the simple structure from DeepSPI already enables imagination-based training.
> Although unstable, some environments highlight improvements and different behaviors than the other baselines. This suggests that a more dedicated approach – combining SPI insights with carefully engineered world-model architectures – may yield meaningful advances. We see this as an exciting direction for future work.
>
> # Question #2
> The theory is not _directly_ compatible with off-policy RL, but off-policy algorithms can be adapted to make it applicable: off-policy PPO variant exists [4], and off-policy mirror learning is valid when using the appropriate **importance-sampling corrections** [5, Appendix F]. However, adopting these techniques would sacrifice the main advantage of **vectorized on-policy environments**, which enable large-scale experiments with massive parallelism (cf. general answer **G2**; off-policy methods might be up to 48x slower).
>
> # Question #3 (relation to MOReL and conservative model-based offline RL).
> Thank you for the pointer. MOReL shares the aim of preventing harmful updates but differs fundamentally from our setting. It is purely offline, operating on a fixed dataset, and its guarantees work directly in the observed state space, without representation learning. In contrast, DeepSPI studies how latent representations interact with on-policy updates.
> MOReL also relies on assumptions that do not appear in our framework (e.g., support and occupancy conditions, hitting-time bounds). DeepSPI instead uses a neighbourhood operator defined through importance ratios, which is the natural tool for online mirror-learning updates.
>
> The connection is interesting, and we will add this line of work to the related work.
>
> # Question #4 (architectural details).
> The policy and value networks follow **exactly** the PPO architecture from CleanRL. The only additions are the **transition and reward networks**, for which we use the architecture from DeepMDPs [1], improved with norm-constrained GroupSort layers (cf. general answer **G3**), giving us Lipschitz control without extra gradient penalties. Since we must handle stochastic transitions, we predict a mixture of 5 diagonal Gaussians. We will make that clearer in the main text.

---

> ### Author Response · Authors · 2025-11-21
> **References**
>
> [1] C. Gelada et al. DeepMDP: Learning Continuous Latent Space Models for Representation Learning, ICML 2019\
> [2] A. Zhang: Learning Invariant Representations for Reinforcement Learning without Reconstruction. ICLR 2021\
> [3] R. Avalos et al.: The Wasserstein Believer: Learning Belief Updates for Partially Observable Environments through Reliable Latent Space Models. ICLR 2024\
> [4] W. Meng et al.: Off-Policy Proximal Policy Optimization. AAAI 2023 \
> [5] J.G. Kuba et al.: Mirror Learning: A Unifying Framework of Policy Optimisation. ICML 2022

---

> > ### Author Response · Authors · 2025-11-27
> > **Paper revision according to your comment**
> >
> > Dear reviewer, we have revised the paper according to your recommendations. The full changelog is available in our general response.
> >
> > First, to address your concerns regarding the experimental results and the practical relevance of our approach, we strengthened the empirical analysis with additional statistical evaluations. We increased the number of seeds to 8 across the full Atari benchmark and added figures showing aggregated performance over all tasks. We now report mean, median, interquartile mean (IQM), and optimality gap, each with 95% confidence intervals computed using the percentile
> > bootstrap with stratified sampling, following the recommendations of [6]. We also report sample-efficiency curves as a function of environment interaction steps. In addition, Appendix H.2 now includes a statistical analysis of the transition-loss improvements, showing that the observed gains are statistically significant.
> >
> > Second, as you suggested, we expanded the related-work section to include the line of research on conservative model-based improvements in offline RL.
> >
> > Finally, we added wall-clock time comparisons with PPO in Appendix H.1, as well as further architectural details in Section 7 and Appendix H.3.
> >
> > We hope these revisions fully address your remaining concerns.
> >
> > ## References
> > [6] Rishabh Agarwal, Max Schwarzer, Pablo Samuel Castro, Aaron C. Courville, Marc G. Bellemare: Deep Reinforcement Learning at the Edge of the Statistical Precipice. NeurIPS 2021: 29304-29320

---

### Official Review · Reviewer_Y6jS · 2025-11-01

**Soundness:** 3
**Presentation:** 3
**Contribution:** 2
**Rating:** 4
**Confidence:** 3

**Summary:**

This paper provides a strong theoretical contribution that generalizes safe policy improvement to deep, on-policy RL with learned world models. The integration of SPI principles, Wasserstein-based transition losses, and Lipschitz-continuous latent dynamics is mathematically interesting. The theoretical development is rigorous, connecting mirror descent formulations (Kuba et al., 2022) to practical PPO-style updates, and the proofs (Appendices B–E) seems to be technically comprehensive (although I did not check every detial). The empirical section, though limited to ALE benchmarks, is well-executed and demonstrates the feasibility of theory-grounded improvements.

However, some aspects could be clarified or strengthened: (i) practical relevance of the γ < 1/ C assumption, (ii) the sensitivity of DeepSPI to neighborhood size and Lipschitz constants, (iii) scalability to continuous control tasks, and (iv) the limited ablation on auxiliary losses.

**Strengths:**

1. The work extends classical SPI (Thomas et al., 2015; Laroche et al., 2019) to high-dimensional, online RL. The neighborhood operator $\mathcal{N}^{C}(\pi)$ (Eq. 2) defines a trust region via importance ratios, bridging SPI and policy regularization in a provably convergent way (Thm. 1).

2. The paper formalizes how transition/reward prediction losses ($L_{P}$, $L_{R}$) act as local regularizers ensuring Lipschitz-consistent latent dynamics (Thm. 2–4). This explicitly ties representation smoothness to policy safety, a novel link not addressed by DeepMDPs or Dreamer.

3. The identification of PPO as a special case of the proposed mirror-learning-based neighborhood update (Eq. 4 vs Eq. 3) is conceptually valuable. DeepSPI’s modified utility (Eq. 5) yields a principled way to blend safety and empirical efficiency.

4. On the ALE-57 suite, DeepSPI consistently outperforms PPO in 43 of 61 environments (Fig. 3), while reducing transition loss and ensuring smoother learning (Fig. 5). The implementation is reproducible and uses standard open frameworks (CleanRL, EnvPool).

**Weaknesses:**

1. Evaluations are restricted to Atari; no tests on continuous-control or partially observable tasks (e.g., DMControl, Procgen). It remains unclear whether Lipschitz constraints and local losses scale to these settings.

2. Although DeepMDP losses are included, the study omits other model-based SPI or regularized RL baselines (e.g., DreamerV3, SAC + KL). Statistical significance and variance across seeds are not discussed (e.g., arxiv.org/abs/1904.06979).

3. While claimed “on-policy,” DeepSPI introduces additional transition modeling networks. The paper lacks runtime, sample-efficiency, or wall-clock comparisons versus PPO, given the relatively limited improvement versus the increased algorithmic complexity.

4. This paper does noticeably lack a dedicated “proof roadmap” or overview section, which would guide the reader through the logical structure and dependencies of its multiple theorems (e.g., Theorems 1–5).

**Questions:**

The choice of neighborhood constant C and coefficients αₚ, αᵣ (Eq. 5) critically affects guarantees and performance, but no ablation or stability analysis is reported.

---

> ### Author Response · Authors · 2025-11-21
>
> We thank the reviewer for the positive evaluation of the theoretical development and the overall contribution. We appreciate the careful reading and the constructive suggestions. We address each point below.
>
> # Weakness #1 (continuous control, scalability, POMDPs).
> Lipschitzness of the latent transition function can be enforced either through **norm-constrained GroupSort architectures** [1] or obtained directly when working with **discrete latent spaces**, as in Dreamer-V2 [2], as in our own naive DreamSPI version. These mechanisms ensure that the assumptions required by our analysis remain valid beyond the Atari setting (this is thus also compliant with continuous settings).
>
> Concerning domain choice: as mentioned in our general answer **G2**, our paper focuses on environments in which **representation learning is critical**, and Atari is precisely such a regime. In continuous-control benchmarks like Mujoco, the underlying state variables (positions, velocities, etc.) are already *meaningful*, so the benefits of representation learning are far less pronounced. By contrast, learning a latent abstraction from raw image frames is essential in Atari, and this is where the interaction between policy updates and representation learning becomes most relevant. As highlighted in [3], ALE remains an important testbed precisely because of its diversity, its wide range of different dynamics, and nontrivial stochasticity [4]; this fits the goals of our work.
>
> Regarding POMDPs: our method is designed for Markovian environments, and therefore, benchmarking against POMDPs would not be appropriate. While some recent works (e.g., [5]) consider representation learning in POMDPs, extending SPI-type guarantees to POMDPs is mathematically much harder; standard results already show a complexity jump from MDPs (P-time) to POMDPs (PSPACE-hard or even undecidable in general). Such an extension would be a significant and interesting research direction, but it is not a minor variation of our current setting.
>
> # Weakness #2 (missing baselines, variance, statistical reporting).
> Our plots **already report medians and inter-quantile ranges**. We agree that additional statistical analysis could be added in the final version. _We are currently running experiments to collect additional seeds for this reason_.
> We intentionally restrict our experimental comparison to **on-policy** methods, because the theoretical framework we rely on (mirror learning with SPI-style constraints) does not directly apply to off-policy algorithms without additional importance-sampling corrections (see [6, Appendix F] for a detailed discussion). Off-policy methods such as SAC or Dreamer rely on replay buffers and update dynamics that differ fundamentally from our setting.
>
> As mentioned in our general answer **G2**, a second reason is computational: we explicitly target methods that scale with **vectorized environments**, which enable large-scale experiments across **61 Atari environments**. Dreamer and SAC benefit from replay buffers, but this slows down the learning process in terms of wall clock time (~1h per environment for DeepSPI vs. 1.25+ days for SAC/Dreamer-v2 for 10M steps). Reporting them as direct baselines would therefore be misleading: both their training dynamics and their evaluation cost are fundamentally different. We now clarify this motivation more explicitly.
>
> # Weakness #3 (implementation complexity and runtime).
> We will report wall-clock times for PPO, DeepSPI, and DeepMDP as requested in the Appendix. In practice, the runtime overhead of DeepSPI is about ×4 relative to vectorized PPO, which we believe is reasonable given the additional SPI-style properties and the introduction of a learned model. This is still far from the pace of off-policy methods.
>
> From an implementation standpoint, the changes from PPO are modest:
> 1. Algorithmically, we only modify the advantage by incorporating the auxiliary losses.
> 2. Architecturally, we add transition and reward networks.
> Enforcing Lipschitzness is handled directly by the norm-constrained GroupSort architecture [1], which is conceptually motivated but straightforward to implement.
>
> We also note that Reviewer 2xGk explicitly described the additional implementation effort as mild. The overhead is minimal compared to the implementation burden of the standard PPO itself (cf. [7]).
>
> # Weakness #4 (lack of proof roadmap).
> Thank you for this suggestion; we will add a clearer structural guide to the theoretical section in the final version.
>
> # Questions (constants and hyperparameters).
> For fairness, we use the same neighbourhood constant as cleanRL’s PPO [7], i.e., $C=1+\varepsilon$. For the auxiliary-loss coefficients​, we conducted a small hyperparameter search on a representative subset (Atari-5 [8]). We then applied the best-performing values across all 61 environments. These hyperparameters are consistent with those used in DeepMDPs. We will include the grid and selected values in the Appendix.

---

> > ### Author Response · Authors · 2025-11-21
> > **References**
> >
> > [1] C. Anil et al. Sorting Out Lipschitz Function Approximation. ICML 2019 \
> > [2] D. Hafner, et al.: Mastering Atari with Discrete World Models. ICLR 2021 \
> > [3] P.S. Castro: The Formalism-Implementation Gap in Reinforcement Learning Research, arXiv 2025 \
> > [4] C.M. Machado et al.: Revisiting the Arcade Learning Environment: Evaluation Protocols and Open Problems for General Agents. JAIR 2018 \
> > [5] R. Avalos et al.: The Wasserstein Believer: Learning Belief Updates for Partially Observable Environments through Reliable Latent Space Models. ICLR 2024 \
> > [6] J.G. Kuba et al.: Mirror Learning: A Unifying Framework of Policy Optimisation. ICML 2022 \
> > [7]  S. Huang et al.: CleanRL: High-quality Single-file Implementations of Deep Reinforcement Learning Algorithms, JLMR 2022 \
> > [8] M. Aitchison et al.: Atari-5: Distilling the Arcade Learning Environment down to Five Games. ICML 2023

---

> > > ### Author Response · Authors · 2025-11-27
> > > **Paper revision according to your comments**
> > >
> > > Dear reviewer, we have now implemented the changes you requested in our paper.
> > > The full list of changes made in our paper is provided in the changelog included in our general response.
> > >
> > > First, we added several additional figures and statistical analyses. We now report mean, median, interquartile mean (IQM), and optimality gap as aggregate metrics (new Figure 5), following the recommendations in [9], together with 95% stratified-bootstrap confidence intervals over the full ALE benchmark. We also increased the number of runs to 8 for all algorithms and environments. Moreover, Appendix H.2 discusses in detail the statistical significance of the transition-loss improvements.
> > >
> > > Second, we now include runtime measurements (Appendix H), an aggregate sample-efficiency curve (new Figure 6; with 95% percentile stratified-bootstrap confidence intervals), and wall-clock comparisons (Appendix H).
> > >
> > > Third, as recommended, we added a "proof roadmap" at the end of Section 3.
> > >
> > > We hope that our rebuttal and the corresponding changes address your concerns.
> > >
> > > ### References
> > > [9] Rishabh Agarwal, Max Schwarzer, Pablo Samuel Castro, Aaron C. Courville, Marc G. Bellemare: Deep Reinforcement Learning at the Edge of the Statistical Precipice. NeurIPS 2021: 29304-29320

---

> > > > ### Comment · Reviewer_Y6jS · 2025-11-28
> > > > **My concerns have been addressed**
> > > >
> > > > I'd love to increase my score to 6 and tend to accept this work.

---

> > > > > ### Author Response · Authors · 2025-11-28
> > > > >
> > > > > Dear reviewer,
> > > > >
> > > > > Thank you for your response. We appreciate your willingness to raise your score and are glad that our revisions addressed your concerns. Please feel free to reach out if any further clarification would be useful.
> > > > >
> > > > > (As a gentle reminder: if you do decide to update your score, please remember to do so by editing your original review.)

---

### Official Review · Reviewer_q5uR · 2025-11-11

**Soundness:** 4
**Presentation:** 3
**Contribution:** 3
**Rating:** 8
**Confidence:** 3

**Summary:**

The paper presents a method for safe policy improvement (SPI) for an online learning setting in combination with world models and representation learning. The authors offer proof that restricting the update of the policy gives guarantees of convergence. They also introduce auxiliary loss functions on the latent space ensure its stability (i.e. prevents the collapse of states in the representation). Afterwards experiments on the Atari benchmark show the performance improvement over PPO/DeepMDPs. The authors also present an investigation into the quality of the resulting world model.

**Strengths:**

- The problem addressed in the paper for online on-policy safe improvements is very relevant.
- The proofs and formulations presented in the paper appear correct/rigorous and the analysis done so far forms also a good basis for future works.
- The evaluations are well done over the full Atari suite, and include meaningful benchmarks, albeit all of them designed by the authors themselves.

**Weaknesses:**

- A lot of the background in section 2 lacks references for some of the very general statements the authors make about the field. E.g., L125-126 or L132-134. Similarly other mentions of "in literature" like L263 don't have references.
- Numeric results seem to only show minor improvement over PPO (however this is not directly comparable since the authors' approach provides SPI guarantees).

**Questions:**

- In section 4 when bounding the trust region, the choice of setting 1 < C < 2 seems to be arbitrary, why specifically bound it up to 2?
- What happens in the evaluations when changing the stochasticity of the system, i.e., how do DeepSPI compare to PPO when p_a and n_{NOOP} are closer to zero, and similarly when they are even higher? This would further motivate the presented results as it shows that the system is not susceptible to small parameter changes.

---

> ### Author Response · Authors · 2025-11-21
>
> We thank the reviewer for the positive assessment of our theoretical analysis, the clarity of the presentation, and the relevance of the problem studied. We also appreciate the constructive comments, which helped us clarify several points of the paper. We address them below.
>
> # Weakness #1 (missing references).
>
> Thank you for pointing this out. We agree that additional references would strengthen the background section. The statements around lines 125–126 and 132—134 model-based and representation learning in deep RL. We will carefully recall relevant references from the dedicated “related work” section covering those topics (cf. lines 69–78). For the “in literature” statement around line 263, we will re-cite the related papers detailed earlier in the paper (related work section, line 72).
>
> # Weakness #2 (perceived modest improvements).
> As discussed in our general answer (**G1**), although some environments show similar performance to PPO, DeepSPI brings clear benefits in a large portion of the benchmark. As shown in Figure 3, the improvements are substantial in many games, for example, _BattleZone, ChopperCommand, Enduro, Gopher, Hero, Kangaroo, Tutankham, Venture, Zaxxon_, among others. This is due to the stronger representation learning induced by our auxiliary losses.
> More importantly, achieving **PPO-level performance while adding SPI-type guarantees** should be viewed as a positive outcome rather than a drawback. In many settings, one must choose between strong empirical performance and theoretical control; our approach narrows this gap by retaining competitive results while ensuring the policy cannot degrade in undesirable ways.
>
> We also emphasise that our experimental evaluation is intentionally broad: **61 (!) Atari environments** with highly diverse behaviours and degrees of stochasticity. To the best of our knowledge, this is the largest SPI experimental study to date (and this is enabled thanks to our theory supporting general spaces). Furthermore, as highlighted in [1], ALE remains a valuable benchmark precisely because its diversity exposes algorithmic behaviour across many dynamics, rather than serving only as a SOTA leaderboard. Our study follows this view.
>
> # Question Q1 (choice of C).
> The choice of $C \in \mathopen(1,2 \mathclose)$ is not arbitrary. As shown in Equation 2, selecting $C = 1 + \varepsilon$ enforces
> $$1 - \varepsilon \leq \frac{\pi(a\mid s)}{\pi_b(a\mid s)} \leq 1 + \varepsilon,$$
> which is in the spirit of PPO’s clipping rule. This fixes a “trust-region width” through the parameter $\varepsilon$. Our choice of $C$ is therefore directly aligned with this well-established update rule.
>
> # Question Q2 (effect of stochasticity).
> Our focus is on environments where stochasticity – either in the policy or the transitions – produces undesirable behaviours that can destabilise both learning and representation quality. Evaluating DeepSPI in such settings is meaningful because **safe policy improvement is much more challenging under stochastic behaviors** (either due to the environment or the policy): stochasticity pushes the agent into regions that would otherwise not be visited, making it harder to guarantee that updates do not harm performance (cf. the paper examples).
> Our method applies equally to deterministic environments (as in DeepMDPs [2], originally tested in the pure deterministic setting). Still, the **interesting behaviours and challenges arise precisely in the stochastic regime**. Atari already offers a wide range of behavioral dynamics with nontrivial stochasticity [3,4], making it a suitable and significant testbed. As discussed earlier, this aligns with the perspective of the analysis of ALE in [1].
> For future work, we plan to explore environments with different or higher levels of stochasticity, but we believe our current evaluation already captures the core difficulties that motivate our approach.
>
> ## References
> [1] P.S. Castro: The Formalism-Implementation Gap in Reinforcement Learning Research, arXiv 2025. \
> [2] C. Gelada et al. DeepMDP: Learning Continuous Latent Space Models for Representation Learning, ICML 2019. \
> [3] C.M. Machado et al.: Revisiting the Arcade Learning Environment: Evaluation Protocols and Open Problems for General Agents.” JAIR 2018 \
> [4] M.J. Hausknecht and P. Stone: The Impact of Determinism on Learning Atari 2600 Games. AAAI Workshop: Learning for General Competency in Video Games 2015

---

> > ### Comment · Reviewer_q5uR · 2025-11-27
> >
> > Thank you for the clarifications. I've also seen the comments and issues addressed, that were raised by the other reviewers. Overall, I think this improves the clarity of the paper, and I maintain my rating.
> >
> > I just wanted to clarify regarding Q2, my point was not to do drastic changes in the environment or in the stochasticity of the environment. Merely that by jittering/ablating the amount of added stochasticity while keeping all the other parameters fixed would provide better empirical proof that the proposed approach is not sensitive to tuning/overfitted to the problem.

---

### Author Response · Authors · 2025-11-21
**General Response (Common Concerns)**

We thank all reviewers for their insightful comments. Several concerns were shared across reviews, and we address them here jointly.
# G1. On the perceived weakness of the experimental results

While some reviewers viewed the empirical improvements as modest, we would like to emphasize that DeepSPI provides **substantial gains** in a significant subset of Atari environments, as shown in Fig. 3 (note that we have achieved improvements across the majority, 43/61). These improvements exceed 100% in some cases and are achieved while maintaining **formal SPI-style guarantees**. Obtaining PPO-level performance together with such theoretical control is an important contribution, since SPI-type results are normally obtained at the cost of practical performance and extreme data collection.

More importantly, ALE is used here not as a leaderboard but as a **broad, diverse testbed** to study how representation learning interacts with on-policy updates across **61 (!) environments** with a **broad range of different dynamics and nontrivial stochasticity** [1,6]. This approach aligns with the recent position taken in [1], arguing that ALE remains an appropriate benchmark suite for **understanding the behaviour and limitations of RL algorithms rather than solely chasing SotA scores**.

Finally, comparisons to Dreamer-type results [8] are not directly meaningful. Dreamer is designed for non-Markovian observations and off-policy learning, featuring numerous engineering components such as replay buffers, latent imagination, RSSMs, KL-balancing, and variational reconstruction losses. Our goal is to understand the synergies between on-policy updates, world models, and representation learning, rather than competing with highly engineered off-policy world-model systems.

# G2. On experimental breadth and missing baselines/domains
Our theoretical results apply generally, but **Atari is exactly the kind of setting where representation learning is _critical_**, making it the natural domain to study SPI-consistent representation learning. In contrast, continuous-control domains (e.g., Mujoco) involve **explicit, meaningful state variables** (positions, velocities), making representation learning far less central; hence, less suitable for validating the ideas developed in this work.

Regarding POMDPs: our theory is explicitly Markovian, and extending SPI-style results to POMDPs would be mathematically difficult (complexity-wise, solving MDPs is in P; POMDPs are PSPACE-hard with finite horizon and undecidable in general [2, 3]). We therefore treat this as a **substantial and separate line of future work**.

As for baselines, we deliberately restrict our comparison to **on-policy** methods compatible with vectorized environments and mirror-learning theory (PPO, DeepMDP-style representation losses). Off-policy algorithms such as SAC or DreamerV3 operate under fundamentally different learning dynamics/data regimes that do **not** directly apply to our theoretical setting. Moreover, off-policy replay buffers introduce different wall-clock dynamics (e.g., ~**1h for DeepSPI** on A40 vs. ~**30h for SAC** on A100 [7] and **2 days for Dreamer-v2** on V100 [8], all for 10M steps = 40M frames in Atari), which makes direct comparison misleading. We now state this design choice more clearly.

# G3. On implementation details and runtime comparisons
DeepSPI differs from PPO by:

(i) modifying the advantage to include auxiliary losses,

(ii) adding a transition and reward model.

The policy/value architecture is exactly the PPO architecture from CleanRL [7]; the model networks follow the architecture used in DeepMDPs [5], but implemented via norm-constrained GroupSort architectures [4] to obtain Lipschitzness **without gradient penalties**.

The additional computational cost is modest: enforcing auxiliary losses roughly yields x4 runtime relative to vectorized PPO; an expected overhead given the addition of world-model components. We will add explicit wall-clock measurements to clarify these differences. Essentially, count on ~1h per DeepSPI run vs. ~16 (resp. 21) min per vectorized (resp. sequential) PPO run. As mentioned above, this is marginal compared to the ~1.25 to 2 days per off-policy run (SAC, Dreamer-v2) [7, 8].


We thank the reviewers for highlighting these recurring points. We believe the clarifications above, together with the improvements planned for the final version, significantly strengthen both the empirical and theoretical presentation of the work.

---

> ### Author Response · Authors · 2025-11-21
> **References**
>
> [1] P.S. Castro: The Formalism-Implementation Gap in Reinforcement Learning Research, arXiv 2025\
> [2] C.H. Papadimitriou and J.N. Tsitsiklis: The Complexity of Markov Decision Processes, Math. Oper. Res. 1987\
> [3] O. Madani et al., On the Undecidability of Probabilistic Planning and Infinite-Horizon Partially Observable Markov Decision Problems. AAAI/IAAI, 1999\
> [4] C. Anil et al. Sorting Out Lipschitz Function Approximation. ICML 2019\
> [5] C. Gelada et al. DeepMDP: Learning Continuous Latent Space Models for Representation Learning, ICML 2019\
> [6] C.M. Machado et al.: Revisiting the Arcade Learning Environment: Evaluation Protocols and Open Problems for General Agents, JAIR 2018\
> [7] S. Huang et al.: CleanRL: High-quality Single-file Implementations of Deep Reinforcement Learning Algorithms, JLMR 2022 (for SAC wall clocks, see https://tinyurl.com/ttyvkf28) \
> [8] D. Hafner, et al: Mastering Atari with Discrete World Models. ICLR 2021

---

### Author Response · Authors · 2025-11-27
**List of changes made to the paper**

We thank the reviewers for their recommendations again. We have updated the paper to implement the changes that were requested. We highlight the revisions **in blue** in the paper. Notably, below is a detailed changelog.

# Changelog

- We revised the related-work section to incorporate the line of work on model-based offline RL with SPI-type guarantees, as suggested by Reviewer 9StS, and we added discussion of theoretical abstraction frameworks in MDPs (including the paper highlighted by Reviewer 2xGk, as well as bisimulation relations and pseudometrics)
- We fixed the reference issues highlighted by Reviewer q5uR
- We added a theory roadmap at the end of Section 3, as recommended by Reviewer Y6jS
- We fixed the formatting issue highlighted by Reviewer 2xGk
- We added a clear assumption block in Section 5 as well as a clear reference to the local reward/transition losses to avoid any confusion, as recommended by Reviewer 2xGk
- We added a brand-new didactic experiment with a new environment with raw-pixel visual observations (Section 6.1 and additional details in Appendix G), as requested by Reviewer 2xGk.
- We added a statistical analysis of our claim that the transition loss is improved, as requested by reviewer 2xGk.
- Reviewers Y6jS and 2xGk raised concerns regarding the statistical significance of the DeepSPI results, and Reviewer 9StS was not fully convinced by the empirical benefits of the approach. To reinforce the analysis and provide statistically rigorous evaluations, we followed the recommendations of [9] and updated the evaluation section accordingly; namely:
    - we replaced point estimates, which overlook statistical uncertainty, with 95% confidence intervals computed via stratified bootstrap over the full Atari benchmark;
    - we now report mean, median, interquartile mean (IQM), and optimality gap as aggregate metrics; IQM typically yields tighter confidence intervals than the median, while the optimality gap is more robust to outliers than the mean (we moved the relative-improvement plots to the appendix, but believe the aggregate-metrics figure provides a more statistically meaningful comparison);
    - we added a sample-efficiency curve based on IQM human-normalized scores, again with 95% percentile stratified-bootstrap confidence intervals; and
    - we increased the number of seeds to 8 for all algorithms across the full benchmark.
- We added wall-clock times in Appendix H.1, as requested by Reviewer Y6jS and Reviewer 9StS
- We added additional implementation details in Section 7 and Appendix H, as requested by Reviewer 9StS, notably on the way we enforce the Lipschitz condition, the density functions, as well as the overall architecture of our algorithm. As mentioned in our general response (**G3**), all the remaining parameters are the default CleanRL parameters.
- We added supplementary details on the choice of hyperparameters, as requested by Reviewer Y6jS.

[9] Rishabh Agarwal, Max Schwarzer, Pablo Samuel Castro, Aaron C. Courville, Marc G. Bellemare: Deep Reinforcement Learning at the Edge of the Statistical Precipice. NeurIPS 2021: 29304-29320

---

### Meta-Review · Area_Chair_hozh · 2026-01-07

**Summary:**

The paper proposes DeepSPI, a framework for safe policy improvement (SPI) in online, on-policy reinforcement learning (RL) with learned world models and representation learning. The paper provides a theoretical analysis showing that constraining policy updates to a neighborhood of the current policy yields monotonic improvement and convergence, even in general (i.e., non-tabular) state spaces. A key aspect of this analysis is an explicit link between the quality of the learned representations and local transition and reward prediction losses, resulting in “deep” analogues of classical SPI guarantees. Based on this theory, the paper introduces the DepSPI algorithm, which augments PPO-style updates with auxiliary world-model losses and neighborhood regularization. Empirical results on the ALE benchmark demonstrate that DeepSPI achieves performance comparable to or exceeding PPO and DeepMDP baselines while retaining formal SPI-style guarantees.

The paper was evaluated by four reviewers who agree on several of the paper's key strengths and weaknesses. Among them, they largely agree on the value of the paper's theoretical contributions, particularly the connection that it draws between representation smoothness and policy safety. Several reviewers point to the conceptual value of unifying SPI, mirror-learning updates, and representation learning, as well as the principled interpretation of PPO as a special case within the proposed framework. Additionally, reviewers comment that the paper is well-written and that the theoretical and algorithmic results are clearly presented. At the same time, several reviewers raised concerns about the practical benefits of the DeepSPI algorithm in light of the marginal (at best) performance gains over PPO. Related, there were questions about what can be learned from the experimental results, which at least one reviewer found to be inconclusive. Additionally, several reviewers questioned the adequacy of the baselines, and raised concerns that the experiments were limited to Atari and lacked continuous-control or partially observable domains, and that the paper lacked sufficient statistical analysis and runtime analysis.

**Reviewer Concerns:**

The authors provided a thorough response to the reviewers, addressing the majority of their questions and concerns. These included clarifications to the related work discussion and theoretical results, including the addition of a proof roadmap requested by at least one reviewer. On the empirical side, the authors  strengthened the evaluation and discussion by increasing the number of seeds, adding aggregate metrics (mean, median, IQM, optimality gap) with confidence intervals, reporting sample-efficiency curves and wall-clock runtimes, and providing a statistical analysis of transition-loss improvements. They also clarified design choices regarding Atari as a testbed, the exclusion of off-policy baselines (i.e., DreamerV3), and the computational trade-offs involved. In their follow-up comments, several reviewers noted that their concerns had been addressed and either increased or stated their intention to increase their scores accordingly.

**Reviewer Scores:**

Reviewer q5uR kept their rating of 8. Reviewer Y6jS stated their desire to increase their score to a 6. While Reviewer 2xGk gave the paper a 4, they commented during the discussion that the paper deserves nomination as a spotlight or oral.

---

### Decision · Program_Chairs · 2026-01-26

Accept (Poster)